# Hyperbaric oxygen enhances tumor penetration and accumulation of engineered bacteria for synergistic photothermal immunotherapy

Ke-Fei Xu[1], Shun-Yu Wu[1], Zihao Wang[1], Yuxin Guo[1], Ya-Xuan Zhu[2], Chengcheng Li[3], Bai-Hui Shan[1], Xinping Zhang[1], Xiaoyang Liu[1] & Fu-Gen Wu [1] ✉

Bacteria-mediated cancer therapeutic strategies have attracted increasing interest due to their intrinsic tumor tropism. However, bacteria-based drugs face several challenges including the large size of bacteria and dense extracellular matrix, limiting their intratumoral delivery efficiency. In this study, we find that hyperbaric oxygen (HBO), a noninvasive therapeutic method, can effectively deplete the dense extracellular matrix and thus enhance the bacterial accumulation within tumors. Inspired by this finding, we modify *Escherichia coli* Nissle 1917 (EcN) with cypate molecules to yield EcN-cypate for photothermal therapy, which can subsequently induce immunogenic cell death (ICD). Importantly, HBO treatment significantly increases the intratumoral accumulation of EcN-cypate and facilitates the intratumoral infiltration of immune cells to realize desirable tumor eradication through photothermal therapy and ICD-induced immunotherapy. Our work provides a facile and noninvasive strategy to enhance the intratumoral delivery efficiency of natural/engineered bacteria, and may promote the clinical translation of bacteria-mediated synergistic cancer therapy.

Recent advances in synthetic biology and genetic engineering have led to the emergence of bacteria-mediated therapy, which can serve as a promising and robust cancer treatment modality[1,2]. The hypoxic intratumoral environment has been found to be conducive to the proliferation of facultative anaerobes such as *Escherichia*[3,4] and *Salmonella*[5], as well as obligate anaerobes such as *Clostridium*[6] and *Bifidobacterium*[7], which possess the inherent ability to target solid tumors. In addition to their capacity of tumor-targeted proliferation, bacteria can also activate the antitumor immune response, for instance, by promoting macrophage infiltration[8], thereby enabling them to serve as effective therapeutic agents/carriers. To promote the efficacy of bacteria-based cancer therapy, plasmid transfection and surface modification technologies have been employed to endow bacteria with exogenous and naturally unachievable functions[9–11]. For example, Wang et al. engineered *Escherichia coli* BL21(DE3) to express photothermal melanin by transforming plasmid pET-28a-melA into the bacteria[12]. Meanwhile, anti-programmed death-1 (anti-PD-1) antibodies were modified on the surface of the engineered bacteria to finally realize combined photothermal therapy (PTT) and immunotherapy. Despite the development of diverse bacteria-based therapeutic strategies, the insufficient intratumoral delivery efficiency and low penetration depth of engineered bacteria remain inevitable challenges due

[1]State Key Laboratory of Digital Medical Engineering, Jiangsu Key Laboratory for Biomaterials and Devices, School of Biological Science and Medical Engineering, Southeast University, 2 Southeast University Road, Nanjing 211189, P. R. China. [2]Shanghai Tenth People's Hospital, Tongji University School of Medicine, Shanghai 200072, P. R. China. [3]International Innovation Center for Forest Chemicals and Materials and Jiangsu Co-Innovation Center for Efficient Processing and Utilization of Forest Resources, Nanjing Forestry University, Nanjing 210037, P. R. China. ✉e-mail: wufg@seu.edu.cn

to the large bacterial size, dense extracellular matrix (ECM), and high tumor interstitial pressure, limiting the effectiveness of cancer therapy. Consequently, the development of facile and universal strategies for realizing efficient intratumoral delivery and deep intratumoral penetration of bacteria is still urgently demanded.

Hyperbaric oxygen (HBO) therapy is one of the most efficient methods to overcome tumor hypoxia[13], deplete the ECM[14,15], and promote the penetration of drugs in solid tumors[16]. As a noninvasive Food and Drug Administration (FDA)-approved therapeutic strategy, HBO has been widely utilized as an auxiliary method to improve the therapeutic performance of chemotherapy[17], photodynamic therapy[18], radiotherapy[19], PTT[20], and immunotherapy[21]. For instance, Liu et al. adopted HBO to boost intratumoral delivery of PD-1 antibody and infiltration of T cells to achieve desirable immunotherapy[22]. Although HBO is a convenient and effective tool that has been demonstrated to improve the intratumoral delivery efficiency of nanodrugs[16], leveraging HBO to boost intratumoral accumulation and penetration of bacteria has not been previously reported.

In this work, we use the facultative anaerobic probiotic *Escherichia coli* Nissle 1917 (EcN) as a model bacterium to investigate the influence of HBO on bacterial activity during tumor therapy. EcN, which has been widely utilized in the biomedical field[23–25], can selectively target solid tumors due to the hypoxic tumor microenvironment (TME)[26]. In our hypothesis, HBO can increase the oxygen pressure in blood and tissues, which may guide the facultative anaerobe (EcN) to present improved targeting ability toward hypoxic tumors during HBO treatment. On the other hand, HBO can also deplete dense ECM to enhance intratumoral penetration and accumulation of EcN. As shown in Fig. 1a,

the surface of EcN is decorated with photothermal fluorophores (cypate) via amidation reaction to yield EcN-cypate. Afterwards, the application of HBO treatment promotes the accumulation of EcN-cypate in tumors. Upon near-infrared (NIR) laser irradiation, EcN-cypate achieves PTT and triggered immunogenic cell death (ICD), which sparks systemic immune responses such as dendritic cell (DC) maturation to eradicate tumors (Fig. 1b). Although bacteria-mediated PTT has been reported for tumor therapy[27], the limited infiltration of immune cells constrains the efficacy of bacteria-mediated therapy[28,29]. Importantly, HBO can promote the intratumoral infiltration of immune cells by depleting the ECM to achieve desirable therapeutic results. In addition, we also introduce PD-1 blockade therapy to complement the above bacteria-mediated PTT, and demonstrate that the PD-1 blockade therapy can realize long-term immunosurveillance to inhibit lung metastasis.

## Results

### HBO treatment promotes the tumor accumulation of EcN
Although the bacteria-mediated tumor therapy has attracted much interest from many researchers, realizing efficient intratumoral accumulation and penetration of micrometer-sized bacteria is still challenging, which restricts the further application and clinical translation of these bacteria. According to a previous study, HBO can deplete the ECM to enhance intratumoral drug delivery[22]. However, the influence of HBO on the intratumoral bacterial delivery still remains unexplored. Therefore, we investigated if HBO can deplete the dense ECM to promote the accumulation and penetration of bacteria (EcN) within solid tumors, which is essential for achieving effective bacteria-based cancer

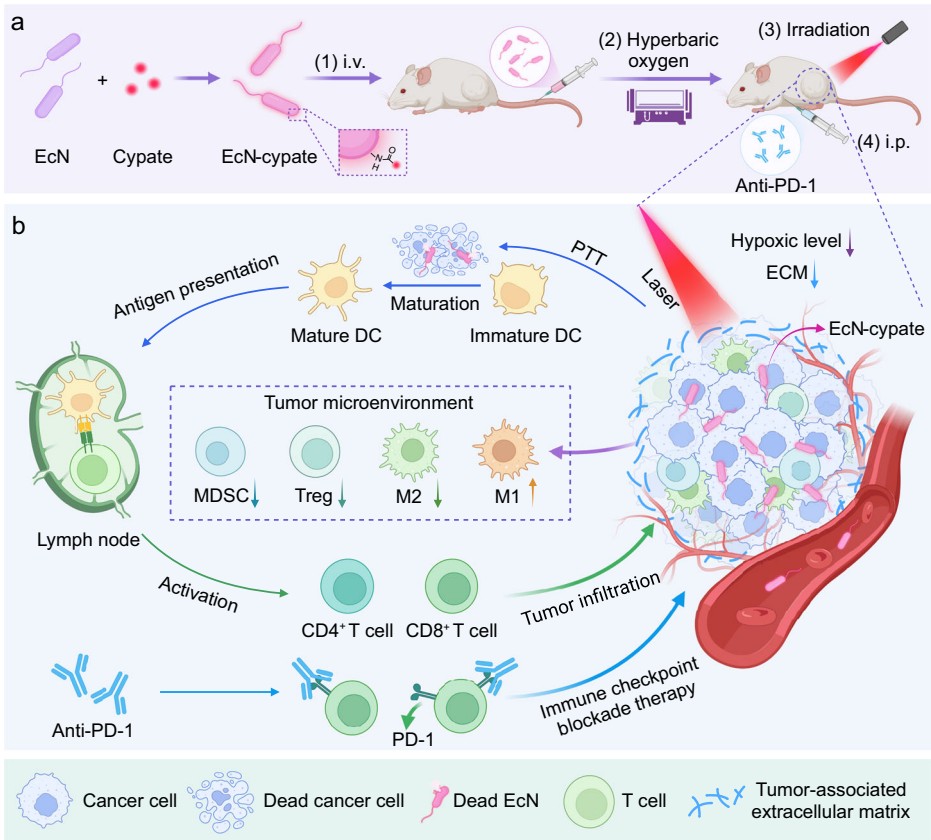

**Fig. 1 | Schematic illustration of HBO-enhanced intratumoral delivery of engineered bacteria for photothermal immunotherapy. a** Scheme showing the preparation of photothermal fluorophores (cypate)-modified bacteria and the procedures for tumor therapy. **b** Scheme showing the engineered bacteria (EcN-cypate)-mediated PTT and HBO (and anti-PD-1)-enhanced systemic immune

responses for tumor eradication. ECM, extracellular matrix; Treg, regulatory T cell; DC, dendritic cell; MDSC, myeloid-derived suppressor cell; M2, M2-type macrophage; M1, M1-type macrophage; PD-1, programmed death-1. **a, b** were created with BioRender.com (publishing license: MC26QF09XE).

therapy (Fig. 2a). To verify the ECM-depleting capacity of HBO, we collected the tumors from the mice in the "HBO+" group (in which the tumor-bearing mice were treated with HBO at 1.5 atmosphere absolute (ATA) for 2 h) and the control group (without HBO treatment, "HBO−") for transcriptomic analysis. In detail, the volcano plot and heat map results revealed that a total of 184 (95 upregulated and 89 down-regulated) differentially expressed genes (DEGs) were detected (Fig. 2b, c), suggesting the significant changes in the tumors after HBO treatment. Moreover, the GO (Gene Ontology) and KEGG (Kyoto

Encyclopedia of Genes and Genomes) enrichment analyses of the tumor tissues were performed. As shown in the GO enrichment analysis results (Fig. 2d), the ECM-related DEGs (e.g., extracellular region, extracellular space, extracellular matrix, external encapsulating structure, and collagen-containing extracellular matrix) and some biological process (BP)-related DEGs (e.g., cell adhesion, biological adhesion, and cell−cell adhesion) were significantly influenced after HBO treatment. The detailed upregulated/downregulated GO enrich-ment analysis results further demonstrated the depletion of ECM via

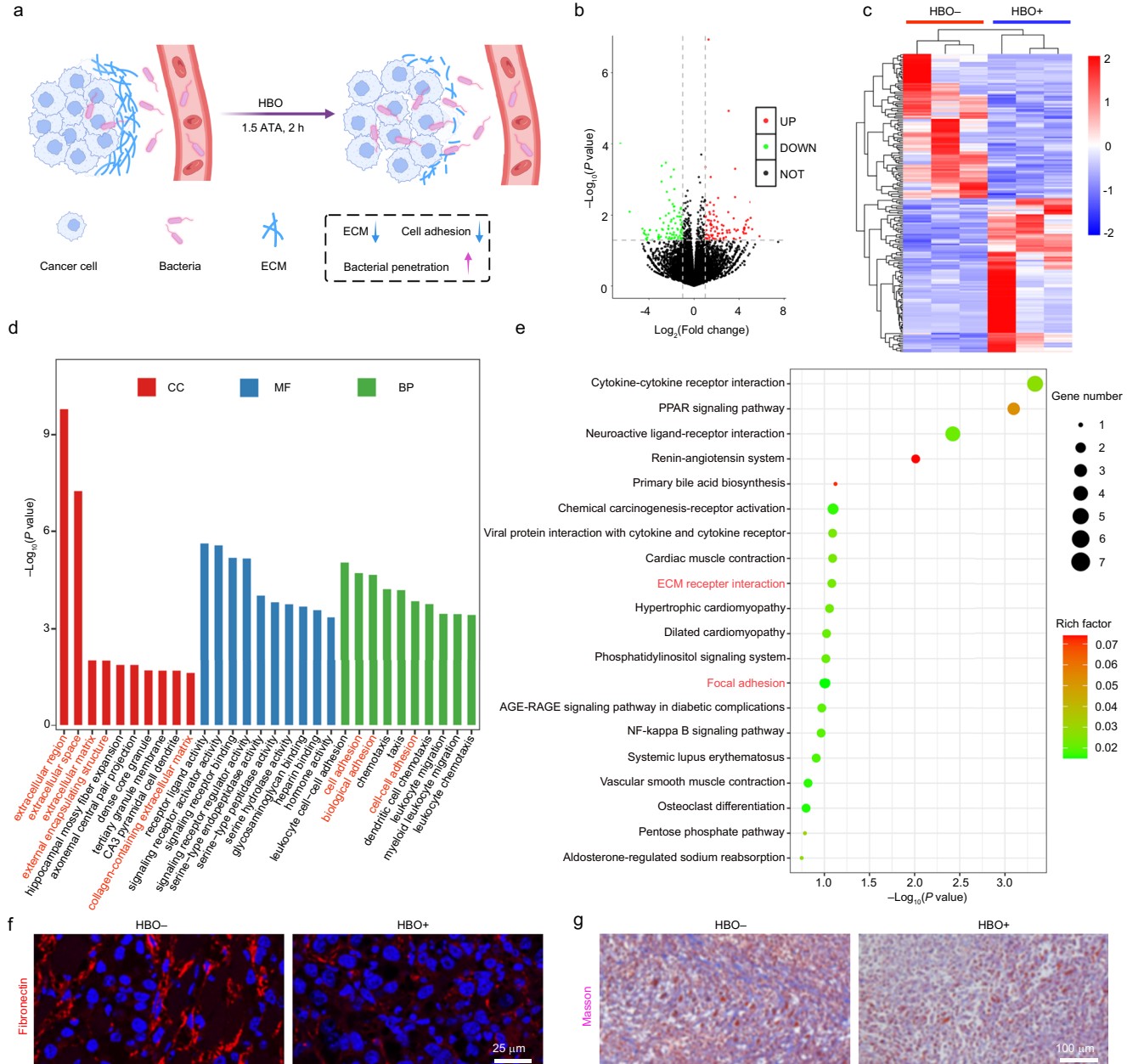

**Fig. 2 | HBO-induced ECM depletion. a** Schematic illustration of HBO-enhanced ECM depletion and intratumoral accumulation of bacteria. **a** was created with BioRender.com (publishing license: HO26QF3LQP). **b**−**e** Transcriptomic analysis results of the tumors from the 4T1 tumor-bearing mice with different treatments. **b** Volcano plot of all DEGs between the HBO-treated group (HBO+) and the control group (HBO−). **c** Heat map of the selected DEGs between the "HBO+" group and the "HBO−" group. Red and blue colors indicate the up-regulation and down-regula-tion, respectively. **d** Histogram presenting the GO enrichment analysis results of some selected DEGs between the "HBO+" group and the "HBO−" group. CC cellular

component, MF molecular function, and BP biological process. **e** Dot plot illus-trating the KEGG enrichment analysis results of some selected DEGs between the "HBO+" group and the "HBO−" group. Statistical significance in **b**, **d**, and **e** was calculated via two-tailed Student's *t* test. *n* = 3 mice. Representative **f** immuno-fluorescence images of fibronectin and **g** Masson's trichrome staining results (which can reflect the collagen contents) in the tumor slices from the mice with different treatments as indicated. The fibronectin and Masson's tri-chrome staining were repeated for three times independently with similar results.

HBO treatment (Supplementary Fig. 1a, b), and the protein–protein interaction (PPI) network revealed the relationships between these DEGs (Supplementary Fig. 2). Additionally, the KEGG enrichment analysis results suggested that the DEGs were enriched in the ECM-related pathways (e.g., ECM receptor interaction) and cell adhesion-associated signaling pathways (e.g., focal adhesion) (Fig. 2e). Furthermore, two major components (fibronectin and collagen) of the ECM were stained, and a significant decrease of fibronectin and collagen was observed after HBO treatment (Fig. 2f, g), further demonstrating the depletion of ECM. Collectively, we demonstrated that HBO can deplete the ECM of tumors and influence the BP of tumor cells (e.g., cell adhesion), which may benefit the intratumoral delivery and penetration of bacteria.

Motivated by the aforementioned findings, we next investigated the impact of HBO on the intratumoral delivery of a facultative anaerobe (EcN). To monitor the distribution of EcN, mCherry-expressing EcN (termed EcN-mCherry), which emits red fluorescence, was chosen as the model bacterium. To conduct in vitro studies, we constructed 3D cultured multicellular spheroids (MCSs) with a diameter of ~500 μm. As illustrated in Fig. 3a, the EcN-mCherry cells were incubated with the MCSs for 12 h, and the confocal images were acquired to record the initial state (0 h) of MCSs. Afterwards, the MCS in the "HBO+" group was subjected to HBO treatment (1.5 ATA, 2 h) and subsequently observed by confocal microscopy. Initially, EcN-mCherry was predominantly located in the outer region of the MCS in both the "HBO−" and "HBO+" groups. However, after HBO treatment, promoted penetration/accumulation of EcN-mCherry was observed in the "HBO+" group, as demonstrated by the significantly increased red dots within the MCS (Fig. 3b). To better verify the effect of HBO, the fluorescence images of MCSs were presented in Supplementary Fig. 3a, and the quantified results further confirmed the enhanced accumulation of EcN-mCherry after HBO treatment (Supplementary Fig. 3b). Owing to their hypoxia-targeting nature, the EcN cells tend to migrate towards oxygen-deficient regions. During the HBO treatment, the oxygen concentration in the external environment of the MCS increased, creating a relatively hypoxic condition within the MCS. Consequently, EcN prefers to penetrate the more hypoxic inner region of the MCS, thereby achieving enhanced accumulation following HBO treatment. To validate our hypothesis, we intravenously (i.v.) injected EcN-mCherry into the 4T1 (a murine mammary carcinoma cell line) tumor-bearing mice, which were further treated with HBO (1.5 ATA, 2 h) at 12 h post injection. Interestingly, a significant increase and broader distribution of EcN-mCherry in the tumor tissue were observed in the HBO-treated group (Fig. 3c), indicating that the HBO treatment facilitated the accumulation and penetration of the bacteria within the TME. Besides, the "63 ×" magnified images from the "HBO+" group displayed a less compact distribution of tumor cells, which was indicated by the number of the cell nuclei that were stained by 4′,6-diamidino-2-phenylindole (DAPI), as compared with the control group (Fig. 3c). This result suggests that HBO may deplete the ECM and influence the cell adhesion, in consistence with the GO enrichment analysis results (Fig. 2d). To further elucidate the effect of HBO on the infiltration of EcN into the tumor, we treated the mice with HBO (1.5 ATA, 2 h) for 3 times at three time points (12, 36, and 60 h) after the injection of $1 \times 10^7$ colony forming units (CFU) EcN, and quantified the bacterial number in the tumors by colony counting. As demonstrated in Fig. 3d, the colony counts of EcN in the HBO-treated groups at 24/48/72 h were substantially higher than those in the control groups, which was in agreement with the statistical outcomes (Fig. 3e). Besides, we demonstrated that the HBO treatment has a negligible influence on the distribution of EcN within the major organs of the treated mice (Supplementary Fig. 3c, d). It is noteworthy that the level of EcN almost reached a plateau at 48 h for the "HBO+" group (Fig. 3e), indicating that only two times of HBO treatment were sufficient to achieve the significantly increased bacterial accumulation within tumors. Specifically,

the average colony count of EcN in the HBO-treated group was ~$1.1 \times 10^7$ CFU/g (tumor tissue) (at 48 h), which was almost 5 times higher than that of the control group (~$2.4 \times 10^6$ CFU/g (tumor tissue)) (at 48 h) (Fig. 3e), demonstrating the crucial role of HBO in elevating the intratumoral bacterial load. Collectively, the above results suggested that HBO is a facile, efficacious, and noninvasive method to improve the penetration and accumulation of natural bacteria (EcN) within solid tumors in a spatiotemporally controllable manner, which may benefit the bacteria-based cancer therapy.

## EcN-cypate retains the viability of EcN and presents excellent PTT effect

The above experiments have demonstrated the potential of HBO treatment for augmenting the penetration and accumulation of natural bacteria (EcN) within solid tumors. However, it remains to be determined whether HBO can exert a similar impact on engineered/modified bacteria for enabling more future clinical applications. Notably, the diverse chemical groups on the surface of bacteria, including carboxyl and amino groups, enable bacteria to be decorated with various synthetic materials via chemical/physical methods[30,31]. Initially, a photothermal agent (cypate) was synthesized according to previous studies[32,33], and the successful synthesis of cypate was verified through electrospray ionization-mass spectrometry (ESI-MS) (Supplementary Fig. 4). To identify the optimal cypate concentration for bacterial modification, we modified EcN with different concentrations of cypate through an amidation reaction to yield EcN-cypate. Remarkably, EcN-cypate displayed the strongest fluorescence signals when the cypate concentration was 1000 μg/mL (Supplementary Fig. 5a–c). Afterwards, the conjugation efficiency of cypate in EcN-cypate was determined to be ~60% according to the ultraviolet–visible (UV–vis) absorption spectra and the standard concentration curve of free cypate (Supplementary Fig. 5d, e). According to the scanning electron microscopy (SEM) results (Fig. 4a), the morphology of EcN-cypate was similar to that of EcN, indicating the negligible disturbance of cypate modification to the bacteria. Subsequently, a series of experiments were carried out to demonstrate the successful conjugation of cypate on the surface of EcN. As shown in Fig. 4b, the red fluorescence of cypate displayed on the surface of the bacteria. In addition, the absolute value of the negative zeta potential of EcN-cypate was increased relative to that of EcN ($-48.5 \pm 4.7$ mV for EcN-cypate versus $-21.7 \pm 3.4$ mV for EcN, Fig. 4c) because each of the conjugated cypate molecules contains two carboxyl groups. Further, EcN-cypate ($1185 \pm 188$ nm) presented an increased hydrodynamic diameter relative to that of EcN ($953 \pm 142$ nm) (Fig. 4d), as well as a similar absorption peak (~790 nm) to that of free cypate (Fig. 4e), verifying the successful cypate conjugation onto the bacteria. Afterwards, we proved that the cypate modification was safe for EcN since the growth curves of EcN and EcN-cypate were very similar (Fig. 4f). Overall, the above results demonstrated that the modified EcN-cypate retains its original morphology and viability, making it a promising candidate for further biomedical applications.

To test the potential of EcN-cypate for PTT, we measured the real-time temperature changes of a suspension containing EcN-cypate using a thermal imaging camera. As illustrated in Fig. 4g, obvious temperature changes were recorded when the EcN-cypate-containing suspension was continuously irradiated with an NIR laser (808 nm, 1 W/cm²) for a time period of 7 min, leading to the morphological alteration of EcN-cypate (Supplementary Fig. 6a). Furthermore, EcN-cypate exhibited multiple heating potential which may enable further repeated PTT (Fig. 4h). Besides, EcN-cypate had a negligible hemolytic effect on red blood cells (Supplementary Fig. 6b), indicating its good hemocompatibility. According to the 3-(4,5-dimethylthiazol-2-yl)-2,5-diphenyl-2H-tetrazolium bromide (MTT) assay results, EcN-cypate presented good cytocompatibility without laser irradiation (Supplementary Fig. 7) and significant cytotoxicity at a concentration

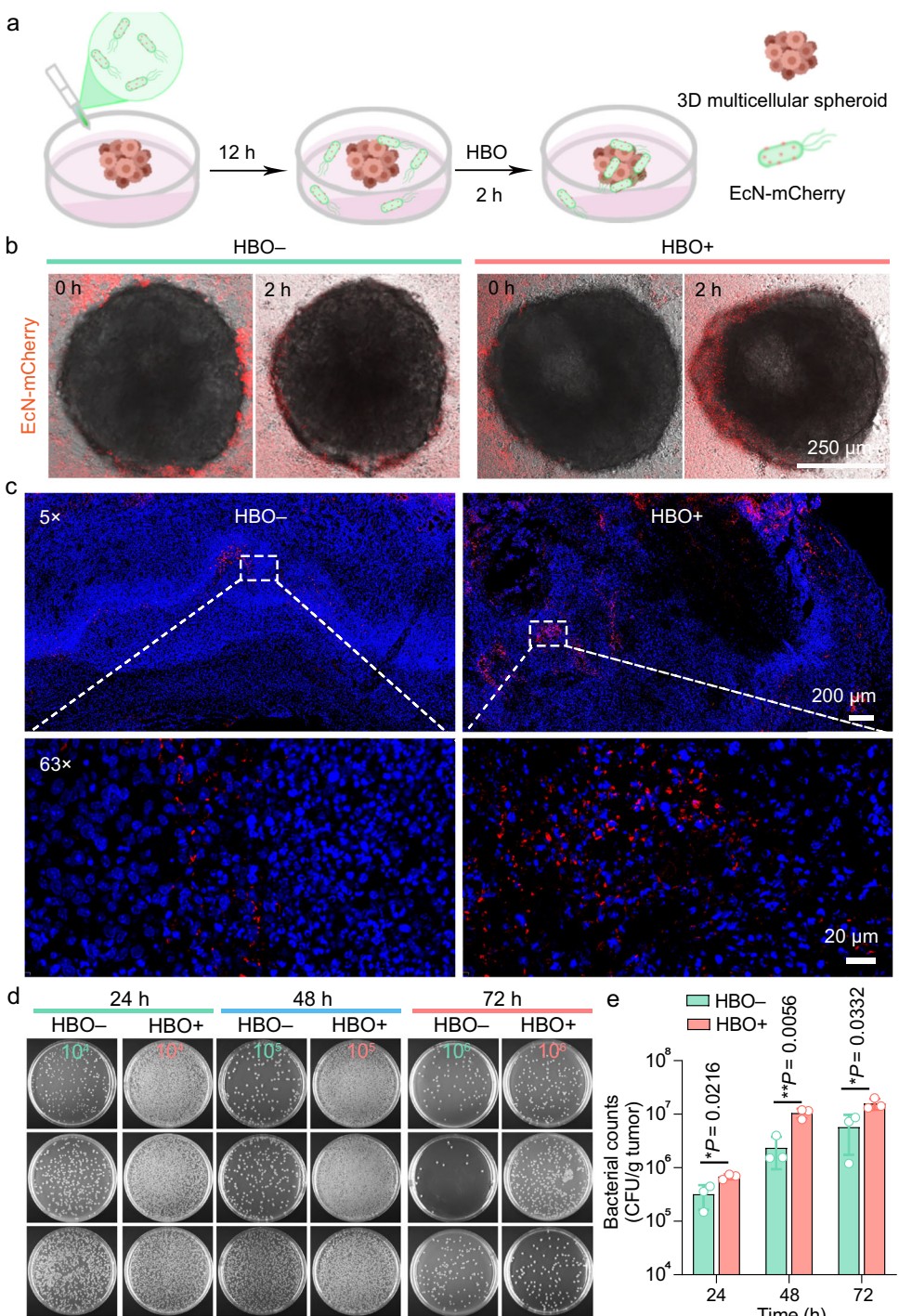

**Fig. 3 | HBO treatment enhances the tumor accumulation of EcN. a** Scheme showing the HBO-enhanced accumulation of EcN (using EcN-mCherry for fluorescence tracking) within an MCS. **b** Confocal images showing the EcN-mCherry-treated MCSs in the absence or presence of HBO treatment (0 or 2 h). **c** Representative fluorescence microscopic images of the tumor tissue slices collected from tumor-bearing mice post intravenous injection of EcN-mCherry without/with HBO treatment. **d** Representative photographs showing EcN colonization amounts inside the tumors from the 4T1 tumor-bearing mice at 24/48/72 h post intravenous injection of EcN without/with HBO treatment. The numbers ($10^4$, $10^5$, $10^6$) in the image indicated the dilution factors of the original bacterial suspensions. **e** Quantitative bacterial counts in per gram of tumor tissues. Data are presented as mean ± standard deviation (SD) and analyzed by two-tailed Student's *t* test (*$P < 0.05$, **$P < 0.01$). $n = 3$ mice. Source data are provided as a Source Data file.

of 5 µg/mL under laser irradiation (Fig. 4i). To clearly visualize the PTT effect of EcN-cypate, 4T1 cells after different treatments were costained with calcein acetoxymethyl ester (calcein-AM) and propidium iodide (PI), and the red fluorescence signals from the dead cells demonstrated the efficient PTT-based cell killing capacity of EcN-cypate (Fig. 4j).

## EcN-cypate-mediated PTT induces ICD of tumor cells

It has been reported that PTT-induced tumor cell death may result in ICD of tumor cells by releasing danger-associated molecular patterns (DAMPs)[34–38]. The immature DCs can then be activated by the DAMPs to become mature DCs, which can stimulate the immune system for eliminating cancer cells[39,40]. In our hypothesis, EcN-cypate-mediated

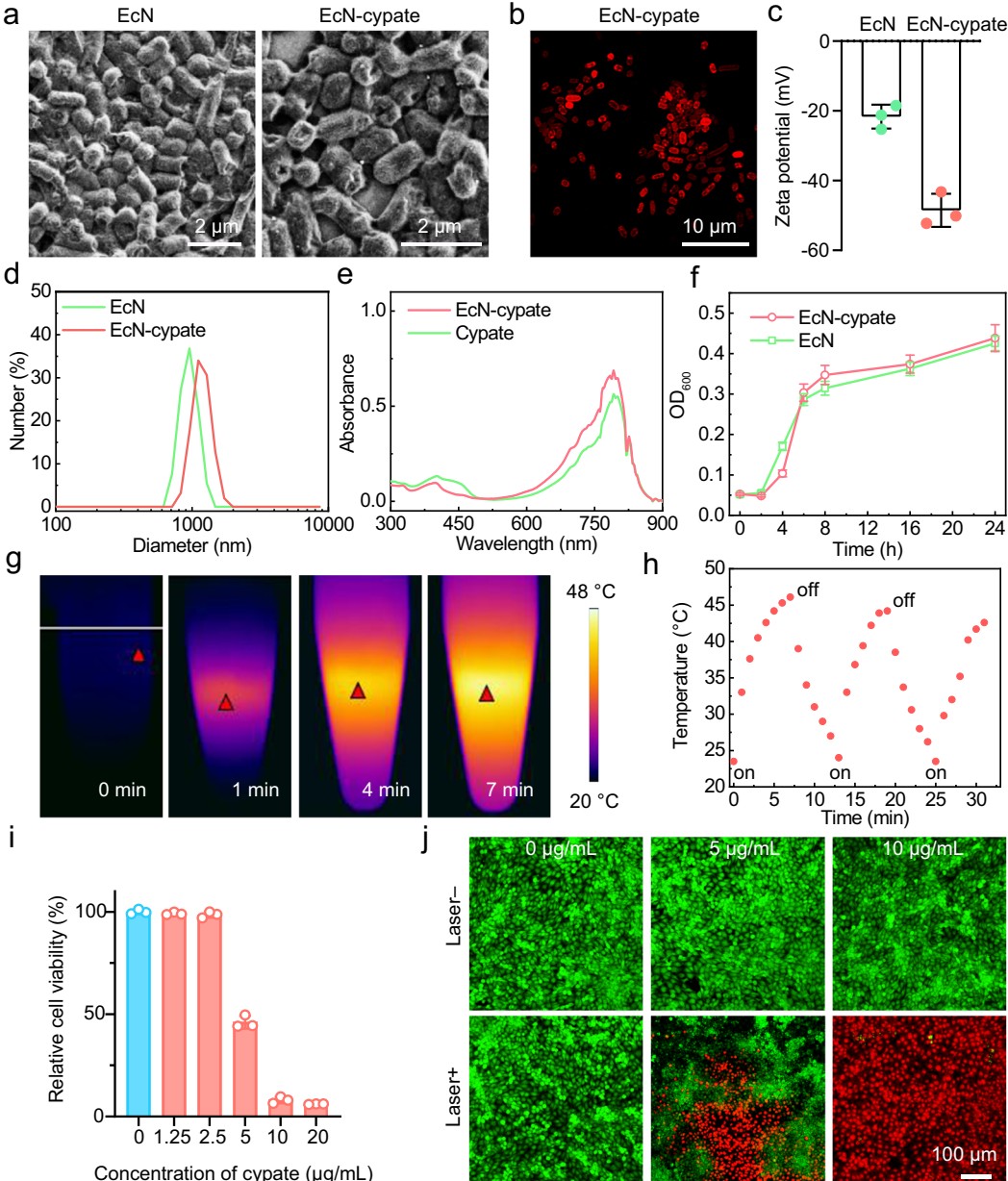

**Fig. 4 | Characterization of EcN-cypate. a** SEM images of EcN and EcN-cypate. **b** Confocal fluorescence image of EcN-cypate. **c** Zeta potentials of EcN and EcN-cypate in phosphate-buffered saline (PBS) (10 mM, pH = 7.4). Data are presented as mean ± SD. $n = 3$ experimental replicates. **d** Hydrodynamic diameters of EcN and EcN-cypate dispersions. **e** UV–vis absorption spectra of EcN-cypate dispersion and cypate solution. **f** Time-dependent growth curves of EcN and EcN-cypate cultured at 37 °C. Data are presented as mean ± SD. $n = 3$ experimental repeats. **g** Thermal images of an EcN-cypate (cypate: 20 μg/mL) dispersion upon NIR laser irradiation (808 nm, 1 W/cm²) for different time periods. **h** Temperature changes of an EcN-cypate (cypate: 20 μg/mL) dispersion during 2.5 cycles of laser on/off treatments. **i** Relative viabilities of 4T1 cells incubated with different concentrations of EcN-cypate upon 808 nm laser irradiation (1 W/cm², 5 min). Data are presented as mean ± SD. $n = 3$ experimental repeats. **j** Confocal fluorescence images of the calcein-AM- and PI-costained 4T1 cells incubated with different concentrations of EcN-cypate without/with 808 nm laser irradiation (1 W/cm², 5 min). Source data are provided as a Source Data file.

PTT can kill cancer cells to release DAMPs, which in turn promotes the maturation of DCs (Fig. 5a). To test this hypothesis, we detected the release of three major DAMPs, namely adenosine triphosphate (ATP), high-mobility group box 1 protein (HMGB1), and calreticulin (CRT), in different experimental groups. Specifically, the concentrations of extracellular ATP and HMGB1 were measured using a luciferin-based ATP assay and an enzyme-linked immunosorbent assay (ELISA), respectively. As shown in Fig. 5b, c, the cells incubated with EcN-cypate followed by laser irradiation ("Laser+" group) exhibited a significant increase in the concentrations of extracellular ATP and HMGB1 compared with other groups. Moreover, the CRT exposure level was also

measured, and the "Laser+" group presented the highest level of CRT (Fig. 5d), with obvious CRT signals detected on the plasma membrane surface (Fig. 5e). The confocal imaging results revealed that the "Laser+" group displayed fewer fluorescence signals inside the cell nuclei compared with the "Laser−" group (Fig. 5f), further validating the secretion of HMGB1 induced by PTT. Collectively, these results demonstrated that EcN-cypate-mediated PTT can effectively cause the release of DAMPs, thus potentially inducing ICD of 4T1 cells.

To study the maturation of DCs, bone marrow-derived dendritic cells (BMDCs) were treated with the medium suspension of the cells from each group (control, "Laser−", and "Laser+"). Afterwards, flow

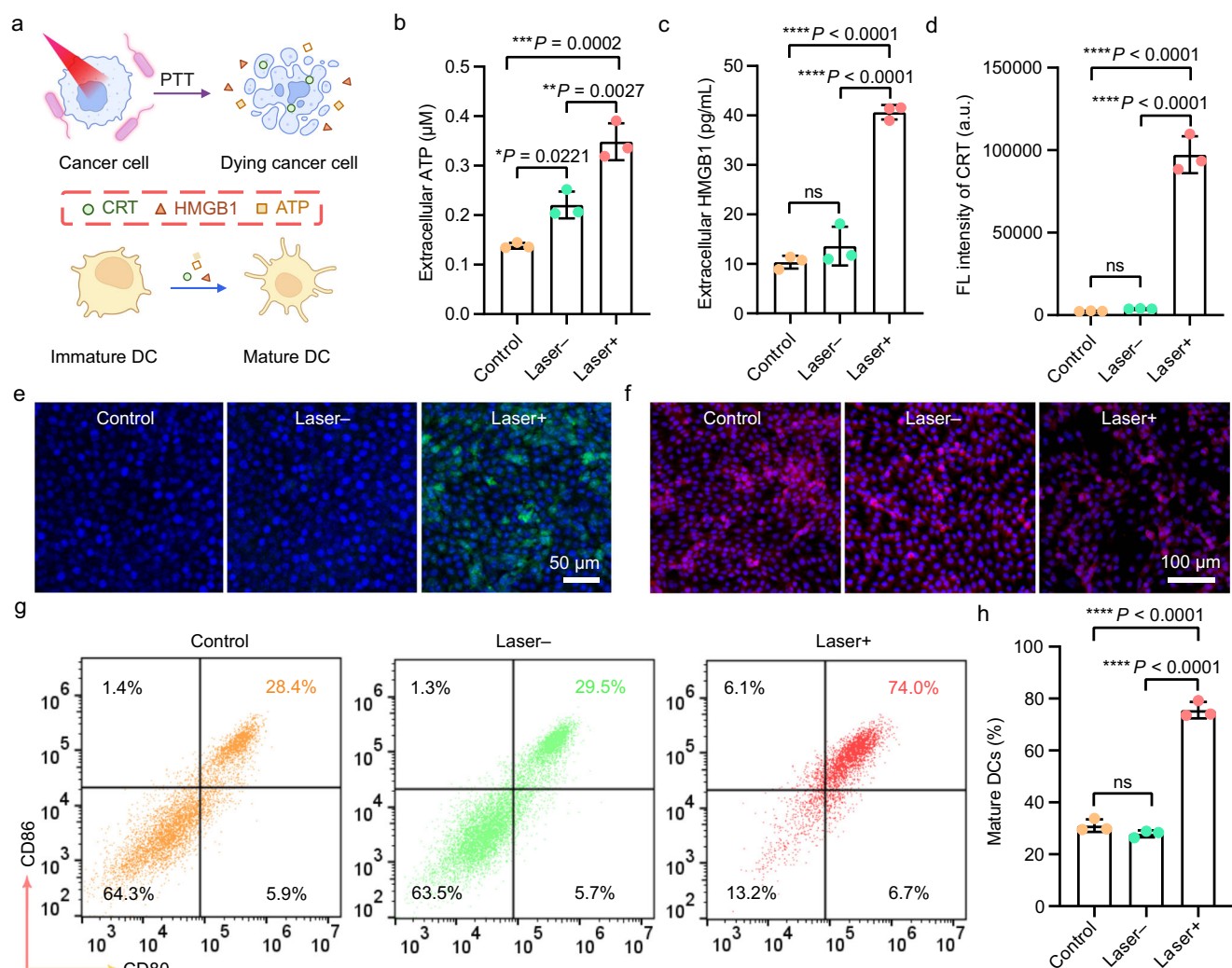

**Fig. 5 | In vitro ICD and immunostimulation triggered by EcN-cypate-mediated PTT. a** Scheme showing the DAMP secretion and ICD-induced BMDC maturation. **a** was created with BioRender.com (publishing license: HO26QF3LQP). Quantification of extracellular release of **b** ATP and **c** HMGB1 from the 4T1 cells with different treatments. **d** Quantification of CRT exposure on the 4T1 cells with different treatments. **b**–**d** Data are presented as mean ± SD. *n* = 3 experimental repeats. Representative immunofluorescence images showing the **e** CRT and **f** HMGB1 in 4T1 cells after different treatments. The cell nuclei stained by Hoechst 33342 present blue fluorescence. The red fluorescence in **e** indicates the presence of HMGB1. The green fluorescence in **f** indicates the presence of CRT.

**g** Representative flow cytometric analysis results of mature DCs (CD11c⁺CD80⁺CD86⁺) and **h** corresponding quantitative statistics of mature DCs after different treatments. Data are presented as mean ± SD. *n* = 3 experimental repeats. "Laser−": The 4T1 cells were incubated with EcN-cypate (cypate: 5 μg/mL) for 4 h without laser irradiation. "Laser+": The 4T1 cells were incubated with EcN-cypate (cypate: 5 μg/mL) for 4 h and then irradiated by an 808 nm laser (1 W/cm², 5 min). Statistical significance in **b**–**d** and **h** was calculated via one-way analysis of variance (ANOVA) with a Tukey's post-hoc test. *$P < 0.05$, **$P < 0.01$, ***$P < 0.001$, ****$P < 0.0001$. "ns" stands for nonsignificant difference. Source data are provided as a Source Data file.

cytometry was utilized to analyze the maturation of BMDCs (CD11c⁺CD80⁺CD86⁺), and the gating strategy was illustrated in Supplementary Fig. 8. The representative results revealed that the "Laser+" group had a significant higher proportion of mature BMDCs (74.0%) than the control (28.4%) and "Laser−" (29.5%) groups (Fig. 5g). Moreover, the corresponding quantitative analysis further confirmed that the percentage of mature BMDCs in the "Laser+" group was the highest (Fig. 5h). Taken together, the EcN-cypate-mediated PTT efficiently eliminated tumor cells and promoted the maturation of DCs via ICD, indicating that the EcN-cypate-mediated PTT may have a potential in vivo antitumor effect.

## EcN-cypate-mediated PTT effectively eradicates subcutaneous tumors
To investigate whether HBO has a comparable impact on engineered bacteria (EcN-cypate) as on natural bacteria (EcN), we detected the in vivo distribution of EcN-cypate in 4T1 tumor-bearing mouse models.

As demonstrated in Fig. 6a, EcN-cypate-treated group presented more effective intratumoral delivery of cypate than the free cypate-treated group, which might be attributed to the hypoxia-targeting ability of EcN. Additionally, HBO treatment significantly improved the intratumoral delivery of EcN-cypate, with the maximal fluorescence intensity observed at 48 h post injection (Fig. 6a). The ex vivo images of major organs revealed that cypate was primarily cleared by liver (Fig. 6a). On the other hand, the semiquantitative analysis of cypate distribution further validated the crucial role of HBO in promoting the delivery of the bacteria-based drug system (Fig. 6b). The intratumoral fluorescence of cypate persisted for up to one week in the "EcN-cypate + HBO" group, indicating that HBO can largely prolong the tumor retention of the bacterial drug (Supplementary Fig. 9). After HBO treatment, the expression of hypoxia-inducible factor-1α (HIF-1α) decreased (Fig. 6c), indicating that HBO overcame the tumor hypoxia. Notably, the tumor region still displayed the maximal fluorescence signals of HIF-1α relative to major organs (Supplementary Fig. 10), indicating that EcN-

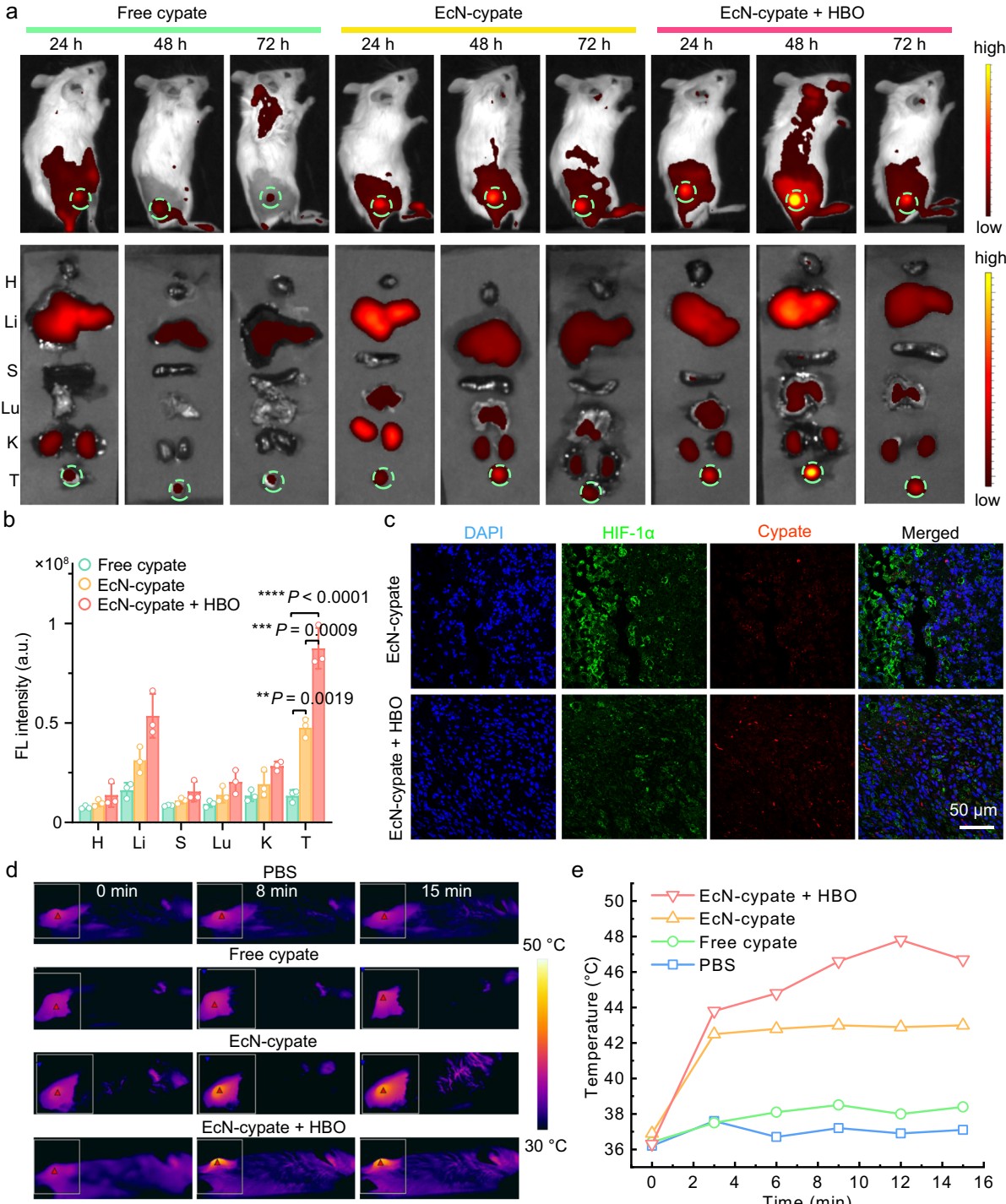

**Fig. 6 | Evaluation of the biodistribution and in vivo PTT effect of EcN-cypate.** **a** In vivo and ex vivo distributions of cypate at different time points (24, 48, and 72 h) post intravenous injection of cypate or EcN-cypate. "EcN-cypate + HBO": The mice were treated with HBO (1.5 ATA, 2 h) post injection of EcN-cypate. **b** Semiquantitative distribution results of cypate in the tumors and major organs at 48 h post injection in different groups. Data are presented as mean ± SD. $n = 3$ mice. The abbreviations in **a** and **b** represent different organs/tissues as follows: H: heart, Li: liver, S: spleen, Lu: lung, K: kidneys, and T: tumor. **c** Representative immunofluorescence images of HIF-1α expression in tumor slices. This experiment was repeated for three times independently with similar results. **d** Representative thermal images of tumor-bearing mice with different treatments after receiving NIR laser irradiation (808 nm, 1 W/cm$^2$) for different time periods. **e** Temperature changes of the tumor areas in **d**. The doses of cypate in free cypate, EcN-cypate, and "EcN-cypate + HBO" groups were 10 mg/kg. Statistical significance in **b** was calculated via one-way ANOVA with a Tukey's post-hoc test. **$P < 0.01$, ***$P < 0.001$, ****$P < 0.0001$. Source data are provided as a Source Data file.

cypate can still target the hypoxic tumor after HBO treatment. Next, we assessed the in vivo PTT effect of EcN-cypate. As shown in Fig. 6d, the EcN-cypate and "EcN-cypate + HBO" groups both showed remarkable tumoral temperature elevation after NIR laser irradiation within 15 min.

Specifically, the temperature in the "EcN-cypate + HBO" group could increase to ~48 °C during irradiation (Fig. 6e), which is sufficient for realizing effective PTT. To sum up, these results demonstrated that HBO can enhance the intratumoral delivery efficiency of engineered

bacteria (EcN-cypate), and the HBO-promoted bacterial delivery strategy can help achieve enhanced PTT efficacy.

Encouraged by the desirable HBO-enhanced PTT effect, we further investigated the in vivo antitumor performance of such an HBO-combined PTT treatment. As illustrated in Fig. 7a, we sequentially applied HBO (1.5 ATA, 2 h) and laser irradiation (808 nm, 1 W/cm², 15 min) to the EcN-cypate-injected mice, and repeated the above procedure on the second day to enhance the therapeutic effect. As shown in Fig. 7b–g, the "EcN-cypate + HBO + laser" group exhibited the best therapeutic outcome among all the groups, validating the high antitumor effectiveness of the combination of HBO and EcN-cypate-mediated PTT. Additionally, the "EcN-cypate + HBO" group displayed a

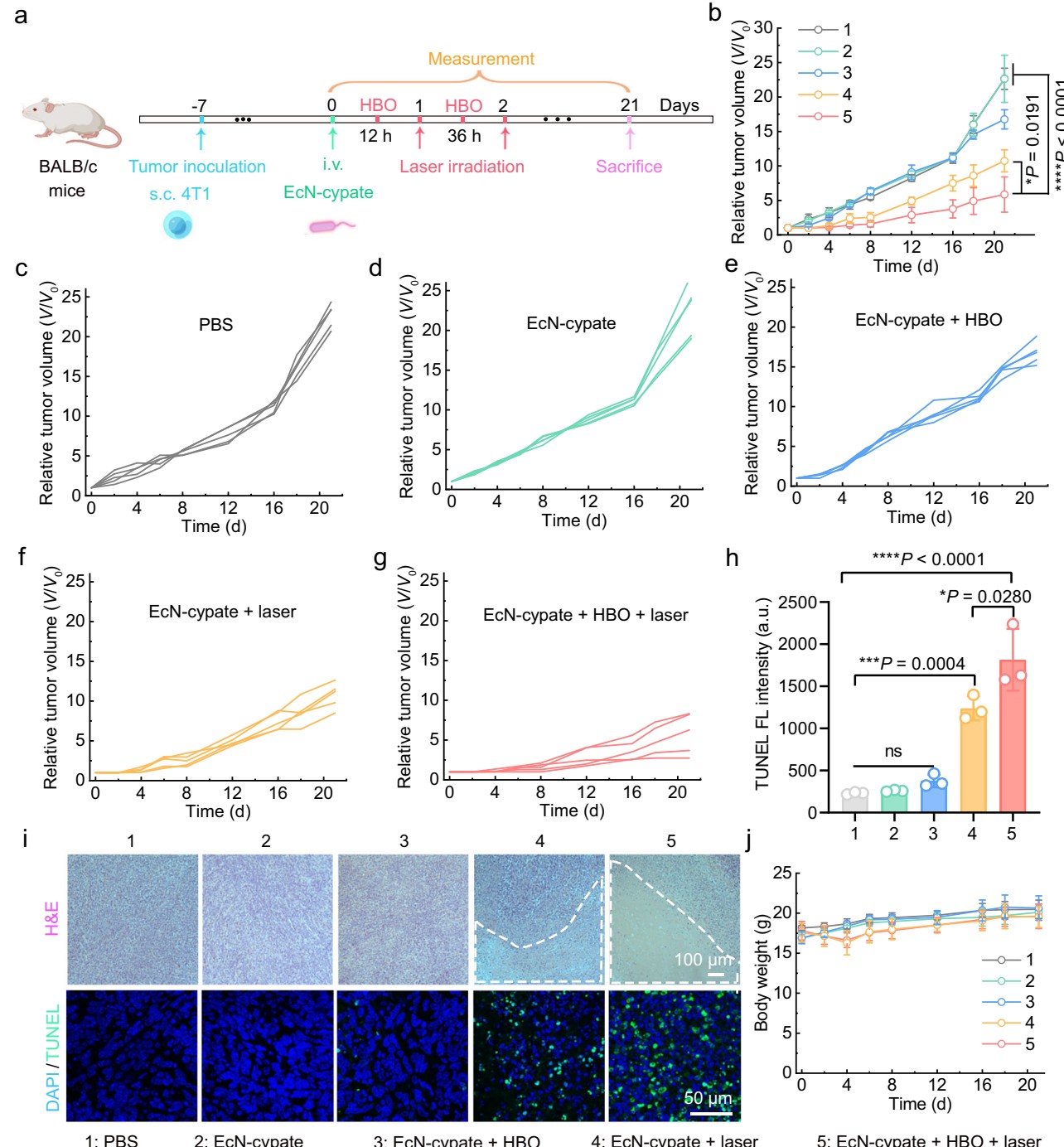

**Fig. 7 | HBO-enhanced PTT for treating subcutaneous 4T1 tumors.**
**a** Experimental outline showing the procedures for evaluating the HBO-coupled PTT effect in 4T1 tumor-bearing mice. **b** Relative tumor volumes of 4T1 tumor-bearing mice after different treatments. Data are presented as mean ± SD. $n = 5$ mice. Detailed tumor growth curves of each 4T1 tumor-bearing mouse in the **c** PBS, **d** EcN-cypate, **e** "EcN-cypate + HBO", **f** "EcN-cypate + laser", and **g** "EcN-cypate + HBO + laser" groups, respectively. **h** Quantitative TUNEL fluorescence intensities of the tumor slices in different groups. Data are presented as mean ± SD. $n = 3$ mice.

**i** Representative H&E- and TUNEL assay kit-stained tumor slices from the 4T1 tumor-bearing mice after different treatments. The damaged regions in the H&E-stained images were marked by white lines. This experiment was repeated for three times independently with similar results. **j** Body weight fluctuations of the mice in different groups. Data are presented as mean ± SD. $n = 5$ mice. Statistical significance in **b** and **h** was calculated via one-way ANOVA with a Tukey's post-hoc test. $*P < 0.05$, $***P < 0.001$, $****P < 0.0001$. "ns" stands for nonsignificant difference. Source data are provided as a Source Data file.

stronger antitumor effect than the EcN-cypate group (Fig. 7b, d, e), indicating that HBO alone can slightly inhibit tumor growth.

To visualize the damage in the tumor regions caused by different treatments, we stained the tumor slices with hematoxylin and eosin (H&E) and terminal deoxynucleotidyl transferase (TdT)-mediated dUTP nick-end labeling (TUNEL) assay kit, respectively. According to the quantitative TUNEL assay kit-staining results, the "EcN-cypate + HBO + laser" group revealed the most TUNEL fluorescence signals (Fig. 7h), which was also reflected by the representative fluorescence images (Fig. 7i), indicating the most apoptotic cells in this group. The H&E-stained images showed that the "EcN-cypate + HBO + laser" group exhibited the largest damaged region marked by the white lines (Fig. 7i). Taken together, these staining results further confirmed the excellent therapeutic performance of HBO-coupled PTT treatment.

We then evaluated the in vivo biosafety of EcN-cypate. The curves of body weight presented negligible fluctuation among all the groups (Fig. 7j), and the H&E staining results of the major organs from EcN-cypate-injected mice did not have notable difference compared with those from the PBS-treated mice (Supplementary Fig. 11a), revealing the good biocompatibility of EcN-cypate. Besides, the hemanalysis and biochemical analysis results also confirmed the excellent biocompatibility of EcN-cypate (Supplementary Fig. 11b–n).

## HBO-coupled and EcN-cypate-mediated PTT (HBO-coupled PTT) reprograms TME

During tumor progression, macrophages are recruited into tumors to mainly promote tumor growth, and these tumor-associated macrophages (TAMs) are roughly classified into two phenotypes: M1-like and M2-like TAMs[41]. In detail, M1-like TAMs exhibit pathogen clearance and antitumor immunity effects, while M2-like TAMs influence antiinflammatory response and protumorigenic properties[42,43]. Hence, promoting the polarization toward M1-like TAMs can improve antitumor immunity and reprogram the immunosuppressive TME. According to previous studies[44,45], PTT can modulate immune response to realize the repolarization of M2-like TAMs to M1-like TAMs. Encouraged by the promising therapeutic results of HBO-enhanced PTT as shown above, we conducted a comprehensive investigation of the in vivo immune responses to evaluate whether HBO therapy can enhance PTT-induced immune cell activation to realize the reprogramming of the TME for achieving potentiated immunotherapy. As illustrated in Fig. 8a, the 4T1 tumor-bearing mice in the "EcN-cypate + HBO + laser" group received HBO treatment (1.5 ATA, 2 h) at 12 and 36 h followed by a 15 min continuous laser irradiation (808 nm, 1 W/cm$^2$) at 24 and 48 h, and the resulting PTT results were captured via photographing (Supplementary Fig. 12). 8 d after the administration of EcN-cypate, we collected blood, tumors, spleens, and tumor-draining lymph nodes (TDLNs) from the mice to assess the immune responses, including polarization of TAMs, maturation of DCs, activation of T cells, and secretion of cytokines. To begin with, we detected the levels of cytokines in the tumor and serum samples with ELISA, and found that the "EcN-cypate + HBO + laser" group showed significantly increased levels of proinflammatory cytokines such as interferon-γ (IFN-γ), interleukin-1β (IL-1β), interleukin-6 (IL-6), and tumor necrosis factor-α (TNF-α) (Supplementary Fig. 13a and b), indicating the polarization of TAMs towards the M1-like phenotype. Furthermore, the level of antiinflammatory interleukin-10 (IL-10) was significantly reduced after treatment (Supplementary Fig. 13a and b), suggesting a decrease of M2-like macrophages. This result was further confirmed by the reduced red fluorescence signals of CD206, a marker of M2-like TAMs, in tumor tissues (Supplementary Fig. 13c). Then, we visualized the CRT expression to assess the in vivo ICD effect, and found that the "EcN-cypate + HBO + laser" group exhibited the strongest fluorescence signals, indicating the highest ICD level among all the groups (Supplementary Fig. 13c). Besides, cytotoxic CD8$^+$ T cells and two

immunosuppressive cells (myeloid-derived suppressor cells (MDSCs) and regulatory T cells (Tregs)) were also examined. The expression of granzyme B produced by CD8$^+$ T cells[46,47] was detected in the tumor tissues, and the "EcN-cypate + HBO + laser" group presented the highest fluorescence signals (Supplementary Fig. 13c). In contrast, fewer fluorescence signals of FoxP3 (the marker of Tregs) and Ly6G (the marker of MDSCs) were observed after treatment (Supplementary Fig. 13c). Taken together, these results preliminarily demonstrated the potential of HBO-coupled PTT for stimulating immunological responses in 4T1 tumor-bearing mice.

To comprehensively evaluate the systemic antitumor immune responses, we utilized flow cytometry to analyze the immune cells (i.e., M1-like TAMs, DCs, and T cells) in tumors, spleens, and TDLNs. The representative flow cytometric results indicated the similar polarization degrees of M1-like TAMs (F4/80$^+$CD11c$^+$) in the PBS, EcN-cypate, and "EcN-cypate + HBO" groups, with the corresponding M1-like TAM rates of 25.7%, 28.5%, and 23.1%, respectively (Supplementary Fig. 14), indicating that the TMEs in these groups were immunosuppressive. However, the levels of the M1-like TAMs in the "EcN-cypate + laser" and "EcN-cypate + HBO + laser" groups were higher than those in the corresponding non-laser-irradiated groups (EcN-cypate, and "EcN-cypate + HBO" groups) (Supplementary Fig. 14), which was confirmed by the quantitative analysis results (Fig. 8b), suggesting that the PTT effect can facilitate the M1-like polarization of TAMs. Furthermore, the levels of M1 macrophages in spleens and TDLNs were also considerably increased in the "EcN-cypate + HBO + laser" group (Fig. 8d, f), indicating that HBO-coupled PTT could elicit desirable systemic immune responses. We also evaluated the levels of mature DCs (CD11c$^+$CD80$^+$CD86$^+$) in tumors, spleens, and TDLNs that can present antigens to T cells, and the representative flow cytometric plots were presented in Supplementary Fig. 15. As expected, the highest level of mature DCs was found in the "EcN-cypate + HBO + laser" group (Fig. 8c, e, g), which was possibly due to the PTT-induced ICD effect. Encouraged by these results, we further determined the levels of cytotoxic T cells (CD3$^+$CD8$^+$) and activated CD4$^+$ helper T cells (CD3$^+$CD4$^+$) in TDLNs. The flow cytometric results revealed a significant increase in the proportions of CD8$^+$ and CD4$^+$ T cells in the "EcN-cypate + HBO + laser" group (Fig. 8h–j). Not surprisingly, the "EcN-cypate + HBO + laser" group exhibited the highest proportions of CD8$^+$ and CD4$^+$ T cells (26.4% and 64.1%, respectively), which were much higher than those of the control group (6.0% CD8$^+$ T cells and 17.6% CD4$^+$ T cells) (Fig. 8j), demonstrating that the HBO-combined PTT can substantially increase the levels of cytotoxic and helper T cells, which may help to realize effective tumor immunotherapy. Notably, the "EcN-cypate + HBO + laser" group presented a significantly higher level of the M1-like TAMs and mature DCs in the tumor regions compared with the "EcN-cypate + laser" group (Fig. 8b and c), possibly indicating the potential role of HBO treatment in enhancing the infiltration of immune cells. Additionally, the immunofluorescence staining results revealed the highest fluorescence signals of the CD8$^+$ T cells in the "EcN-cypate + HBO + laser" group (Fig. 8k), indicating the evident promotion effect of HBO treatment on the intratumoral infiltration of cytotoxic T cells. Collectively, the above results demonstrated that HBO-combined PTT may not only activate the systemic antitumor responses but also facilitate the intratumoral infiltration of immune cells via HBO-induced ECM depletion.

To explore the killing efficacy of PTT towards the engineered bacteria (EcN-cypate), we investigated the remaining viable EcN cells in the tumor post PTT. As shown in Supplementary Fig. 16a, the viability of EcN significantly decreased after a single laser irradiation ("Laser+" group), and few EcN cells were still alive following a second laser exposure ("Laser++" group) (Supplementary Fig. 16b). We speculated that the dead bacteria caused by repeated PTT could enhance the immune responses induced by ICD. To study the antitumor and immunostimulation capabilities of repeated PTT, we

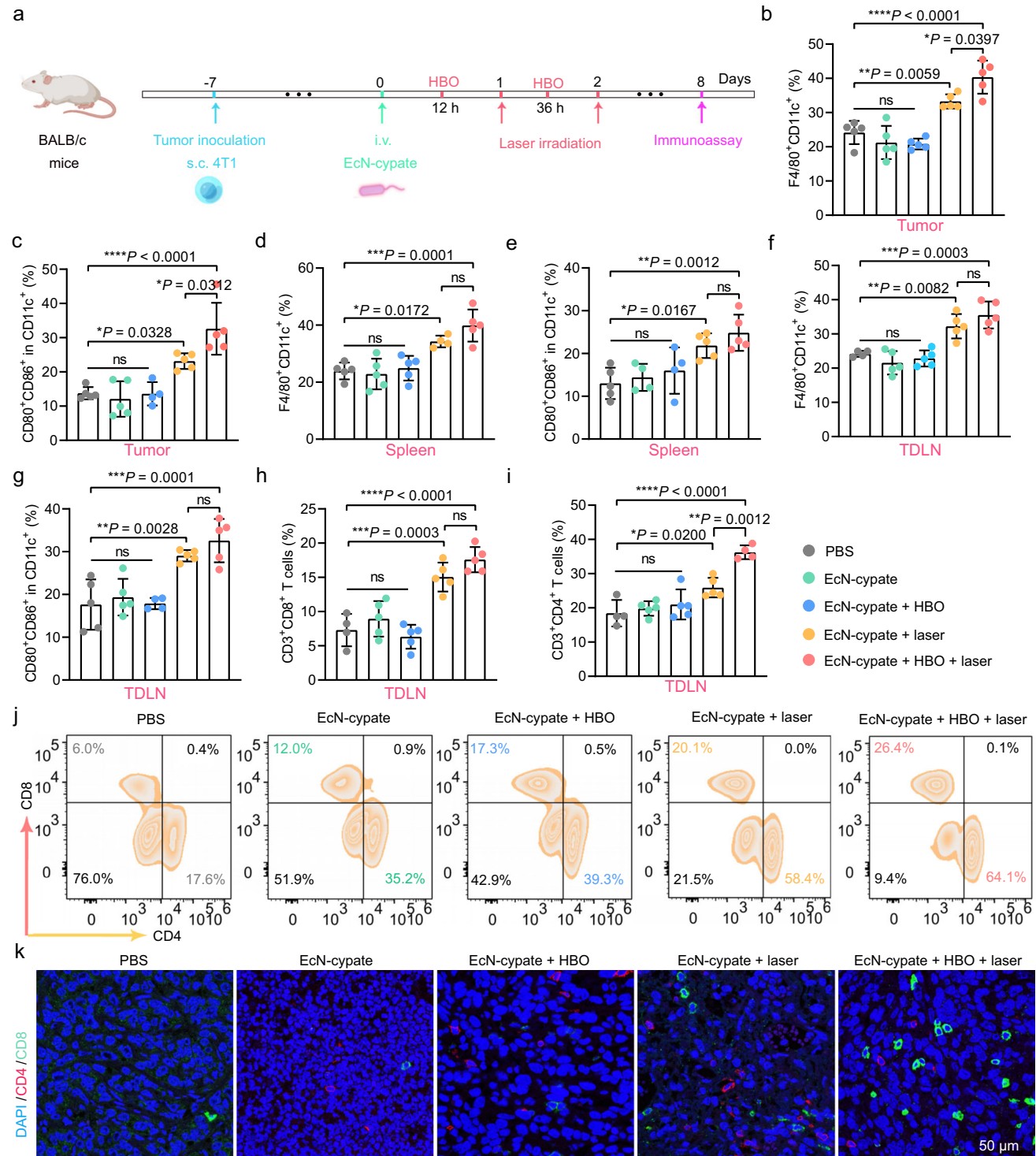

**Fig. 8 | Analysis of the antitumor immune responses triggered by HBO-enhanced PTT. a** Schematic illustration of the experimental schedule. Flow cytometric results showing the levels of **b** M1 macrophages (F4/80⁺CD11c⁺) and **c** mature DCs (CD11c⁺CD80⁺CD86⁺) in the tumors. Flow cytometric results showing the levels of **d** M1 macrophages (F4/80⁺CD11c⁺) and **e** mature DCs (CD11c⁺CD80⁺CD86⁺) in the spleens. Flow cytometric results showing the levels of **f** M1 macrophages (F4/80⁺CD11c⁺), **g** mature DCs (CD11c⁺CD80⁺CD86⁺), **h** cytotoxic T cells (CD3⁺CD8⁺), and **i** activated CD4⁺ helper T cells (CD3⁺CD4⁺) in the TDLNs. **b–i** Data are presented as mean ± SD. *n* = 5 mice (when the number of immune cells collected from a certain mouse was too low, this mouse was not used for data collection). **j** Representative flow cytometric analysis results of helper T cells (CD3⁺CD4⁺) and cytotoxic T cells (CD3⁺CD8⁺) in the TDLNs. **k** Representative confocal fluorescence images of the immunofluorescence staining results of CD4⁺ (as indicated by red fluorescence) and CD8⁺ (as indicated by green fluorescence) T cells in tumor slices. This experiment was repeated for three times independently with similar results. The tumors, spleens, and TDLNs were collected from the 4T1 tumor-bearing mice in different groups as indicated. Statistical significance in **b–i** was calculated via one-way ANOVA with a Tukey's post-hoc test. *P < 0.05, **P < 0.01, ***P < 0.001, ****P < 0.0001. "ns" stands for nonsignificant difference. Source data are provided as a Source Data file.

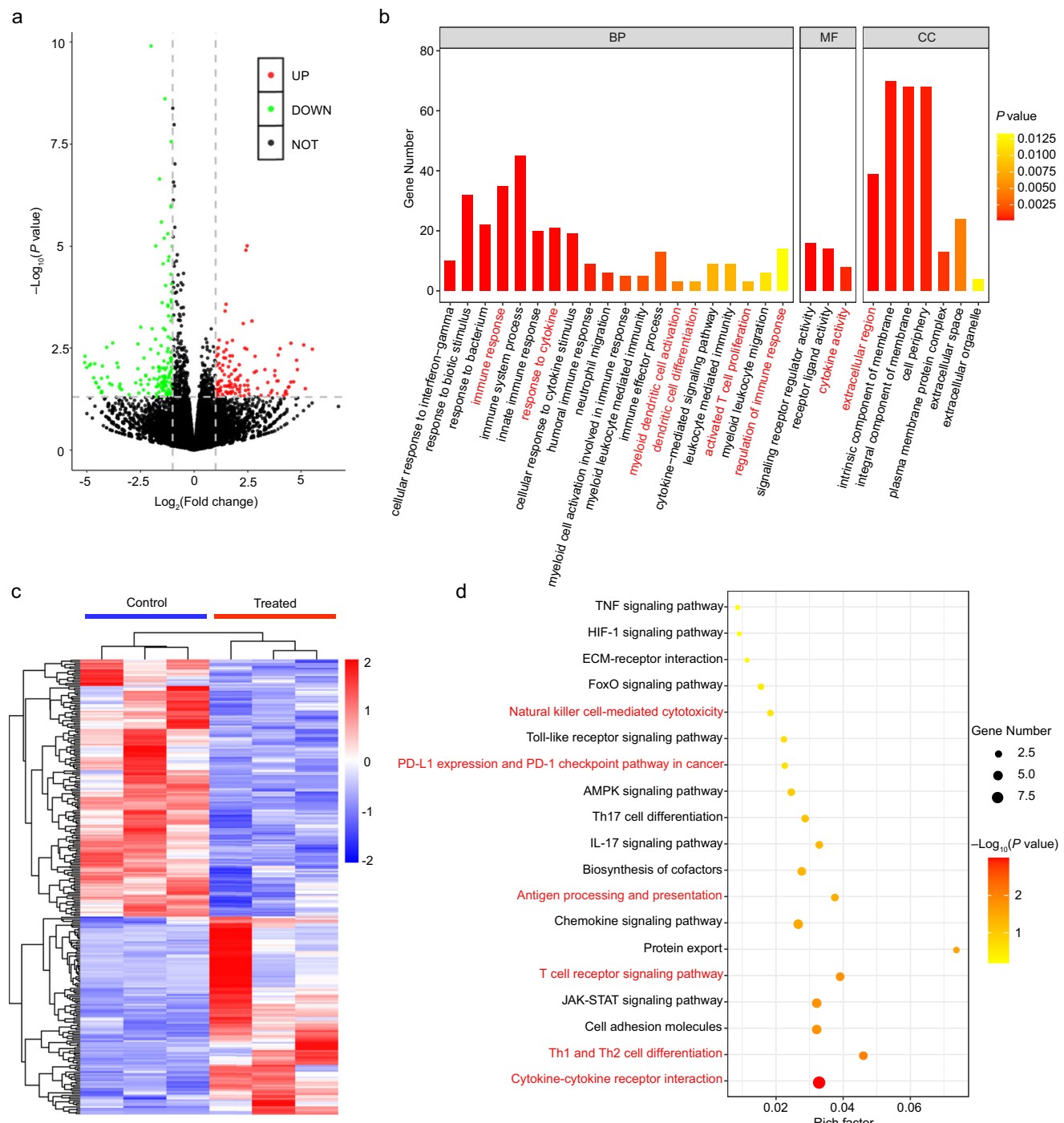

**Fig. 9 | Transcriptomic analysis of 4T1 tumor tissues before and after HBO-coupled PTT treatment. a** Volcano plot of all the DEGs between the control group and the treated ("EcN-cypate + HBO + laser") group. **b** Histogram presenting the GO enrichment analysis results of some selected DEGs between the control group and the treated group. **c** Heat map of the selected DEGs between the control group and the treated group. Red and blue colors indicate the up-regulation and down-regulation, respectively. **d** Dot plot illustrating the KEGG enrichment analysis results of some selected DEGs between the control group and the treated group. Statistical significance in **a**, **c**, and **d** was calculated via two-tailed Student's $t$ test. $n = 3$ mice.

recorded the growth curves of tumors with different treatments (Supplementary Fig. 16c–e). Notably, the "Laser++" group exhibited superior therapeutic results (Supplementary Fig. 16f), highlighting the crucial role of repeated laser irradiation in realizing a satisfactory antitumor outcome. Furthermore, the quantified results of tumor weights and H&E/TUNEL staining results further validated an enhanced tumor killing effect after repeated PTT (Supplementary Fig. 16g, h). Furthermore, we assessed the level of lung metastasis

after PTT treatment. As shown in Supplementary Fig. 16i, fewer tumor nodules were observed in the "Laser++" group, which was confirmed by the corresponding quantitative data (Supplementary Fig. 16j). Besides, we also evaluated the distributions of $CD3^+CD4^+$ and $CD3^+CD8^+$ T cells within tumors, and the "Laser++" group also presented the highest levels of these T cells (Supplementary Fig. 16k), indicating the potentiated immune responses induced by repeated PTT. Collectively, repeated PTT effectively eliminated bacteria and

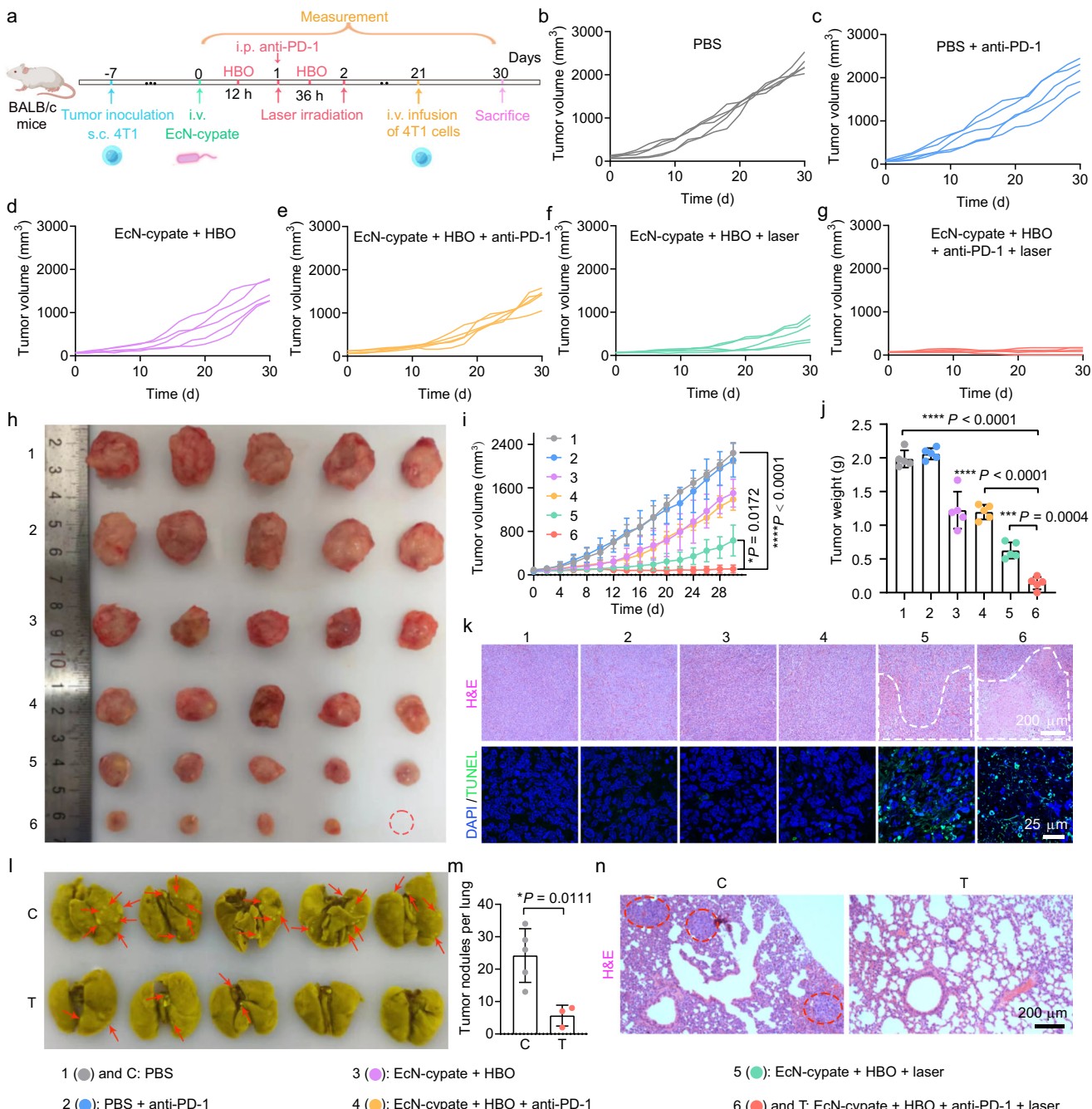

**Fig. 10 | Metastasis prevention via PTT (coupled with HBO)-induced and anti-PD-1-enhanced long-term immune effects. a** Scheme showing the experimental schedule. Detailed tumor growth curves of each 4T1 tumor-bearing mouse in the **b** PBS, **c** "PBS + anti-PD-1", **d** "EcN-cypate + HBO", **e** "EcN-cypate + HBO + anti-PD-1", **f** "EcN-cypate + HBO + laser", and **g** "EcN-cypate + HBO + anti-PD-1 + laser" groups, respectively. **h** Photographs of the tumor tissues collected from the 4T1 tumor-bearing mice at day 30 after different treatments. **i**, Tumor volume changes of 4T1 tumor-bearing mice in different groups. **j** Tumor weights in different groups at day 30. **i**, **j** Data are presented as mean ± SD. *n* = 5 mice. **k** Representative H&E- and TUNEL assay kit-stained tumor slices of 4T1 tumor-bearing mice after different treatments. The damaged regions in the H&E-stained images were marked by white lines. **l** Photographs of the lungs collected from the mice in the PBS ("C") and "EcN-cypate + HBO + anti-PD-1 + laser" ("T") groups after fixation in Bouin's solution. The tumor nodules were indicated by red arrows. **m** Quantification results of the tumor nodules in **l**. Statistical data are presented as mean ± SD and calculated via two-tailed Student's *t*-test. *n* = 5 mice. **n** Representative H&E-stained lung slices after different treatments. The numbers/abbreviations in **h**−**n** represent different groups as follows: "1 and C": PBS, "2": PBS + anti-PD-1, "3": EcN-cypate + HBO, "4": EcN-cypate + HBO + anti-PD-1, "5": EcN-cypate + HBO + laser, and "6 and T": EcN-cypate + HBO + anti-PD-1 + laser. Statistical significance in **i** and **j** was calculated via one-way ANOVA with a Tukey's post-hoc test. *$P$ < 0.05, ***$P$ < 0.001, ****$P$ < 0.0001. Source data are provided as a Source Data file.

tumor cells to induce enhanced immune responses, yielding improved therapeutic outcomes.

To elucidate the underlying mechanisms responsible for the immunostimulatory effects of this bacteria-based HBO-coupled PTT strategy, we conducted transcriptomic analyses on the 4T1 tumor tissues. Specifically, we set the tumor tissues from the PBS-treated 4T1 tumor-bearing mice as the control group, and those from the mice in the "EcN-cypate + HBO + laser" group as the treated group. The volcano plot and heat map results revealed that a total of 276 (120 upregulated and 156 downregulated) DEGs were detected (Fig. 9a, c),

suggesting the significant changes in the tumors after HBO-facilitated PTT treatment. Subsequently, we performed GO and KEGG enrichment analyses on the tumor tissues to identify the evident changes after the HBO-coupled PTT treatment. As demonstrated in the GO enrichment analysis results (Fig. 9b and Supplementary Fig. 17), the BP of tumor cells was significantly influenced after the treatment. Not surprisingly, the component and organization of ECM, as shown in the cellular component (CC) part (Fig. 9b), were also influenced due to the HBO operation. Importantly, we found that most of the DEGs were enriched in immune response-associated signaling pathways, including immune response, response to cytokine, myeloid dendritic cell activation, dendritic cell differentiation, activated T cell proliferation, regulation of immune response, and cytokine activity, indicating the immunostimulatory effect of the HBO-enhanced PTT treatment. Furthermore, the KEGG enrichment analysis results confirmed the activation of the immune response-associated signaling pathways (e.g., natural killer cell-mediated cytotoxicity, antigen processing and presentation, T cell receptor signaling pathway, Th1 and Th2 cell differentiation, and cytokine–cytokine receptor interaction) after treatment (Fig. 9d and Supplementary Fig. 18). Additionally, the PPI network of DEGs in the 4T1 tumors from the control and "EcN-cypate + HBO + laser" groups revealed the relationships between these genes (Supplementary Fig. 19). Collectively, the transcriptomic analysis demonstrated the significant changes of the immune cells, specifically the T cells, in the tumor tissues after treatment. The above results demonstrated that the bacteria-based HBO-coupled PTT strategy can reprogram the immunosuppressive TME and induce strong systemic immune responses.

### HBO-enhanced photothermal immunotherapy can combine PD-1 blockade therapy to prevent lung metastasis

Immunosuppressive proteins, such as programmed death ligand 1 (PD-L1), are expressed on various malignant cancer cells such as 4T1 cells[48–50], and the PD-L1 can enable cancer cells to evade T cell surveillance by interacting with the PD-1 on T cells[51–54]. Notably, the KEGG enrichment analysis results revealed the activation of "PD-L1 expression and PD-1 checkpoint pathway in cancer" by the HBO-coupled PTT treatment (Fig. 9d), which can possibly explain the incomplete tumor elimination effect of this treatment (Fig. 7b). Since monoclonal antibodies blocking the interactions between PD-L1 and PD-1 are crucial for immunotherapy, PD-1 blockade therapy was therefore adopted to be combined with EcN-cypate-based cancer treatment. To investigate the impact of PD-1 blockade therapy on lung metastasis prevention, we employed 4T1 tumor-bearing BALB/c mice as a model system. As depicted in Fig. 10a, at 12 and 36 h post injection of EcN-cypate, HBO (1.5 ATA, 2 h) was applied to the mice. At 24 h post injection, anti-PD-1 (10 mg/kg) was administered into the mice via intraperitoneal injection. While at 24 and 48 h post injection, the mice were irradiated by an 808 nm laser (1 W/cm², 15 min). As shown in Fig. 10b, c, anti-PD-1 treatment alone had a negligible effect on tumor growth compared with the PBS group, possibly due to the immunosuppressive TME. In contrast, the "EcN-cypate + HBO + anti-PD-1 + laser" treatment exhibited the best therapeutic effect among all the groups and achieved a desirable tumor eradication efficiency (Fig. 10b–j). In particular, compared with "EcN-cypate + HBO + laser", the "EcN-cypate + HBO + anti-PD-1 + laser" treatment displayed a better antitumor therapeutic effect, demonstrating the crucial role of anti-PD-1 in realizing improved anticancer performance. Furthermore, the "EcN-cypate + HBO + anti-PD-1 + laser" group exhibited the largest damaged tumor region in the H&E-stained images and the most green fluorescence signals in the TUNEL assay kit-stained images (Fig. 10k), indicating the highest necrosis/apoptosis level of tumor cells in this group. As shown in the Supplementary Fig. 20, the body weight change curves validated the safety of our strategy during the observation period. Besides, the H&E-stained images revealed the negligible changes of major organs

after different treatments (Supplementary Fig. 21), demonstrating negligible tissue toxicity of PTT and PD-1 blockade therapy.

To evaluate the efficacy of the "EcN-cypate + HBO + anti-PD-1 + laser" treatment in preventing tumor metastasis, we i.v. injected 4T1 cells into the mice at day 21 to induce lung metastasis (Fig. 10a), and harvested the lung tissues at day 30 for analysis. As indicated by the red arrows in Fig. 10l, the "EcN-cypate + HBO + anti-PD-1 + laser" group exhibited fewer lung metastatic nodules compared with the PBS group, as further confirmed by the corresponding quantitative analysis (Fig. 10m). The H&E staining results also revealed that compared with the PBS group that displayed several metastatic nodules (marked by red circles), the "EcN-cypate + HBO + anti-PD-1 + laser" treatment significantly prevented the formation of lung metastatic nodules (Fig. 10n), indicating that the long-term immune memory effect induced by such a combined treatment can effectively prevent lung metastasis. Afterwards, tumor rechallenge models were constructed to further assess the long-term immune effect of our strategy (Supplementary Fig. 22a). Both the "EcN-cypate + HBO + anti-PD-1 + laser" and "EcN-cypate + HBO + laser" groups displayed a high efficiency in eradicating distant tumors compared with the PBS group (Supplementary Fig. 22b–d), and the "EcN-cypate + HBO + anti-PD-1 + laser" group realized the strongest therapeutic outcome (Supplementary Fig. 22e–g). Additionally, the immunofluorescence staining results suggested the increased CD3⁺CD4⁺ and CD3⁺CD8⁺ T cells in both the "EcN-cypate + HBO + anti-PD-1 + laser" and "EcN-cypate + HBO + laser" groups (Supplementary Fig. 22h), indicating the enhanced antitumor capacity of the above two treatments. The "EcN-cypate + HBO + anti-PD-1 + laser" group exhibited superior long-term immune effect compared with the "EcN-cypate + HBO + laser" group (Supplementary Fig. 22), probably due to the beneficial immunotherapeutic effect from the PD-1 blockade therapy. Collectively, we combined the HBO-enhanced photothermal immunotherapy with the PD-1 blockade therapy to augment the immune responses and achieve efficacious tumor eradication and metastasis prevention.

## Discussion

In this study, we developed a facile and effective strategy to enhance the accumulation of facultative anaerobes (EcN) within solid tumors via HBO treatment. Although HBO has been found to boost the delivery efficiency of nanoparticles within tumors, there exists uncertainty regarding whether HBO can similarly improve the penetration and accumulation of micrometer-sized living platforms, such as bacteria. We proved that HBO can also facilitate the intratumoral penetration and accumulation of bacteria, expanding the application range of HBO. During HBO treatment, we found that the dense ECM was depleted, and the bacteria could more easily penetrate the loosened tumor tissues.

To explore the potential of this strategy for tumor therapy, we constructed the engineered bacteria (EcN-cypate) by conjugating photothermal molecules (cypate) onto the surface of EcN to enable highly efficacious PTT. The covalent cypate conjugation on bacterial cells has the following merits: (1) Facile one-step conjugation: the covalent conjugation of cypate onto the EcN cell surface is very simple, and only requires a one-step mixture. (2) Excellent biocompatibility: such a covalent one-step cypate conjugation does not affect the viability of the EcN cells. (3) High conjugation stability and robust immunostimulation effect: the chemical conjugation approach employed in our work is more stable compared with the commonly adopted bacterial modification strategies based on physical coating. Furthermore, the close and tight conjugation between cypate and the EcN cell wall can potentiate the cell killing effectiveness of the heat generated during PTT, leading to the release of abundant bacterial antigens to stimulate strong immune responses for achieving enhanced immunotherapy. (4) Simultaneous fluorescence imaging capacity: cypate can not only achieve NIR light-mediated PTT effect,

but also realize excellent fluorescence imaging of engineered bacteria (EcN-cypate) to visualize the real-time intratumoral distribution of bacteria in vivo. (5) Repeated PTT ability of cypate: more importantly, EcN-cypate can realize repeated PTT to efficiently eliminate bacteria and tumor cells for achieving improved therapeutic and immunostimulation outcomes.

By taking advantage of HBO, EcN-cypate achieved more intratumoral accumulation to realize a better PTT-induced ICD effect, and the depletion of ECM facilitated the intratumoral infiltration of immune cells (e.g., matured DCs and CD4+/CD8+ T cells) to realize potentiated immunotherapy. To further stimulate the immune responses, we combined the above HBO-coupled PTT strategy with PD-1 blockade therapy to potentiate the tumor eradication effect and prevent lung metastasis. Collectively, this work provides a robust solution to boost the delivery efficiency of both natural bacteria (EcN) and engineered bacteria (EcN-cypate) via a facile and noninvasive HBO treatment, which may be beneficial for achieving enhanced bacteria-based cancer therapy. This work highlights the critical role of HBO in facilitating the intratumoral delivery of bacteria-based drugs and may foster the future development of new bacteria-based tumor therapies.

## Methods

### Ethics approval

All the animal experiments were carried out in accordance with the permission from the ethics committee of Southeast University (Nanjing, China) with an approval number of 20221010001. All the animal experiments were conducted in compliance with the Regulations for the Administration of Affairs Concerning Experimental Animals of China. All the animal experiments complied with institutional guidelines.

### Materials

Deionized water (18.2 MΩ•cm) was produced from a Milli-Q system (Millipore, Billerica, MA). Collagenase IV was purchased from Bio-Froxx (Guangzhou, China). Deoxyribonuclease I was obtained from Leagene Biotechnology Co., Ltd. (Beijing, China). MTT and agarose were bought from Shanghai Yuanye Bio-Technology Co., Ltd. N,N-Dimethylformamide (DMF), N-hydroxysuccinimide (NHS), bovine serum albumin (BSA), and 1-ethyl-3-(3-dimethylaminopropyl)carbodiimide hydrochloride (EDC•HCl) were obtained from Aladdin Chemistry Co., Ltd. (Shanghai, China). Dimethyl sulfoxide (DMSO) was ordered from Sinopharm Chemical Reagent Co., Ltd. IL-4 and granulocyte-macrophage colony-stimulating factor (GM-CSF) were bought from Pepro Tech Inc. (Rocky Hill, USA). Hoechst 33342 and enhanced ATP assay kit were purchased from Beyotime Institute Biotechnology (Shanghai, China). Rabbit anti-HIF-1α polyclonal antibody (cat. no. bs-20399R) and fluorescein isothiocyanate (FITC)-labeled goat anti-mouse immunoglobulin G (IgG) antibody (cat. no. bs-0296G-FITC) were obtained from Bioss Antibodies (Beijing, China). CRT rabbit polyclonal antibody (cat. no. 27298-1-AP), fibronectin rabbit polyclonal antibody (cat. no. 15613-1-AP), and HMGB1 rabbit polyclonal antibody (cat. no. 10829-1-AP) were purchased from Proteintech (Wuhan, China). Anti-mouse CD4 (cat. no. GB15064), anti-mouse CD3 (cat. no. GB12014), anti-mouse CD8 (cat. no. GB15068), anti-mouse granzyme B (cat. no. GB12093), anti-mouse Ly6G (cat. no. GB11229), anti-mouse FoxP3 (cat. no. GB11093), cyanine3 (Cy3)-labeled goat anti-rabbit IgG antibody (cat. no. GB21303), and anti-mouse CD206 (cat. no. GB113497) were obtained from Wuhan Servicebio Technology Co., Ltd. (China). Anti-mouse CD3-PE (cat. no. 12-0032-82), anti-mouse CD4-FITC (cat. no. 11-0041-82), anti-mouse CD8-PE-Cy7 (cat. no. 25-0081-81), anti-mouse CD11c-FITC (cat. no. 11-0114-82), anti-mouse CD80-PE (cat. no. 12-0801-82), and anti-mouse CD86-PE-Cy7 (cat. no. 25-0862-82) were obtained from Invitrogen (Carlsbad, USA). Anti-mouse F4/80-FITC (cat. no. 124611) and anti-mouse CD11c-PE (cat. no. 117308) were

purchased from BioLegend (San Diego, USA). PD-1 monoclonal antibody (cat. no. BP1046) was obtained from BioXcell (New Hampshire, USA). The HMGB1 ELISA kit was bought from Cusabio Biotech Co., Ltd. (Wuhan, China). Mouse cytokine ELISA kits for IL-1β, IL-6, and TNF-α were ordered from Invitrogen (Carlsbad, USA), and IL-10 and IFN-γ were purchased from MiltiSciences (Lianke) Biotech, Co., Ltd. (Hangzhou, China). EcN (cat. no. T23) was purchased from BioSci Co., Ltd. (China). The TUNEL assay kit and live/dead (calcein-AM/PI) assay kit were purchased from KeyGEN BioTECH Co., Ltd. (Nanjing, China). Lysogeny broth (LB) was purchased from Beijing Land Bridge Technology Co., Ltd. (China). pRSETB-mCherry plasmid was purchased from Hunan Fenghui Biotechnology Co., Ltd.

### Synthesis of EcN-cypate

Cypate was synthesized according to a previous report[32], and purified with Agilent 1290 Infinity high-performance liquid chromatography (HPLC) system. HPLC analyses were conducted at the Agilent 1260 Infinity II HPLC system using a C18 RP column with acetonitrile (0.1% of trifluoroacetic acid) and water (0.1% of trifluoroacetic acid) as the eluent. The successful synthesis was confirmed by electrospray ionization-mass spectrometry (ESI-MS, InfinityLab LC/MSD (Agilent, USA)) as follows: MS: calculated for cypate [M+]: 625.8; observed: ESI-MS [M+]: $m/z$ 625.3 ($n = 3$) (Supplementary Fig. 4), and the raw data were provided in the Source Data file. Cypate was decorated on the surface of EcN through the amidation reaction between the −COOH of cypate and the −NH2 on the surface of EcN. To investigate the optimal concentration of cypate for conjugation, we dissolved 500, 750, and 1000 μg cypate in 100 μL DMF, respectively. Then, the above cypate solutions were separately mixed with different amounts of EDC•HCl and NHS, and the detailed treatments were shown as follows: (1) 0.29 mg EDC•HCl and 0.18 mg NHS were added into 500 μg cypate-containing DMF solution, (2) 0.44 mg EDC•HCl and 0.26 mg NHS were added into 750 μg cypate-containing DMF solution, and (3) 0.58 mg EDC•HCl and 0.35 mg NHS were added into the 1000 μg cypate-containing DMF solution, and the mixtures were further stirred overnight at 0 °C to yield cypate NHS ester-containing solutions. After that, the above cypate NHS ester-containing solutions were separately reacted with $5 \times 10^7$ CFU EcN (dispersed in 4 mL 0.9% NaCl solution), and 400 μL (200 mM, pH 7.4) PBS was added to adjust the reaction pH. After 4 h stirring, the bacteria modified with cypate (termed EcN-cypate) were obtained by centrifugation (3200 × $g$, 5 min) and washed by PBS for three times to remove unreacted cypate. Subsequently, the EcN-cypate samples were observed by a confocal microscope (TCS SP8, Leica, Germany) to determine the optimal concentration of cypate for conjugation.

### Characterization

The morphology of EcN/EcN-cypate was observed by a scanning electron microscope (Zeiss Ultra Plus, Carl Zeiss, Germany), and the hydrodynamic sizes and zeta potentials of EcN and EcN-cypate in PBS (10 mM, pH 7.4) were measured by a zetasizer (Nano ZS, Malvern Instruments, UK). Besides, the absorbance of cypate or EcN-cypate (dispersed in DMSO) was measured by UV−vis spectroscopy using a Duetta fluorescence and absorbance spectrometer (Horiba Scientific, USA). ESI-MS analysis was conducted on an InfinityLab LC/MSD (Agilent, USA).

### Cell culture, bacterial culture, and animal model establishment

The 4T1 cell line (cat. no. KGG2224-1) was obtained from KeyGEN BioTECH, China. 4T1 cells were cultured in Roswell Park Memorial Institute (RPMI) 1640 (Gibco, USA) supplemented with 10% heat-inactivated fetal bovine serum (FBS), 0.08 mg/mL streptomycin, and 80 U/mL penicillin in an incubator (37 °C, 5% CO2). BMDCs, which

were separated from BALB/c mice, were incubated in the X-VIVO 15 culture medium (Lonza, Switzerland) with GM-CSF (20 ng/mL) and IL-4 (10 ng/mL) for 5 d to yield the immature BMDCs. All cell lines were tested for mycoplasma contamination. No mycoplasma contamination was found.

According to a previous study[55], we transformed the mCherry plasmid into EcN to construct the red fluorescent protein-expressing EcN (EcN-mCherry). EcN and EcN-mCherry were cultured in LB medium (160 rpm, 37 °C) and harvested at the exponential growth phase for further experiments. The number of bacteria (dispersed in PBS) was measured by a flow cytometry (NovoCyte 2070R, ACEA Biosciences Inc., USA). To obtain the bacterial colonies, the bacterial suspensions were diluted by PBS and plated on LB agar plates for another 16 h incubation. Subsequently, the number of bacterial colonies was counted.

Female BALB/c mice (6–8 weeks) were purchased from Yangzhou University Medical Center (Yangzhou, China). Mice ($n = 5$/group) were housed in ventilated cage (humidity: 40–70%) with 12 h dark–light cycles at constant room temperature. All mice had access to food and water ad libitum. The subcutaneous tumor model was constructed by inoculating $8 \times 10^6$ 4T1 cells into the BALB/c mice. When the tumor volume reached an average volume of $60 \pm 20$ mm$^3$, the mice were divided into different groups and treated for different purposes. For the maximal tumor size/burden, we obeyed the principle of the maximal tumor diameter of 20 mm according to the national standard (i.e., the Assessment Guidelines for Humane Endpoints in laboratory animal (RB/T 173-2018)), and the maximal tumor size/burden in this study was not exceeded.

### Evaluation of ECM depletion
To investigate the influence of HBO on TME, 4T1 tumor-bearing BALB/c mice ($n = 3$) were treated with HBO at 1.5 ATA for 2 h (set as the "HBO+" group). The 4T1 tumor-bearing BALB/c mice without HBO treatment were set as the "HBO−" group. Afterwards, the mice in the "HBO+" and "HBO−" groups were sacrificed to collect tumors for further studies.

For transcriptomic analysis, the RNA samples of the tumors were collected following the TRIzol-based procedure, and the total RNA samples were analyzed by Applied Protein Technology (Shanghai, China).

For visualizing the major components (collagen and fibronectin) of ECM, the tumors were collected for immunofluorescence staining of fibronectin and Masson's trichrome staining following the standard protocols.

### Multicellular spheroid model assay
The 3D tumor spheroids of 4T1 cells were constructed via a liquid overlay method. A 96-well plate was precoated with 100 μL of the FBS-free RPMI 1640 medium containing sterile agarose (2%, w/v). Afterwards, the 4T1 cells were seeded into each well at a density of 5000 cells/well and cultured in the RPMI 1640 medium supplemented with 10% FBS, and the tumor cells were cultured for 6 d to grow into spheroids with a diameter about 500 μm. Subsequently, the tumor spheroids were incubated with $1 \times 10^7$ CFU EcN-mCherry for 12 h followed by HBO (1.5 ATA, 2 h) treatment. Then, the 3D tumor spheroids were excited by a 552 nm laser of the confocal microscope to visualize the distribution of EcN-mCherry, and the Z-stack scanning was performed on the 3D tumor spheroids at the height of 40 μm by the microscope. Moreover, the integrated intensities of EcN-mCherry inside the 3D tumor spheroids were measured by ImageJ.

### Evaluation of intratumoral distribution of EcN
To visualize the distribution of EcN, EcN-mCherry was chosen as the model bacterium. Specifically, the 4T1 tumor-bearing mice were divided into 2 groups ($n = 3$/group), whose detailed treatments were shown as follows: (1) The mice were i.v. injected with $1 \times 10^7$ CFU EcN-mCherry

(set as the "HBO−" group) and (2) the mice were i.v. injected with $1 \times 10^7$ CFU EcN-mCherry and treated with HBO (1.5 ATA, 2 h) at 12 h post injection (set as the "HBO+" group). At 24 h post injection, the tumors were collected and stained by DAPI for observing the intratumoral distribution of EcN-mCherry using the confocal microscope.

### Agar plate count assay
To quantify the bacterial amounts in tumors after HBO treatment, we i.v. injected $1 \times 10^7$ CFU EcN to the 4T1 tumor-bearing BALB/c mice and treated the mice with HBO (1.5 ATA, 2 h) for 3 times at 12/36/60 h post injection. At 24/48/72 h post injection, the tumor tissues were separated from the mice after sacrifice and wet-weighed and homogenized in 1 mL of sterilized PBS. Besides, the major organs (heart, liver, spleen, lung, and kidneys) were also collected for evaluating the in vivo distribution of EcN at 24 h post injection. Afterwards, these samples were diluted to different concentrations and plated on LB agar plates. Finally, the plates were incubated at 37 °C overnight, and the number of colonies on each plate was recorded.

### In vitro biocompatibility evaluation of EcN-cypate
To study the viability of EcN-cypate, we measured the growth curves of the bacteria. To record the bacterial growth curves, EcN and EcN-cypate were diluted to an initial optical density at 600 nm (OD$_{600}$) of -0.05 in LB medium and incubated in a shaking incubator (160 rpm) at 37 °C, and then the OD$_{600}$ values were recorded by a microplate reader (Thermo Scientific, Multiskan FC, USA) at different time intervals.

To evaluate the dark toxicity of EcN-cypate toward 4T1 cells, the cells were separately seeded into 96-well plates ($5 \times 10^3$ cells/well) and incubated for 24 h. Afterwards, the cells were incubated with different concentrations of EcN-cypate (cypate: 0, 1.25, 2.5, 5, 10, and 20 μg/mL) for 4 h. Subsequently, the treated cells were washed by PBS for 3 times and subjected to MTT assay: 100 μL of RPMI 1640 culture medium and 10 μL of MTT solution (5 mg/mL) were added to each well and incubated with the cells for another 4 h. Afterwards, the culture medium of each well was replaced with 150 μL DMSO, and the absorbance of the solution in each well at 492 nm was measured by the microplate reader.

### Hemolysis assay
To evaluate the in vivo biocompatibility of EcN-cypate, we collected the blood from healthy BALB/c mice for hemolysis assay. Specifically, clean red blood cells (RBCs) were obtained through removing serum from the blood by centrifugation ($900 \times g$, 5 min) and dispersed in PBS. Then, RBC suspensions (0.1 mL) were separately mixed with EcN-cypate suspensions (0.1 mL) to obtain the final cypate concentrations of 0, 5, 10, 25, and 50 μg/mL, followed by incubation at 37 °C for 2 h. RBCs dispersed separately in PBS and water were set as negative control and positive control, respectively. Finally, the absorbance of the released hemoglobin at 450 nm in each group was measured using the microplate reader to calculate the hemolysis rates.

### Evaluation of the in vitro photothermal effects of EcN-cypate
1.5 mL of EcN-cypate (cypate: 20 μg/mL) suspension was irradiated by an NIR laser (808 nm, 1 W/cm$^2$) for different time periods, and the thermal images were recorded using an FLIR T540 thermal imaging camera (FLIR Systems Inc., USA). To test the photothermal stability of EcN-cypate, we recorded the temperature changes of an EcN-cypate suspension (cypate: 20 μg/mL) for 2.5 successive cycles of heating/cooling processes. To visualize the PTT-induced damage towards EcN, the EcN-cypate suspension (cypate: 20 μg/mL) was irradiated by an NIR laser (808 nm, 1 W/cm$^2$, 10 min). Then, the EcN-cypate cells were collected by centrifugation ($3200 \times g$, 5 min) and observed by SEM.

To evaluate the PTT effect of EcN-cypate, 4T1 cells were separately seeded into 96-well plates at a density of $5 \times 10^3$ cells/well and

incubated for 24 h. Afterwards, the cells were incubated with different concentrations of EcN-cypate (cypate: 0, 1.25, 2.5, 5, 10, and 20 µg/mL) for 4 h. Subsequently, the treated cells were irradiated by an 808 nm laser (1 W/cm$^2$) for 5 min and incubated for another 4 h. After washing the cells by PBS for 3 times, we conducted the MTT assay to determine the relative cell viabilities.

Live/dead cell staining assay was utilized to investigate the EcN-cypate-mediated PTT effect. Specifically, the 4T1 cells in 96-well plates (5 × 10$^3$ cells/well) were treated with EcN-cypate (cypate: 0, 5, and 10 µg/mL) and incubated for 4 h. Then, the cells were treated with laser irradiation (808 nm, 1 W/cm$^2$) for 5 min and incubated for another 4 h. Afterwards, the cells were washed by PBS for 3 times and costained by 10 µM calcein-AM (Ex/Em: 490/515 nm) and 10 µM PI (Ex/Em: 535/617 nm) for 30 min in a 37 °C incubator. After being washed by PBS, the cells were imaged by the confocal microscope.

### Detection of ICD biomarkers

To investigate the generation of DAMPs induced by ICD, 4T1 cells were seeded into Lab-Tek 8-well chamber slides (1 × 10$^4$ cells/well) and cultured for 24 h. Then, the cells were divided into 3 groups, whose detailed treatments were shown as follows: (1) The cells were left untreated (set as the control group), (2) the cells were treated with EcN-cypate (cypate: 5 µg/mL) (set as the "Laser−" group) for 4 h, and (3) the cells were treated with EcN-cypate (cypate: 5 µg/mL) for 4 h and irradiated by an NIR laser (808 nm, 1 W/cm$^2$, 5 min) (set as the "Laser+" group). The cells in the control, "Laser−", and "Laser+" groups were further incubated (37 °C, 12 h) for the following experiments. To investigate the CRT expression on the cell membrane, immuno-fluorescence assay and flow cytometry were utilized to visualize and quantify the CRT expression. Specifically, the cells in the control, "Laser−", and "Laser+" groups were washed with PBS and fixed in 4% glutaraldehyde solution for 10 min. Afterwards, the fixed cells were blocked by 5% BSA (25 °C, 2 h), and subsequently incubated with the anti-CRT antibody (1/400) at 4 °C for 12 h. Next, the cells were washed by PBS, and then treated with the FITC-labeled goat anti-mouse IgG antibody (1/300) (37 °C, 2 h). After being stained by Hoechst 33342, the cells were observed under the confocal microscope. For flow cytometry analysis, the sample treatment procedure was similar to that of confocal microscopic imaging. The difference is that the cells treated with FITC-labeled goat anti-mouse IgG antibody were quantified by flow cytometry without Hoechst 33342 staining.

The sample treatment procedure of HMGB1 staining was similar to that of CRT staining except that the cells were blocked by 5% BSA (25 °C, 3 h), treated with the anti-HMGB1 antibody (1/400) overnight (4 °C), washed with PBS, and then incubated with the Cy3-labeled goat anti-rabbit IgG antibody (1/300) (37 °C, 2 h).

To detect the contents of the HMGB1 and ATP released by the 4T1 cells after different treatments, the suspensions of the cells from different groups (control, "Laser−", and "Laser+") were collected and analyzed by the HMGB1 ELISA kit and enhanced ATP assay kit following the manufacturers' protocols.

### In vitro assessment of immune responses

To evaluate the in vitro immunological effects of EcN-cypate, 4T1 cells were seeded into 24-well plates (2 × 10$^4$ cells/well) and incubated for 24 h. Then, the cells were divided into 3 groups, whose detailed treatments were shown as follows: (1) The cells were left untreated (set as the control group, (2) the cells were treated with EcN-cypate (cypate: 5 µg/mL) (set as the "Laser−" group), and (3) the cells were treated with EcN-cypate (cypate: 5 µg/mL) for 4 h and irradiated by an NIR laser (808 nm, 1 W/cm$^2$, 5 min) (set as the "Laser+" group). After another 12 h incubation, the suspensions of each group (control, "Laser−", and "Laser+") were collected. To observe the influence of the obtained suspensions on BMDCs, immature BMDCs (1 × 10$^5$ cells) were seeded into 24-well plates with the suspensions from the control,

"Laser−", or "Laser+" groups, and incubated for 24 h. Next, BMDCs were collected and stained with anti-mouse CD11c-FITC, anti-mouse CD80-PE, and anti-mouse CD86-PE-Cy7 and analyzed by flow cytometry.

### Evaluation of the in vivo distributions of EcN-cypate after HBO treatment

To study the in vivo distribution of EcN-cypate, 4T1 tumor-bearing BALB/c mice were divided into 3 groups (n = 3/group), whose detailed treatments were shown as follows: (1) The mice were i.v. injected with free cypate (cypate dose: 10 mg/kg) (set as the "free cypate" group), (2) the mice were i.v. injected with EcN-cypate (cypate dose: 10 mg/kg) (set as the "EcN-cypate" group), and (3) the mice were i.v. injected with EcN-cypate (cypate dose: 10 mg/kg) and treated with HBO (1.5 ATA, 2 h) at 12, 36, and 60 h (set as the "EcN-cypate + HBO" group). Then, the mice were anesthetized by inhalation of a mixture of isoflurane (5%) with oxygen under general anesthesia and imaged by a PerkinElmer animal imaging system (IVIS Spectrum) at different time points post injection (Ex: 675 nm, Em: 720 nm). For ex vivo fluorescence imaging, the treated mice were sacrificed at day 1, 2, 3, or 7 post injection, and their tumor tissues, hearts, livers, spleens, lungs, and kidneys were excised and imaged. After that, the PerkinElmer Image Analysis Software was used to quantify the fluorescence signals.

### Evaluation of the hypoxia levels in major organs and tumors

To study the hypoxia levels in tumor regions, 4T1 tumor-bearing BALB/c mice were divided into 2 groups (n = 3/group), whose detailed treatments were shown as follows: (1) The mice were i.v. injected with EcN-cypate (cypate dose: 10 mg/kg) (set as the "EcN-cypate" group) and (2) the mice were i.v. injected with EcN-cypate (cypate dose: 10 mg/kg) and treated with HBO (1.5 ATA, 2 h) at 12 h post injection (set as the "EcN-cypate + HBO" group). At 24 h post injection, the mice were sacrificed and their tumors were excised for the immunofluorescence staining of HIF-1α following the standard protocol.

To assess the relative hypoxia levels in organs and tumors after HBO treatment, the 4T1 tumor-bearing BALB/c mice that were left untreated or treated with HBO were sacrificed, and their major organs and tumors were excised for the immunofluorescence staining of HIF-1α following the standard protocol.

### In vivo photothermal performance evaluation

4T1 tumor-bearing BALB/c mice were divided into 4 groups (n = 3/group), whose detailed treatments were shown as follows: (1) The mice were i.v. injected with PBS (set as the PBS group), (2) the mice were i.v. injected with free cypate (cypate dose: 10 mg/kg) (set as the "free cypate" group), (3) the mice were i.v. injected with EcN-cypate (cypate dose: 10 mg/kg) (set as the EcN-cypate group), and (4) the mice were i.v. injected with EcN-cypate (cypate dose: 10 mg/kg) and treated with HBO (1.5 ATA, 2 h) at 12 and 36 h (set as the "EcN-cypate + HBO" group). At 48 h post injection, the mice were irradiated by an NIR laser (808 nm, 1 W/cm$^2$) for 15 min, and the temperature changes were recorded by the thermal imaging camera.

### In vivo tumor therapy

4T1 tumor-bearing BALB/c mice were divided into 5 groups (n = 5/group), whose detailed treatments were shown as follows: (1) The mice were i.v. injected with PBS (set as the "PBS" group), (2) the mice were i.v. injected with EcN-cypate (cypate dose: 10 mg/kg) (set as the "EcN-cypate" group), (3) the mice were i.v. injected with EcN-cypate (cypate dose: 10 mg/kg) and treated with HBO (1.5 ATA, 2 h) at 12 and 36 h (set as the "EcN-cypate + HBO" group), (4) the mice were i.v. injected with EcN-cypate (cypate dose: 10 mg/kg) and irradiated by NIR laser (808 nm, 1 W/cm$^2$, 15 min) at 24 and 48 h (set as the "EcN-cypate + laser" group), and (5) the mice were i.v. injected with EcN-cypate

(cypate dose: 10 mg/kg), treated with HBO (1.5 ATA, 2 h) at 12 and 36 h, and irradiated by NIR laser (808 nm, 1 W/cm$^2$, 15 min) at 24 and 48 h (set as the "EcN-cypate + HBO + laser" group). The tumor size and body weight of the mice were recorded for 21 d, and then the mice were sacrificed. The tumor tissues were collected for H&E and TUNEL staining. Tumor volume ($V$) was calculated as width$^2$ × length/2. The green fluorescence intensities in the TUNEL assay kit-stained images were measured by ImageJ.

### In vivo biosafety assessment
For in vivo biosafety assessment, healthy female BALB/c mice were divided into 2 groups ($n = 3$/group) and separately i.v. injected with EcN-cypate (cypate dose: 10 mg/kg) and PBS. At day 14 post injection, the mice were sacrificed and the major organs (heart, liver, spleen, lung, and kidneys) and blood were collected for further evaluation. The major organs were stained by H&E following the standard protocol. The blood cells were analyzed using an automatic hematology analyzer (HBVET-1, Sinnowa, China). Biochemical analyses were performed on an automated biochemical analyzer (Chemifastar V, Sinnowa, China).

### In vivo assessment of immune responses in tumor tissues, spleens, and TDLNs
4T1 tumor-bearing BALB/c mice were divided into 5 groups ($n = 5$/group), whose detailed treatments were shown as follows: (1) The mice were i.v. injected with PBS (set as the "PBS" group), (2) the mice were i.v. injected with EcN-cypate (cypate dose: 10 mg/kg) (set as the "EcN-cypate" group), (3) the mice were i.v. injected with EcN-cypate (cypate dose: 10 mg/kg) and treated with HBO (1.5 ATA, 2 h) at 12 and 36 h (set as the "EcN-cypate + HBO" group), (4) the mice were i.v. injected with EcN-cypate (cypate dose: 10 mg/kg) and irradiated by NIR laser (808 nm, 1 W/cm$^2$, 15 min) at 24 and 48 h (set as the "EcN-cypate + laser" group), and (5) the mice were i.v. injected with EcN-cypate (cypate dose: 10 mg/kg), treated with HBO (1.5 ATA, 2 h) at 12 and 36 h, and irradiated by NIR laser (808 nm, 1 W/cm$^2$, 15 min) at 24 and 48 h (set as the "EcN-cypate + HBO + laser" group). At 48 h post injection, the photographs of the mice in the "EcN-cypate + laser" and "EcN-cypate + HBO + laser" groups were captured to record the PTT effect. After 8 d post injection, the tumor tissues, spleens, and TDLNs of the 4T1 tumor-bearing BALB/c mice in different groups (PBS, EcN-cypate, "EcN-cypate + HBO", "EcN-cypate + laser", and "EcN-cypate + HBO + laser") were collected for flow cytometry analysis. Specifically, the tumor tissues were incubated in the dissociation buffer (1 mg/mL collagenase IV and 100 μg/mL deoxyribonuclease I) at 37 °C for 1 h. Afterwards, we collected the single-cell suspensions from the tumor tissues using a cell strainer (70 μm). The spleens and TDLNs were stored in PBS and passed through a cell strainer (70 μm) to obtain the single-cell suspensions. The obtained suspensions were washed by PBS and then stained with anti-CD11c-FITC, anti-CD86-PE-Cy7, and anti-CD80-PE following the standard protocol to detect the mature DCs (CD11c$^+$CD86$^+$CD80$^+$). The suspensions were stained with anti-F4/80-FITC and anti-CD11c-PE following the standard protocol to detect the M1-like TAMs (F4/80$^+$CD11c$^+$). The suspensions were stained with anti-CD4-FITC, anti-CD3-PE, and anti-CD8-PE-Cy7 following standard protocol to detect the cytotoxic T cells (CD3$^+$CD8$^+$) and helper T cells (CD3$^+$CD4$^+$). Notably, all the antibodies were utilized at a dilution of 1:200, and the antibodies-stained single-cell suspensions were analyzed by flow cytometry.

To further evaluate the immune responses, the tumors in the PBS (control) and "EcN-cypate + HBO + laser" groups were collected for transcriptomic analysis. The RNA samples of the tumors were collected following the TRIzol-based procedure, and the total RNAs were analyzed by Applied Protein Technology (Shanghai, China).

The intratumoral and serum levels of IL-6, IL-1β, IFN-γ, and IL-10 and the intratumoral level of TNF-α were measured by the ELISA kits following the standard protocols.

### Tumor immunofluorescence analysis
Briefly, the 4T1 tumor-bearing BALB/c mice were sacrificed after various treatments, and their tumors were collected for the immunofluorescence analyses of CD3, CD4, CD8, CRT, Ly6G, CD206, FoxP3, and granzyme B following the standard protocols.

### Evaluation of the viability of EcN post PTT
4T1 tumor-bearing BALB/c mice were divided into 3 groups ($n = 3$/group), whose detailed treatments were shown as follows: (1) The mice were i.v. injected with EcN-cypate (cypate dose: 10 mg/kg) (set as the "Laser–" group), (2) the mice were i.v. injected with EcN-cypate (cypate dose: 10 mg/kg) and irradiated by NIR laser (808 nm, 1 W/cm$^2$, 15 min) at 24 h post injection (set as the "Laser+" group), and (3) the mice were i.v. injected with EcN-cypate (cypate dose: 10 mg/kg) and irradiated by NIR laser (808 nm, 1 W/cm$^2$, 15 min) at 24 and 48 h post injection (set as the "Laser++" group). At 48 h post injection of EcN-cypate, the tumor tissues were separated from the mice after sacrifice and wet-weighed and homogenized in 1 mL of sterilized PBS. Afterwards, these samples were diluted to different concentrations and spread on LB agar plates. After that, the plates were incubated at 37 °C overnight, and the number of colonies on each plate was recorded.

### Evaluation of the therapeutic and immunostimulation outcomes of repeated PTT
4T1 tumor-bearing BALB/c mice were divided into 3 groups ($n = 4$/group), whose detailed treatments were shown as follows: (1) The mice were i.v. injected with PBS (set as the "PBS" group), (2) the mice were i.v. injected with EcN-cypate (cypate dose: 10 mg/kg) and irradiated by NIR laser (808 nm, 1 W/cm$^2$, 15 min) at 24 h post injection (set as the "Laser+" group), and (3) the mice were i.v. injected with EcN-cypate (cypate dose: 10 mg/kg) and irradiated by NIR laser (808 nm, 1 W/cm$^2$, 15 min) at 24 and 48 h post injection (set as the "Laser++" group). The tumor sizes were recorded for 30 d, and then the mice were sacrificed. The tumor tissues were weighed and collected for H&E and TUNEL staining. Tumor volume ($V$) was calculated as width$^2$ × length/2.

To evaluate the immunostimulation outcomes of repeated PTT, another 12 4T1 tumor-bearing BALB/c mice were divided into 3 groups ($n = 4$/group), and they received the same treatments as those for the PBS, "Laser+", and "Laser++" groups. At 21 d post injection of PBS or EcN-cypate, the mice in the PBS, "Laser+", and "Laser++" groups were i.v. injected with $1 \times 10^5$ 4T1 tumor cells. After another 7 d, the mice were sacrificed and the lungs in the PBS, "Laser+", and "Laser++" groups were fixed in the Bouin's solution (Nanjing SenBeiJia Biological Technology Co., Ltd.). The number of metastatic nodules on the surface of the lungs was determined by counting the yellow regions after the fixation of the Bouin's solution. Besides, the establishment of 4T1 tumors in the lung tissues was also examined by H&E staining. Furthermore, the tumor tissues were collected for the immunofluorescence analyses of CD3, CD4, and CD8 using corresponding antibodies.

### Evaluation of the therapeutic outcomes and antimetastasis effects of different treatments
4T1 tumor-bearing BALB/c mice were divided into 6 groups ($n = 5$/group), whose detailed treatments were shown as follows: (1) The mice were i.v. injected with PBS (set as the "PBS" group), (2) the mice were i.v. injected with PBS and intraperitoneally (i.p.) injected with anti-PD-1 (10 mg/kg) (set as the "PBS + anti-PD-1" group), (3) the mice were i.v. injected with EcN-cypate (cypate dose: 10 mg/kg) and treated with HBO (1.5 ATA, 2 h) at 12 and 36 h (set as the "EcN-cypate + HBO" group), (4) the mice were i.v. injected with EcN-cypate (cypate dose: 10 mg/kg), treated with HBO (1.5 ATA, 2 h) at 12 and 36 h, and i.p. injected with anti-PD-1 (10 mg/kg) at 24 h (set as the "EcN-cypate + HBO + anti-PD-1" group), (5) the mice were i.v. injected with EcN-cypate (cypate dose: 10 mg/kg), treated with HBO (1.5 ATA, 2 h) at 12 and 36 h, and irradiated

by NIR laser (808 nm, 1 W/cm$^2$, 15 min) at 24 and 48 h (set as the "EcN-cypate + HBO + laser" group), and (6) the mice were i.v. injected with EcN-cypate (cypate dose: 10 mg/kg), treated with HBO (1.5 ATA, 2 h) at 12 and 36 h, i.p. injected with anti-PD-1 (10 mg/kg) at 24 h, and irradiated by NIR laser (808 nm, 1 W/cm$^2$, 15 min) at 24 and 48 h (set as the "EcN-cypate + HBO + anti-PD-1 + laser" group). The tumor size and body weight were recorded for 30 d, and then the mice were sacrificed. The tumor tissues were collected and weighed, and the H&E/TUNEL staining was utilized to check the therapeutic effects of each group (PBS, "PBS + anti-PD-1", "EcN-cypate + HBO", "EcN-cypate + HBO + anti-PD-1", "EcN-cypate + HBO + laser", and "EcN-cypate + HBO + anti-PD-1 + laser").

To confirm the safety of PTT, the major organs from different groups (PBS, "EcN-cypate + HBO + laser", and "EcN-cypate + HBO + anti-PD-1 + laser") were collected after repeated laser irradiation. The major organs were stained by H&E following the standard protocol.

To construct the lung metastasis model, another 10 4T1 tumor-bearing BALB/c mice were divided into 2 groups ($n = 5$/group), and they received the same treatments as those for the PBS and "EcN-cypate + HBO + anti-PD-1 + laser" groups. At 21 d post injection of PBS or EcN-cypate, the mice in the PBS and "EcN-cypate + HBO + anti-PD-1 + laser" groups were i.v. injected with $1 \times 10^5$ 4T1 tumor cells. After another 7 d, the mice were sacrificed and the lungs in the PBS and "EcN-cypate + HBO + anti-PD-1 + laser" groups were fixed in the Bouin's solution. The number of metastatic nodules on the surface of the lungs was determined by counting the yellow regions after the fixation of the Bouin's solution. Besides, the establishment of 4T1 tumors in the lung tissues was also examined by H&E staining.

## Evaluation of the therapeutic and immunostimulation outcomes of different groups in tumor rechallenge models

To evaluate if bacteria-mediated PTT can elicit a long-term immune effect and inhibit the growth of distant tumors, the tumor rechallenge models were constructed as follows: The primary tumor was constructed by inoculating $1 \times 10^6$ 4T1 cells into the left flank of each BALB/c mouse. When the primary tumor volume reached an average volume of $60 \pm 20$ mm$^3$, the mice were divided into 3 groups ($n = 5$/group), whose detailed treatments were shown as follows: (1) The mice were i.v. injected with PBS (set as the "PBS" group), (2) the mice were i.v. injected with EcN-cypate (cypate dose: 10 mg/kg), treated with HBO (1.5 ATA, 2 h) at 12 and 36 h, and irradiated by NIR laser (808 nm, 1 W/cm$^2$, 15 min) at 24 and 48 h (set as the "EcN-cypate + HBO + laser" group), and (3) the mice were i.v. injected with EcN-cypate (cypate dose: 10 mg/kg), treated with HBO (1.5 ATA, 2 h) at 12 and 36 h, i.p. injected with anti-PD-1 (10 mg/kg) at 24 h, and irradiated by NIR laser (808 nm, 1 W/cm$^2$, 15 min) at 24 and 48 h (set as the "EcN-cypate + HBO + anti-PD-1 + laser" group). Afterwards, the distant tumor was constructed at day 19 after the establishment of the primary tumor by injecting $5 \times 10^5$ 4T1 cells into the right flank of each BALB/c mouse. After the establishment of distant tumors, the tumor sizes of distant tumors were further recorded for 20 d. Afterwards, the mice were sacrificed, and the tumor tissues were weighed and collected for the immunofluorescence analyses of CD3, CD4, and CD8.

## Statistical and reproducibility

Most of the numeric data are expressed as mean ± SD. The significance between two groups was analyzed by two-tailed Student's $t$-test. For multiple comparisons, one-way analysis of variance (ANOVA) with Tukey's post-hoc test was adopted. $P$ values of less than 0.05 were considered significant. *$P < 0.05$, **$P < 0.01$, ***$P < 0.001$, ****$P < 0.0001$. All statistical analyses were performed by GraphPad Prism 9 or Excel 2019. The flow cytometry data were processed using FlowJo (version 10) and NovoExpress (version 1.5).

## Reporting summary

Further information on research design is available in the Nature Portfolio Reporting Summary linked to this article.

## Data availability

The main data supporting the findings in this study are available within the paper and its Supplementary Information. The sequencing data of the transcriptomic analyses in this study are available from the Sequence Read Archive (SRA) Run Selector of the National Center Biotechnology Information (NCBI) database with the NCBI BioProject accession number PRJNA1105037. The sequencing data of the transcriptomic analyses were analyzed using the FastQ Screen Trimmomatic (version 0.23.2) and HISAT2 (version 2.2.1). GO (http://www.geneontology.org/) and KEGG analyses were explored to evaluate the biological function of the DEGs. PPI network of DEGs was analyzed via the STRING database (http://string-db.org/). All remaining data are available in the Article, Supplementary, and Source Data files. Source data are provided with this paper.

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

## Acknowledgements

This work was supported by the Fundamental Research Funds for the Central Universities (2242023K5007), the Natural Science Foundation of Jiangsu Province (BK20211510), the National Natural Science Foundation of China (82372127), the Guangxi Key Laboratory of Early Prevention and Treatment for Regional High-Frequency Tumor, and the Key Laboratory of Early Prevention and Treatment for Regional High-Frequency Tumor (Guangxi Medical University), Ministry of Education (GKE-KF202305).

## Author contributions

K.F.X. and F.G.W. conceived the research and designed the experiments. K.F.X., S.Y.W., Z.W., Y.G., Y.X.Z., C.L., B.H.S., X.Z., and X.L. performed the experiments. K.F.X., S.Y.W., Z.W., Y.G., and F.G.W. analyzed the experimental results. K.F.X. and F.G.W. wrote the manuscript. F.G.W. supervised the project. All authors read and approved the manuscript.

## Competing interests

The authors declare no competing interests.
