## [Peer Review File · Nature Communications]

REVIEWER COMMENTS

Reviewer #1 (Remarks to the Author):

This is a comprehensive study that combines three cancer treatment modalities: 1) Hyperbaric oxygen (HBO), 2) engineered bacteria (EcN), and 3) photothermal therapy (PTT). Overall, Xu et al show an breadth of work that systematically evaluates the effects of HBO, PTT and EcN on tumor treatment both in vitro and in vivo.

The authors provide evidence that HBO treatment can deplete tumor ECM and in turn increase EcN tumor colonization both in tumor spheroids and in a mouse tumor model. They then modify EcN with cypate (fluorophore which rapidly increases in temperature upon excitation) and show that EcN-cypate + PTT can induce tumor cell death, along with activating immune cells in vitro. Finally, Xu et al treat 4T1 tumor-bearing mice with HBO + EcN-cypate + PTT and show that this combination therapy 1) can localize to tumors (along with liver), 2) reduce tumor growth while maintaining biocompatibility, 3) stimulate systemic anti-tumor immune response, and 4) HBO + EcN-cypate + PTT + anti-PD-1 treatment provides increased tumor clearance and mitigate lung metastasis.

While the authors do show the efficacy of this combination treatment, some questions and concerns remain:

1) Recent studies (Moen et al, *Target Oncol*, 2012; Liu et al, *Nano Today*, 2021, Wang et al, *Nano Today*, 2022; Liu et al, *Adv Sci*, 2021; Li et al, *ACS Appl Mater Interfaces*, 2018) have used HBO in conjunction with radiation, chemotherapy, or targeted nanoparticles to achieve increased efficacy. Additionally, the use of tumor-colonizing microbes and PTT has also been explored recently in the literature (Liang et al, *Front Bioeng Biotech*, 2022). The novelty here is using HBO with bacteria and PTT. While the data here appears convincing, what is the efficacy of this methodology compared to previous efforts (e.g. bacteria vs. nanoparticles)?

2) Figure 6d shows increased temperatures in bodies of mice treated with EcN-cypate + HBO + PTT more broadly vs. other treatment groups. Specifically, EcN-cypate + HBO + PTT also appears to increase temperature in the mouse head region much more significantly vs. treatments without HBO. Does this increased temperature (not just isolated to tumor region) have any physiological effect? Why does temperature increase more broadly with HBO treatment?

3) 4T1 tumors in the mice models were subcutaneous, making them easily accessible to PTT exposure. NIR lasers can penetrate skin up to a few millimeters. For non-subcutaneous tumors, would EcN-cypate + HBO + PTT be as effective?

4) Figure 10l shows that EcN-cypate + HBO + PTT + anti-PD-1 therapy reduces metastatic nodules on mice lungs vs. PBS treatment. It would be helpful to also see the lungs from mice treated with just EcN-cypate + HBO + PTT (no anti-PD-1) to determine fully if long term immune effects are derived from EcN-cypate + HBO + PTT or anti-PD-1 therapy.

5) How much viable EcN-cypate cells remain in the tumor post-PTT? Data from Figure 4f shows that EcN-cypate can be excited multiple times. Would repeated PTT exposures enhance tumor clearance and/or anti-tumor immune response?

Reviewer #2 (Remarks to the Author):

The authors depleted extracellular matrix (ECM) with HBO to promote accumulation and penetration of engineered bacteria within tumor tissues and facilitate immune cell infiltration for enhanced photothermal immunotherapy. Although this manuscript is well-designed and well-written, I do not recommend to accept this manuscript for the following issues. Simply replacing nanoparticles with engineered bacteria is insufficient to justify the publication in Nature Communications.

Major concerns:

1.The key issue with this manuscript is lack of novelty. Conceptually, the findings of this work are trivial. First, HBO has been utilized to promote tumor accumulation and penetration of commercialized nanomedicines, such as Doxil and Abraxane, homemade nanomedicine, and bacteria synthesized melanin for synergistic photothermal immunotherapy. Therefore, the results illustrated in Figures 4, 5, 8 and 9 are nothing new. Second, HBO has been leveraged to deplete ECM. The mechanism by which HBO depletes ECM has been revealed that HBO overcomes hypoxia to suppress cancer associated fibroblasts. From this aspect, the results presented in Figure 2 of the current manuscript are nothing new. Third, HBO has been used to regulate tumor immune microenvironment and facilitate immune cell infiltration. Therefore, most of the results presented in this manuscript is trivial. Furthermore, HBO alone is sufficient to modulate tumor immunosuppressive microenvironment, what are the advantages of adding engineered bacteria and photothermal therapy.

2.The mechanistic studies in insufficient. For instance, the groups in Figure 9 are not enough. The authors need to set 5 groups, similar to the results in Figure 8. It is not clear which therapy plays a decisive role in the combination therapy, since there are so many treatments. Furthermore, there is no correlation between Figure 8 and Figure 9, in terms of molecules, proteins or immune cells. The authors need to establish a tight connection between Figure 8 and Figure 9. Similar problems exist in Figure 2 and Figure

3. The mechanism by which HBO enhances bacteria tumor accumulation and penetration is not clear. Is this mechanism similar to the case of HBO-boosted nanomedicine?

3. The results cannot support the authors' claims. For instance, no result demonstrates that the combination therapy induces a long-term immune effect. To corroborate this conclusion, the authors are suggested to perform a rechallenge tumor model.

Minor concerns:

1. There is no discussion in this manuscript. The authors are suggested to compare their results with published results. The contribution of the current study is not clear.

2. It is not clear whether the engineered bacteria proliferate within tumors or normal tissues?

3. The results presented in Figure 10 are inconsistent with the description. No complete tumor eradication has been achieved in the last group although so many therapies have been combined together.

Reviewer #3 (Remarks to the Author):

In this manuscript, the authors claimed that hyperbaric oxygen (HBO) can deplete the extracellular matrix (ECM) and thus enhance the accumulation and penetration of bacteria within the tumor. Based on that, the authors modify an anaerobe-Escherichia coli Nissle 1917 (EcN) with a photothermal fluorophore (cypate) (EcN-cypate) to achieve photothermal therapy (PTT), which also can induce the immunogenic cancer cell death (ICD) in vitro. Then the combination of PD-1 blockade therapy and EcN-cypate-based PTT after HBO treatment could realize long-term immunosurveillance to inhibit lung metastasis. The design of this work was lack of novelty and some of the data was not solid enough. I don't suggest the publication of this manuscript in Nature Communication. Some questions and suggestions are listed below:

1. many papers have proved that anaerobic bacteria can load photosensitizers after chemical modification to perform PDT or PTT on tumors and activate immune response therapy (Adv. Sci. 2021, 8, 2003572). It also has been reported that after hyperbaric oxygen (HBO) treatment of tumors, drug delivery efficiency will be improved (Nano Today 2021, 40, 101248), and the accumulation and penetration of immune cells in tumor tissues will be increased (Adv. Sci. 2021, 8, 2100233). Based on these publications, the authors verified that HBO treatment increases the accumulation of EcN-cypate in tumors and facilitates the intratumoral infiltration of immune cells to realize tumor therapy through PTT and ICD-induced immunotherapy. It seems to be a combination of previous reports without obvious innovation. What is the innovation in this paper?

2. Figure 3b shows that the tumor multicellular spheroids have differences in shape and size in the two groups of HBO- and HBO+. Figure 3c shows that the tumor cell densities are different in these two groups. The penetration and aggregation effect of anaerobic bacteria will of course be poor in HBO- group due to the higher tumor cell density of that. The authors need to re-perform the experiment to exclude the influence of the size and density of tumor multicellular spheroids on the experimental results.

3. EcN can target tumor tissues due to its anaerobic nature. However, after HBO treatment, although the influence of ECM is reduced, it also increases the oxygen content of tumor tissues. The author needs to supplement the HE staining data of HIF-1 α in major organs (heart, liver, spleen, lung, kidney) of mice after HBO treatment to confirm whether tumors can maintain a relatively oxygen-deficient state compared with these major organs after HBO.

4. On page 9, line 192, the author claimed that 'Besides, we demonstrated that the HBO treatment had a negligible influence on the distribution of EcN within the major organs of the treated mice (Supplementary Fig. 3c and d).' Figure S3d in Supplementary showed that there was no significant difference in EcN-cypate concentration of the heart, liver, spleen, and kidney after hyper-pressure oxygen treatment. However, EcN-cypate concentration in the lungs increased significantly (it seems to have more than doubled) after HBO treatment. How do the authors explain that?

5. In Figure 4g, why does the temperature at the top of the centrifuge tube drop after 1 minute of laser irradiation? And after 4 minutes, and 7 minutes of exposure, the temperature at the top of the centrifuge tube went up compared with that in 1 minute. It seems illogical. It is hoped that the author can re-examine the photothermal data of the material.

6. In Figure 4i, significant cytotoxicity (the cell survival rate is about 50%) at the concentration of 5 $\mu\text{g}/\text{mL}$ under laser irradiation. However, in Figure 4j, at the same condition, the cell survival rates are well above 50%. The two results are contradictory. How do the authors explain that?

7. In figure 5c and figure 5e, The two results are also clearly contradictory. Please check and re-perform the experiments carefully.

8. Figure 6b shows that the distribution of EcN-cypate in major organs (heart, liver, spleen, lung, and kidney) in mice is significantly increased regardless of 24, 48, or 72 hours after being treated with HBO. This seems to contradict the conclusion declared by the author on page 9, line 192. 'Besides, we demonstrated that the HBO treatment had a negligible influence on the distribution of EcN within the major organs of the treated mice (Supplementary Fig. 3c and d).' How does the author explain that?

9. Figure 6d shows the thermal images of tumor-bearing mice with different treatments after receiving NIR laser irradiation. However, the temperature of the mouse body in the EcN-cypate+HOB group is higher than other groups at any time point. Thus It's hard to come to that conclusion the authors claim.

10. It is hoped that the author could provide the weight change curve and HE staining data of important organs of mice such as the heart, liver, spleen, lung, and kidney in different groups during photothermal therapy combined with checkpoint immunotherapy, to demonstrate the toxicity of EcN-cypate+HOB based photothermal - checkpoint immunotherapy combined therapy.

11. In the part on experimental methods, the author did not mention some key data such as the size of the tumor at the beginning of treatment. Please complete the experimental procedures.

12. The author mentioned that two blood biochemical experiments were carried out (In Page S10, line 284, 'In vivo biosafety assessment' and Page S12, line 334, 'Evaluation of the in vivo biocompatibility of EcN-cypate'), but only a set of blood biochemical data was found in Figure S10 in the paper, which is confusing. The author should check and proofread it.

Responses to Reviewer 1:

General comment: This is a comprehensive study that combines three cancer treatment modalities: 1) Hyperbaric oxygen (HBO), 2) engineered bacteria (EcN), and 3) photothermal therapy (PTT). Overall, Xu et al show an breadth of work that systematically evaluates the effects of HBO, PTT and EcN on tumor treatment both in vitro and in vivo. The authors provide evidence that HBO treatment can deplete tumor ECM and in turn increase EcN tumor colonization both in tumor spheroids and in a mouse tumor model. They then modify EcN with cypate (fluorophore which rapidly increases in temperature upon excitation) and show that EcN-cypate + PTT can induce tumor cell death, along with activating immune cells in vitro. Finally, Xu et al treat 4T1 tumor-bearing mice with HBO + EcN-cypate + PTT and show that this combination therapy 1) can localize to tumors (along with liver), 2) reduce tumor growth while maintaining biocompatibility, 3) stimulate systemic anti-tumor immune response, and 4) HBO + EcN-cypate + PTT + anti-PD-1 treatment provides increased tumor clearance and mitigate lung metastasis. While the authors do show the efficacy of this combination treatment, some questions and concerns remain:

Author reply: We deeply appreciate the respected reviewer for his/her very professional comments and suggestions, which help us to improve the quality of the manuscript significantly! We have tried our best to improve the manuscript according to your very important comments/suggestions.

Herein, we would also like to summarize the key findings of this work as shown below:

(1) **Uniqueness of Micrometer-Sized Platform:** While previous studies have explored the synergy of hyperbaric oxygen (HBO) with nanoparticles to enhance the tumor accumulation and penetration of the latter, aiming to achieve improved antitumor therapeutic outcomes, our work introduces a unique perspective. Notably, no existing reports have demonstrated the similar effects when employing micrometer-sized platforms such as engineered bacteria. Our approach reveals the potential of combining HBO with micrometer-sized bacteria, providing insights into enhancing the accumulation and penetration of engineered bacteria for realizing effective cancer therapy. In addition, as shown below, we found that the other two micrometer-sized cellular drugs modified with cypate (i.e., *the cypate-conjugated red blood cell (RBC-cypate)* and *the cypate-conjugated probiotic Bacillus coagulans (BC-cypate)*) **exhibit strikingly different HBO-dependent tumor accumulation behaviors**, as compared with the micrometer-sized EcN-cypate, indicating that **a suitable micrometer size is also important for realizing an enhanced tumor accumulation after HBO treatment. Nevertheless, this finding cannot be accurately predicted without detailed experimental demonstration.**

(2) **Live Cells with Hypoxia-Targeting Capacity:** Different from the “dead” nanoparticles, EcN-cypate, being a live cell, possesses the inherent hypoxia-targeting capability. This distinction prompted us to investigate the in vivo distribution of bacteria after HBO treatment, revealing that they may differ

significantly from “dead” nanoparticles. According to your very professional suggestions, we expanded our evaluation to include different platforms, encompassing “dead” nanoparticles (i.e., cypate-containing liposomes, serving as a typical nanosized drug) and the above-mentioned two “live” micrometer-sized cellular drugs (i.e., (1) cypate-conjugated red blood cells (RBC-cypate), aiming to investigate **the influence of size** on drug delivery efficiency under HBO treatment; (2) cypate-conjugated *Bacillus coagulans* (BC-cypate), aiming to study the impact of **hypoxia targeting ability** on engineered bacteria delivery efficiency under HBO treatment), to assess their in vivo distributions and therapeutic outcomes. The detailed experimental results and discussions are listed in the following comments. The corresponding results revealed that, in addition to the **size factor** as mentioned in the above paragraph, the **dead/live state** as well as the **hypoxia-targeting ability** can significantly affect the tumor accumulation of drugs after HBO treatment, which can be demonstrated by the corresponding experimental results on **the live cells with hypoxia-targeting ability** (cypate-containing liposomes vs. EcN-cypate), as well as **the live cells bearing different hypoxia-targeting abilities** (BC-cypate vs. EcN-cypate). Also, **such a conclusion cannot be drawn without clear experimental demonstration.**

(3) **Potentiated Bacteria-Mediated Cancer Therapy:** Recently, bacteria-mediated cancer therapies have attracted increasing interest due to their innate ability to boost immune responses (*Science* 2022, 378, 858–864.; *Science* 2023, 382, 211–218.; *Chem. Soc. Rev.* 2023, 52, 6617–6643.). **Improving the intratumoral accumulation and penetration of micrometer-sized engineered bacteria is crucial for bacteria-based cancer treatments.** Our innovative strategy addresses this concern by combining engineered bacteria with a **facile and non-invasive HBO treatment**, which may foster the future development of new and effective bacteria-based tumor therapies.

(4) **Specific Merits of Cypate Covalent Conjugation:** There are 5 merits for the covalent cypate conjugation on bacterial cells: (1) **Facile one-step conjugation:** the covalent conjugation of cypate onto the EcN cell surface is very simple, and only requires a one-step mixture. (2) **Excellent biocompatibility:** such a covalent one-step cypate conjugation does not affect the viability of the EcN cells, as evidenced by the results shown in Figure 4f on Page 13. (3) **High conjugation stability and robust immunostimulation effect:** the chemical conjugation approach employed in our work is more stable compared with the commonly adopted bacterial modification strategies based on physical coating. Furthermore, the close and tight conjugation between cypate and the EcN cell wall can potentiate the cell killing effectiveness of the heat generated during PTT, leading to the release of abundant bacterial antigens to stimulate strong immune responses for achieving enhanced immunotherapy. (4) **Simultaneous fluorescence imaging capacity:** cypate can not only achieve NIR light-mediated PTT effect, but also realize excellent fluorescence imaging of engineered bacteria (EcN-cypate) to visualize the real-time intratumoral distribution of bacteria in vivo. (5) **Repeated PTT ability of cypate:** more importantly, EcN-cypate can realize repeated PTT to efficiently eliminate bacteria and tumor cells for achieving improved therapeutic and immunostimulation outcomes (Fig. S16, Page

S26).

Collectively, the combination of HBO and engineered bacteria (EcN-cypate) represents a novel strategy, **which is not simply replacing nanoparticles with engineered bacteria, and this strategy may benefit the development of bacteria-mediated tumor therapy.**

Comment 1: Recent studies (Moen et al, Target Oncol, 2012; Liu et al, Nano Today, 2021, Wang et al, Nano Today, 2022; Liu et al, Adv Sci, 2021; Li et al, ACS Appl Mater Interfaces, 2018) have used HBO in conjunction with radiation, chemotherapy, or targeted nanoparticles to achieve increased efficacy. Additionally, the use of tumor-colonizing microbes and PTT has also been explored recently in the literature (Liang et al, Front Bioeng Biotech, 2022). The novelty here is using HBO with bacteria and PTT. While the data here appears convincing, what is the efficacy of this methodology compared to previous efforts (e.g. bacteria vs. nanoparticles)?

Author reply: We greatly appreciate the respected reviewer for raising this very professional comment! In the studies the respected reviewer mentioned (Moen et al, Target Oncol, 2012; Liu et al, Nano Today, 2021, Wang et al, Nano Today, 2022; Li et al), HBO was employed to enhance the intratumoral accumulation and penetration of **nanoparticles**, aiming to achieve improved antitumor therapeutic outcomes. However, existing literature lacks evidence demonstrating the existence of a similar effect when utilizing **“micrometer-sized” and “live” platforms**, such as engineered live bacteria. Although the bacteria-mediated PTT has been reported for tumor therapy (Liang et al, Front Bioeng Biotech, 2022), **the limited infiltration of immune cells constrains the efficacy of bacteria-mediated therapy** (*Int. J. Med. Microbiol.* 2007, 297, 151.; *Adv. Sci.* 2021, 8, 2003572). Most of the bacteria are colonized in the core necrosis area of the tumor with less distribution in the outermost area of the tumor. Therefore, the immune cells infiltrated in the tumor may have spatial distribution heterogeneity, resulting in an unsatisfactory antitumor effect. This outcome suggests that the limited infiltration of immune cells constrains the efficiency of bacteria-mediated therapy. **Nevertheless, our strategy addressed this limitation by combining bacteria-mediated therapy with HBO.** In detail, our innovative approach (EcN-cypate + HBO + laser) reveals the potential of combining HBO with micrometer-sized bacteria, achieving superior therapeutic and immunostimulation outcomes compared with single bacteria-mediated PTT (EcN-cypate + laser). Notably, the significant differences in the used materials and tumor models between previous studies and our investigation render it impossible to make an accurate comparison. To comprehensively compare our methodology with previous efforts (e.g., nanoparticles), we synthesized a typical liposomal nanoparticle as the reference. Initially, we compared the intratumoral delivery efficiency of EcN-cypate with cypate-containing liposomal nanoparticles after HBO treatment. Afterwards, we systematically evaluated the efficacy of the above two drugs in inducing immune responses and promoting tumor eradication when combined with HBO. The detailed results and discussions are presented in the following text.

We synthesized LNP-cypate, a cypate-containing liposomal nanoparticle, as the typical nanodrug. Besides, we also designed two micrometer-sized cellular drugs modified with cypate: the cypate-conjugated red blood cell (RBC-cypate) and the cypate-conjugated probiotic *Bacillus coagulans* (BC-cypate), for comparison with the micrometer-sized EcN-cypate. RBCs, which are larger than EcN with an average size of $\sim 7 \mu\text{m}$, have been widely employed in tumor therapy (*Nat. Commun.* 2021, 12, 2637.; *Adv. Funct. Mater.* 2016, 26, 1757–1768.). Herein, we synthesized RBC-cypate to investigate the influence of size on drug delivery efficiency under HBO treatment. *Bacillus coagulans* (BC), a type of probiotic with a weaker hypoxic tendency than EcN, is usually utilized in intestinal therapy (*Int. J. Mol. Sci.* 2018, 19, 2084.). Herein, we synthesized BC-cypate to explore the impact of hypoxia targeting ability on the delivery efficiency of engineered bacteria under HBO treatment.

According to the previous study (*J. Control. Release* 2023, 357, 222–234.), we synthesized the typical liposome-based cypate nanoparticles (LNP-cypate) using the film-hydration and probe-sonication method. Specifically, hydrogenated soy phosphatidylcholine (HSPC), 1,2-distearoyl-*sn*-glycero-3-phosphoethanolamine-*N*-[methoxy(polyethylene glycol)-2000] (ammonium salt) (DSPE-PEG₂₀₀₀-OMe), cholesterol, and cypate were dissolved in a $\text{CHCl}_3/\text{CH}_3\text{OH}$ mixed solution ($\text{CHCl}_3/\text{CH}_3\text{OH} = 65: 35$, vol/vol) at a mass ratio of 3: 1: 1: 0.5. Then, the mixture was blown dry by a nitrogen stream, dried under vacuum overnight, and hydrated with 2 mL PBS solution at 65°C . The hydrated mixture was subjected to probe ultrasonication at the 25% power of an ultrasonic cell crusher (X0-650D, Xianou Tech, China), for 300 s (consisting of 15 cycles, with a 3 s pause after 3 s working per cycle) to obtain LNP-cypate. The synthetic procedures for RBC-cypate and BC-cypate were similar to EcN-cypate. In detail, 0.58 mg EDC \cdot HCl and 0.35 mg NHS were added into the cypate (1 mg)-containing DMF solution, and the mixture was further stirred overnight at $\sim 4^\circ\text{C}$ to yield the cypate NHS ester-containing solution. Subsequently, the cypate NHS ester-containing solution was separately reacted with 5×10^7 colony forming units (CFU) BC cells or 1×10^7 RBCs in PBS. After 4 h stirring, the BC-cypate or RBC-cypate was obtained by centrifugation and washed by PBS for three times to remove unreacted cypate. The concentrations of cypate in RBC-cypate, BC-cypate, and LNP-cypate were determined using a Duetta fluorescence and absorbance spectrometer (Horiba Scientific, USA).

As shown in Fig. R1a, the TEM result revealed that LNPs-cypate were spherical nanoparticles. DLS result revealed that they had a hydrodynamic diameter of $68.5 \pm 5.7 \text{ nm}$ (Fig. R1b). The confocal microscopic images validated the successful conjugation of cypate onto the surface of RBCs (Fig. R1c) and BC (Fig. R1d).

Fig. R1 **a**, TEM image of LNP-cypate. **b**, Hydrodynamic diameter of LNP-cypate. Confocal fluorescence images of **(c)** RBC-cypate and **(d)** BC-cypate.

To investigate the impact of HBO on the intratumoral delivery efficiency of various platforms (EcN-cypate, LNP-cypate, RBC-cypate, and BC-cypate), we assessed cypate distributions in orthotopic 4T1 tumor-bearing mice (Fig. R2a). As shown in Fig. R2b, RBC-cypate exhibited the lowest intratumoral delivery efficiency and fastest metabolism, and **HBO treatment had negligible effect on enhancing the delivery efficiency of RBC-cypate**. The larger size of RBCs ($\sim 7 \mu\text{m}$) probably prevents RBC-cypate to efficiently penetrate the blood vessels and accumulate in tumor regions. Notably, **HBO treatment slightly increased the intratumoral accumulation of BC-cypate, potentially due to the depletion of dense ECM**, which may limit the penetration of micrometer-sized drugs. However, the limited hypoxia targeting capacity of BC hindered the delivery efficiency of BC-cypate. In contrast, EcN-cypate, which are also micrometer-sized cellular drugs with an average size of $3 \mu\text{m}$ (which was similar to that ($3 \mu\text{m}$) of BC-cypate), exhibited significant intratumoral enrichment, which was further improved by HBO treatment. **These results indicate that HBO cannot facilitate the intratumoral delivery efficiency of all micrometer-sized platforms, and the size and hypoxia targeting capacity of drugs play important roles in the delivery process.** Additionally, HBO treatment also promoted the intratumoral delivery efficiency of nanoparticles (LNP-cypate) (Fig. R2b). **However, LNP-cypate displayed lower intratumoral accumulation at 48 h than that at 24 h, possibly due to rapid nanoparticle metabolism. Conversely, EcN-cypate exhibited prolonged intratumoral retention time, lasting for at least 48 h (Fig. R2b), which was beneficial for achieving repeated PTT.** Moreover, the semiquantitative results demonstrated that EcN-cypate realized significantly higher intratumoral accumulation than LNP-cypate at 48 h (Fig. R2c), highlighting the superiority of combining HBO with engineered bacteria (EcN-cypate). Afterwards, by analyzing the fluorescence intensity of major organs, we observed the strongest

cypate signal in the liver, suggesting that the drug may undergo hepatic clearance (Fig. R2d). Collectively, the combination of HBO and engineered bacteria (EcN-cypate) is a novel strategy, which is not simply replacing nanoparticles with engineered bacteria but is rationally designed by considering the size/hypoxia targeting effects and intratumoral retention time of drugs.

Fig. R2 a, Experimental outline showing the procedures for evaluating the distributions of drugs (EcN-cypate, LNP-cypate, RBC-cypate, and BC-cypate) in orthotopic 4T1 tumor models. **b**, Ex vivo distributions of cypate at different time points (24, 48, and 72 h) post intravenous injection of EcN-cypate, LNP-cypate, RBC-cypate, or BC-cypate. “HBO+”: The mice were treated with HBO (1.5 ATA, 2 h) post injection of drugs. **c**, Semiquantitative distribution results of cypate in the tumors at 48 h post injection in different groups. **d**, Heat map showing the distribution of cypate in the major organs (liver, lung, and kidney). The doses of cypate in all groups (EcN-cypate, LNP-cypate, RBC-cypate, and BC-cypate) were 10 mg/kg.

Although the *in vivo* distribution results have already demonstrated the novelty of combining HBO with engineered bacteria (EcN-cypate), we further conducted a comparative analysis of the therapeutic outcomes between EcN-cypate and nanoparticle-based treatment using LNP-cypate. Firstly, we investigated the *in vitro* immune responses induced by PTT mediated by EcN-cypate and LNP-cypate. The photographs suggested that a substantial portion of EcN was eradicated after laser irradiation due to the photothermal effect (Fig. R3a), which was further confirmed by the quantitative results (Fig. R3b). According to previous studies (*Science* 2022, 378, 858–864.; *Chem. Soc. Rev.* 2023, 52, 6617–6643.), dead bacteria can induce immune responses to enhance immunotherapy. To study the potential immune responses stimulated by dead bacteria and LNP-cypate (or EcN-cypate)-mediated PTT, we separately incubated the immature dendritic cells (DCs) with different suspensions including PBS, “LNP-cypate + 4T1”, “LNP-cypate + 4T1 + laser”, “EcN-cypate + laser” (dead bacteria), and “EcN-cypate + 4T1 + laser” groups. As shown in Fig. R3c and d, the “LNP-cypate + 4T1 + laser” group presented a higher percentage of mature DCs than the PBS and “LNP-cypate + 4T1” groups, owing to the PTT-induced ICD. Notably, the “EcN-cypate + laser” group displayed a higher percentage of mature DCs than the PBS group, attributed to the dead bacteria-induced immune responses. Consequently, the “EcN-cypate + 4T1 + laser” group exhibited the highest percentage of mature DCs among all groups, probably due to a combination of PTT-induced ICD and dead bacteria-induced immune responses. **The above results demonstrated that engineered bacteria (EcN-cypate)-mediated PTT could induce stronger immune responses than nanoparticles (LNP-cypate)-mediated PTT.**

Fig. R3 a, Representative photographs showing EcN colonization behaviors before and after laser irradiation (808 nm, 1 W/cm², 5 min). The numbers (10⁴ and 10²) in the image indicated the dilution factor of the original bacterial suspension. **b**, Quantitative bacteria counts in the EcN-cypate and “EcN-cypate + laser” groups. Data are presented as mean ± standard deviation ($n = 3$) and analyzed by two-tailed Student’s *t*-test (**** $P < 0.0001$). **c**, Quantitative statistics of mature DCs after

different treatments and (d) corresponding representative flow cytometric analysis results of mature DCs (CD11c⁺CD80⁺CD86⁺). PBS: The immature DCs were incubated with PBS for 24 h. “LNP-cypate + 4T1”: The 4T1 cells were firstly incubated with LNP-cypate (cypate: 5 µg/mL) for 4 h, and then the suspensions were collected for incubating with immature DCs for 24 h. “LNP-cypate + 4T1 + laser”: The 4T1 cells were firstly incubated with LNP-cypate (cypate: 5 µg/mL) for 4 h, followed by laser irradiation (808 nm, 1 W/cm², 5 min), and then the suspensions were collected for incubating with immature DCs for 24 h. “EcN-cypate + laser”: EcN-cypate (cypate: 5 µg/mL) suspensions were collected after laser irradiation (808 nm, 1 W/cm², 5 min) and then incubated with immature DCs for 24 h. “EcN-cypate + 4T1 + laser”: The 4T1 cells were firstly incubated with EcN-cypate (cypate: 5 µg/mL) for 4 h, followed by laser irradiation (808 nm, 1 W/cm², 5 min), and then the suspensions were collected for incubating with immature DCs for 24 h. Data are presented as mean ± standard deviation ($n = 3$) and analyzed by one-way analysis of variance (ANOVA) (* $P < 0.05$, **** $P < 0.0001$). “ns” stands for nonsignificant difference.

To further assess the in vivo immune responses induced by EcN-cypate- and LNP-cypate-mediated PTT, we constructed orthotopic 4T1 tumor models (Fig. R4a). After laser irradiation, the tumors became darkened, signifying efficient therapeutic outcomes of PTT (Fig. R4b). As shown in Fig. R4c, we analyzed the cytotoxic T cells (CD3⁺CD8⁺) in the tumor-draining lymph nodes (TDLNs), and the “EcN-cypate + HBO + laser” group presented a higher proportion of CD3⁺CD8⁺ T cells (39.07%) than that of “LNP-cypate + HBO + laser” (31.65%), demonstrating that the HBO-combined engineered bacteria can induce more effective killing by the immune system. The quantitative analysis further supported this finding, showing that the “EcN-cypate + HBO + laser” group had the highest proportions of cytotoxic T cells, mature dendritic cells (DCs), and M1 macrophages among all groups (Fig. R4d–f), indicating the efficient immune responses induced by HBO-combined PTT. Furthermore, we evaluated the proportions of mature DCs and M1 macrophages in spleens and tumors, and the results demonstrated that the “EcN-cypate + HBO + laser” group induced desirable systemic immune responses (Fig. R4g–j). As shown in Fig. R4k, the immunofluorescence staining results revealed the higher fluorescence signals of the CD3⁺CD4⁺ and CD3⁺CD8⁺ T cells in the “EcN-cypate + HBO + laser” group than that in the “LNP-cypate + HBO + laser” group, indicating a better tumor eradication potential for the “EcN-cypate + HBO + laser” group.

Fig. R4 a, Schematic illustration of the experimental schedule. **b**, Representative photos of orthotopic 4T1 tumor-bearing mice with different treatments after laser irradiation (808 nm, 1 W/cm², 15 min). The orthotopic 4T1 tumors are marked by the green circles. **c**, Representative flow cytometric analysis results of cytotoxic T cells (CD3⁺CD8⁺) in the TDLNs. Flow cytometric results showing the levels of **(d)** cytotoxic T cells (CD3⁺CD8⁺), **(e)** mature DC cells (CD11c⁺CD80⁺CD86⁺), and **(f)** M1 macrophages (F4/80⁺CD11c⁺) in the TDLNs. Flow cytometric results showing the levels of **(g)** mature DC cells (CD11c⁺CD80⁺CD86⁺) and **(h)** M1 macrophages (F4/80⁺CD11c⁺) in the spleens. Flow cytometric results showing the levels of **(i)** mature DCs (CD11c⁺CD80⁺CD86⁺) and **(j)** M1 macrophages (F4/80⁺CD11c⁺) in the tumors. **k**, Representative confocal fluorescence images of the immunofluorescence staining results of CD3⁺CD4⁺ and CD3⁺CD8⁺ T cells in tumor slices. The tumors, spleens, and TDLNs were collected from the orthotopic 4T1 tumor-bearing mice in

different groups as indicated. Data are presented as mean \pm standard deviation ($n = 4$) and analyzed by one-way analysis of variance ANOVA (* $P < 0.05$, ** $P < 0.01$, *** $P < 0.001$, **** $P < 0.0001$).

In addition, we detected the levels of cytokines in tumor regions with ELISA, and the “EcN-cypate + HBO + laser” group showed significant higher levels of proinflammatory cytokines including interferon- γ (IFN- γ), interleukin-1 β (IL-1 β), interleukin-6 (IL-6), and tumor necrosis factor- α (TNF- α) than the “LNP-cypate + HBO + laser” group (Fig. R5a), indicating the enhanced killing efficiency of bacteria-mediated PTT. The immunofluorescence staining results revealed that decreased fluorescence signals of CD206 (the marker of M2 macrophages), FoxP3 (the marker of Tregs), and Ly6G (the marker of MDSCs) were observed in the “EcN-cypate + HBO + laser” group (Fig. R5b), indicating the reprogramming of immunosuppressive TME. Despite HBO treatment promoting the intratumoral enrichment of both EcN-cypate and LNP-cypate, EcN-cypate demonstrated prolonged intratumoral retention (at least 48 h), resulting in superior PTT outcomes after repeated laser irradiation. Furthermore, the dead bacteria induced by PTT also contributed to the enhancement of immune responses, and thus **the “EcN-cypate + HBO + laser” group achieved a better immune therapeutic outcome than the “LNP-cypate + HBO + laser” group, indicating the superiority of combining HBO with engineered bacteria (EcN-cypate).**

Fig. R5 a, Intratumoral levels of TNF- α , IFN- γ , IL-1 β , and IL-6 analyzed by ELISA. **b**, Representative immunofluorescence staining results of CD206, FoxP3, and Ly6G in the tumor tissue slices from the orthotopic 4T1 tumor-bearing mice after different treatments as indicated. Statistical data are presented as mean \pm standard deviation ($n = 3$) and analyzed by one-way ANOVA (* $P < 0.05$, ** $P < 0.01$, **** $P < 0.0001$).

Encouraged by the strong systemic immune responses induced by HBO-combined PTT, we further conducted a comprehensive evaluation of therapeutic outcomes in orthotopic 4T1 tumor models (Fig. R6a). Tumor growth curves were recorded for various groups, including PBS (Fig. R6b), “LNP-cypate + laser” (Fig. R6c), “LNP-cypate + HBP + laser” (Fig. R6d), “EcN-cypate + laser” (Fig. R6e), and “EcN-cypate + HBO + laser” (Fig. R6f). As shown in Fig. R6g and h, the “EcN-cypate + HBO + laser” group significantly inhibited the growth of tumors, suggesting the potential of HBO-combined engineered bacteria for cancer therapy. Furthermore, the tumor weight data confirmed the antitumor efficiency of the “EcN-cypate + HBO + laser” group (Fig. R6i). Afterwards, we compared the efficacy of EcN-cypate with that of LNP-cypate in preventing tumor metastasis. The results shown in Fig. R6j and k suggested the reduced lung metastasis in the “EcN-cypate + HBO + laser” group, indicating the potential of such a combined treatment for eliciting long-term immune responses. The changes of body weight proved the biocompatibility of LNP-cypate and EcN-cypate (Fig. R6l), and the H&E and TUNEL staining results revealed the desirable tumor killing efficiency of HBO-combined EcN-cypate-mediated PTT (Fig. R6m).

Fig. R6 a, Scheme showing the experimental schedule. Detailed tumor growth curves of each orthotopic 4T1 tumor-bearing mouse in the (b) PBS, (c) “LNP-cypate + laser”, (d) “LNP-cypate + HBP + laser”, (e) “EcN-cypate + laser”, and (f) “EcN-cypate + HBO + laser” groups. g, Tumor volume changes of orthotopic 4T1 tumor-bearing mice in different groups. h, Photographs of the tumor tissues collected from the orthotopic 4T1 tumor-bearing mice at day 30 after different treatments. i, Tumor weights in different groups at day 30. j, Quantification results of the tumor nodules in the lungs collected from different groups. k, Representative photographs and H&E staining images of the lungs collected from different groups. The tumor nodules were indicated by red arrows. l, Body weight fluctuations of the mice in different groups. m, H&E- and TUNEL assay kit-stained tumor slices of orthotopic 4T1 tumor-bearing mice after different treatments. Statistical data are presented as mean \pm standard deviation ($n = 5$) and analyzed by one-way ANOVA (** $P < 0.01$, *** $P < 0.001$, **** $P < 0.0001$).

In conclusion, our strategy of combing HBO with engineered bacteria (EcN-cypate) for tumor therapy was rationally designed, and EcN-cypate

realized superior intratumoral accumulation and prolonged retention compared with nanoparticles (LNP-cypate) after HBO treatment. The synergy of HBO with EcN-cypate, as opposed to LNP-cypate, led to more efficient systemic immune responses and enhanced therapeutic outcomes, underscoring the novelty and efficacy of our approach.

Comment 2: Figure 6d shows increased temperatures in bodies of mice treated with EcN-cypate + HBO + PTT more broadly vs. other treatment groups. Specifically, EcN-cypate + HBO + PTT also appears to increase temperature in the mouse head region much more significantly vs. treatments without HBO. Does this increased temperature (not just isolated to tumor region) have any physiological effect? Why does temperature increase more broadly with HBO treatment?

Author reply: This is a very excellent comment! We must admit that we have an oversight in our temperature scale bar representation in Figure 6d, where the temperatures in the bodies and heads of mice treated with EcN-cypate + HBO + PTT appeared to increase more broadly. Actually, the temperatures recorded in the mouse head regions and bodies for the “EcN-cypate + HBO + PTT” group were within the normal range for mammals (35 ~ 37 °C). To address this mistake, we have normalized the temperature signals in each image and employed the same temperature scale bar for accurate representation (Fig. R7). Additionally, we have replaced the previous Figure 6d in the revised manuscript (Page 17) to reflect this correction. Furthermore, to verify the effect of HBO treatment on temperature change, we measured the temperature of the tumor-bearing mice before and after HBO treatment. The results suggested that HBO treatment alone had no significant impact on increasing the body temperature (Fig. R8).

Fig. R7 Representative thermal images of tumor-bearing mice with different treatments after receiving NIR laser irradiation (808 nm, 1 W/cm²) for different time periods.

Fig. R8 Representative thermal images of tumor-bearing mice before and after HBO treatment after receiving NIR laser irradiation (808 nm, 1 W/cm²) for 5 min.

Comment 3: 4T1 tumors in the mice models were subcutaneous, making them easily accessible to PTT exposure. NIR lasers can penetrate skin up to a few millimeters. For non-subcutaneous tumors, would EcN-cypate + HBO + PTT be as effective?

Author reply: We deeply appreciate the respected review for raising this very professional comment! Given the limited tissue penetration of laser irradiation, PTT is typically deemed suitable for subcutaneous tumors or superficial malignancies such as melanoma or breast cancer. To investigate the therapeutic potential of the “EcN-cypate + HBO + PTT” strategy on non-subcutaneous tumors, we constructed orthotopic 4T1 breast tumor models. As shown in Fig. R6, the “EcN-cypate + HBO + PTT” group consistently exhibited the best therapeutic outcome among all groups, indicating the promising applications of the “EcN-cypate + HBO + PTT” strategy for certain non-subcutaneous tumors. Thank you very much!

Comment 4: Figure 10l shows that EcN-cypate + HBO + PTT + anti-PD-1 therapy reduces metastatic nodules on mice lungs vs. PBS treatment. It would be helpful to also see the lungs from mice treated with just EcN-cypate + HBO + PTT (no anti-PD-1) to determine fully if long term immune effects are derived from EcN-cypate + HBO + PTT or anti-PD-1 therapy.

Author reply: As suggested, to comprehensively determine whether the observed long-term immune effects result from “EcN-cypate + HBO + PTT” or anti-PD-1 therapy, we re-conducted anti-metastatic experiments. As shown in Fig. R9, the “EcN-cypate + HBO + PTT” group exhibited a reduction in metastatic nodules compared with the control group, probably due to the systemic responses induced by HBO-combined PTT. Moreover, rechallenge tumor models were constructed to assess the long-term immune effects of our strategy (Fig. R10a). Both the “EcN-cypate + HBO + PTT + anti-PD-1” and “EcN-cypate + HBO + PTT” groups revealed a strong capacity in eradicating distant tumors compared with the PBS group (Fig. R10b–d), and the “EcN-cypate + HBO + PTT + anti-PD-1” group realized a satisfactory therapeutic outcome (Fig. R10e–g). Besides, the immunofluorescence staining results suggested the increased CD3⁺CD4⁺ and CD3⁺CD8⁺ T cells in both the “EcN-cypate + HBO + PTT + anti-PD-1” and “EcN-cypate + HBO + PTT” groups (Fig. R10h), indicating the long-term immune effects in the above two groups. Collectively, **both**

the “EcN-cypate + HBO + PTT + anti-PD-1” and “EcN-cypate + HBO + PTT” groups exhibited long-term immune effects due to PTT-induced systemic immune responses (Page 23, Fig. 8), and the “EcN-cypate + HBO + PTT + anti-PD-1” group exhibited an enhanced killing capacity of immune cells after combining with PD-1 blockade therapy. The corresponding figure and related discussions were separately added in the supporting information (Page S31, Fig. S22) and manuscript (Page 28).

Fig. R9 a, Quantification results of the tumor nodules in the lungs collected from different groups. **b**, Respective photographs and H&E staining results of the lungs collected from different groups. The tumor nodules were indicated by red arrows. Statistical data are presented as mean ± standard deviation ($n = 5$) and analyzed by one-way ANOVA (** $P < 0.01$, *** $P < 0.001$).

Fig. R10 a, Schematic illustration of the animal experimental design for evaluating the therapeutic and immunostimulation outcomes of different groups in rechallenge tumor models. Detailed growth curves of each distant tumors in the **(b)** PBS, **(c)** “EcN-cypate + HBO + laser”, and **(d)** “EcN-cypate + HBO + anti-PD-1 + laser” groups. **e**, Average tumor volume changes of distant tumors in different groups. **f**, Photographs showing the distant tumor tissues collected from the 4T1 tumor-bearing mice at day 32 after different treatments. **g**, Distant tumor weights in different groups at day 32. **h**, Representative confocal fluorescence images of the immunofluorescence staining results of CD3⁺CD4⁺ and CD3⁺CD8⁺ T cells in distant tumor slices. Statistical data are presented as mean ± standard deviation ($n = 5$) and analyzed by one-way ANOVA (* $P < 0.05$, *** $P < 0.001$, **** $P < 0.0001$).

Comment 5: How much viable EcN-cypate cells remain in the tumor post-PTT? Data from Figure 4f shows that EcN-cypate can be excited multiple times. Would repeated PTT exposures enhance tumor clearance and/or anti-tumor immune response?

Author reply: This comment raised by the respective reviewer is very constructive! As suggested, we investigated the remaining viable EcN cells in the tumor post PTT. As shown in Fig. R11a, the viability of EcN significantly decreased after a single laser irradiation (“Laser+” group), and few EcN cells were still alive following a second laser exposure (“Laser++” group) (Fig. R11b). We speculated that the dead bacteria caused by repeated PTT could enhance the immune responses induced by ICD. To study the antitumor and immunostimulation capabilities of repeated PTT, we recorded the growth curves of tumors with different treatments (Fig. R11c–e). Notably, the “Laser++” group exhibited superior therapeutic results (Fig. R11f), highlighting the crucial role of repeated laser irradiation. Furthermore, the quantified results of tumor weights, H&E staining images, and TUNEL staining results further validated enhanced tumor killing efficiency after repeated PTT (Fig. R11g and h). To investigate the immune responses following varied laser irradiation intervals, we assessed lung metastasis post PTT. As shown in Fig. R11i, fewer tumor nodules were observed in the “Laser++” group, which was confirmed by the corresponding quantitative data (Fig. R11j). Besides, we also evaluated the distributions of CD3⁺CD4⁺ and CD3⁺CD8⁺ T cells within tumors, and the “Laser++” group also presented the highest levels of these T cells (Fig. R11k), indicating the potentiated immune responses induced by repeated PTT. **Collectively, repeated PTT effectively eliminated tumor cells and induced enhanced immune responses, yielding improved therapeutic outcomes.**

Fig. R11 a, Representative photographs showing EcN colonization behaviors inside the tumors from the 4T1 tumor-bearing mice with different treatments. The numbers (10^4 , 10^2) in the images indicated the dilution factors of the original bacterial suspensions. **b**, Corresponding quantitative bacteria counts in (a). Detailed tumor growth curves of each 4T1 tumor-bearing mouse in the (c) PBS, (d) “Laser+”, and (e) “Laser++” groups. **f**, Tumor volume changes of tumors in different groups. **g**, Tumor weights in different groups at day 30. **h**, H&E- and TUNEL assay kit-stained tumor slices of 4T1 tumor-bearing mice after different treatments. **i**, Respective photographs and H&E staining images of the lungs collected from different groups. **j**, Quantification results of the tumor nodules in the lungs collected from different groups. **k**, Representative confocal fluorescence images of the immunofluorescence staining results of CD3⁺CD4⁺ and CD3⁺CD8⁺ T cells in tumor slices collected at day 8 after different treatments. “Laser-”: The mice were i.v. injected with EcN-cypate for 24 h. “Laser+”: The mice were irradiated by laser (808 nm, 1 W/cm², 15 min) at 24 h post injection of EcN-cypate. “Laser++”: The mice were irradiated by laser (808 nm, 1 W/cm², 15 min) at 24 and 48 h post injection of EcN-cypate. Statistical data in (f), (g), and (j) are presented as mean \pm standard deviation ($n = 4$) and analyzed by

one-way ANOVA (* $P < 0.05$, ** $P < 0.01$, **** $P < 0.0001$).

Responses to Reviewer 2:

General comment: The authors depleted extracellular matrix (ECM) with HBO to promote accumulation and penetration of engineered bacteria within tumor tissues and facilitate immune cell infiltration for enhanced photothermal immunotherapy. Although this manuscript is well-designed and well-written, I do not recommend to accept this manuscript for the following issues. Simply replacing nanoparticles with engineered bacteria is insufficient to justify the publication in Nature Communications.

Author reply: We deeply appreciate the respected reviewer's very insightful and helpful comments/suggestions, which are very important for us to improve our work! We are also extremely grateful for the precious time, invaluable expertise, and superb professionalism the respected reviewer has put in improving the quality of our paper! We have carried out many experiments to address your very insightful suggestions to substantiate the novelty of our work, and the key findings are summarized below:

(1) **Uniqueness of Micrometer-Sized Platform:** While previous studies have explored the synergy of hyperbaric oxygen (HBO) with nanoparticles to enhance the tumor accumulation and penetration of the latter, aiming to achieve improved antitumor therapeutic outcomes, our work introduces a unique perspective. Notably, no existing reports have demonstrated the similar effects when employing micrometer-sized platforms such as engineered bacteria. Our approach reveals the potential of combining HBO with micrometer-sized bacteria, providing insights into enhancing the accumulation and penetration of engineered bacteria for realizing effective cancer therapy. In addition, as shown below, we found that the other two micrometer-sized cellular drugs modified with cypate (i.e., *the cypate-conjugated red blood cell (RBC-cypate)* and *the cypate-conjugated probiotic Bacillus coagulans (BC-cypate)*) **exhibit strikingly different HBO-dependent tumor accumulation behaviors**, as compared with the micrometer-sized EcN-cypate, indicating that a **suitable micrometer size is also important for realizing an enhanced tumor accumulation after HBO treatment. Nevertheless, this finding cannot be accurately predicted without detailed experimental demonstration.**

(2) **Live Cells with Hypoxia-Targeting Capacity:** Different from the “dead” nanoparticles, EcN-cypate, being a live cell, possesses the inherent hypoxia-targeting capability. This distinction prompted us to investigate the in vivo distribution of bacteria after HBO treatment, revealing that they may differ significantly from “dead” nanoparticles. According to your very professional suggestions, we expanded our evaluation to include different platforms, encompassing “dead” nanoparticles (i.e., cypate-containing liposomes, serving as a typical nanosized drug) and the above-mentioned two “live” micrometer-sized cellular drugs (i.e., (1) cypate-conjugated red blood cells (RBC-cypate), aiming to investigate **the influence of size** on drug delivery efficiency under HBO treatment; (2) cypate-conjugated *Bacillus coagulans* (BC-cypate), aiming to study the impact of **hypoxia targeting ability** on engineered bacteria delivery efficiency under HBO treatment), to assess their in vivo distributions and therapeutic outcomes. The detailed experimental results

and discussions are listed in the following comments. The corresponding results revealed that, in addition to the **size factor** as mentioned in the above paragraph, the **dead/live state** as well as the **hypoxia-targeting ability** can significantly affect the tumor accumulation of drugs after HBO treatment, which can be demonstrated by the corresponding experimental results on **the live cells with hypoxia-targeting ability** (cypate-containing liposomes vs. EcN-cypate), as well as **the live cells bearing different hypoxia-targeting abilities** (BC-cypate vs. EcN-cypate). Also, **such a conclusion cannot be drawn without clear experimental demonstration.**

(3) **Potentiated Bacteria-Mediated Cancer Therapy:** Recently, bacteria-mediated cancer therapies have attracted increasing interest due to their innate ability to boost immune responses (*Science* 2022, 378, 858–864.; *Science* 2023, 382, 211–218.; *Chem. Soc. Rev.* 2023, 52, 6617–6643.). **Improving the intratumoral accumulation and penetration of micrometer-sized engineered bacteria is crucial for bacteria-based cancer treatments.** Our innovative strategy addresses this concern by combining engineered bacteria with a **facile and non-invasive HBO treatment**, which may foster the future development of new and effective bacteria-based tumor therapies.

(4) **Specific Merits of Cypate Covalent Conjugation:** There are 5 merits for the covalent cypate conjugation on bacterial cells: (1) **Facile one-step conjugation:** the covalent conjugation of cypate onto the EcN cell surface is very simple, and only requires a one-step mixture. (2) **Excellent biocompatibility:** such a covalent one-step cypate conjugation does not affect the viability of the EcN cells, as evidenced by the results shown in Figure 4f on Page 13. (3) **High conjugation stability and robust immunostimulation effect:** the chemical conjugation approach employed in our work is more stable compared with the commonly adopted bacterial modification strategies based on physical coating. Furthermore, the close and tight conjugation between cypate and the EcN cell wall can potentiate the cell killing effectiveness of the heat generated during PTT, leading to the release of abundant bacterial antigens to stimulate strong immune responses for achieving enhanced immunotherapy. (4) **Simultaneous fluorescence imaging capacity:** cypate can not only achieve NIR light-mediated PTT effect, but also realize excellent fluorescence imaging of engineered bacteria (EcN-cypate) to visualize the real-time intratumoral distribution of bacteria in vivo. (5) **Repeated PTT ability of cypate:** more importantly, EcN-cypate can realize repeated PTT to efficiently eliminate bacteria and tumor cells for achieving improved therapeutic and immunostimulation outcomes (Fig. S16, Page S26).

Collectively, the combination of HBO and engineered bacteria (EcN-cypate) represents a novel strategy, **which is not simply replacing nanoparticles with engineered bacteria, and this strategy may benefit the development of bacteria-mediated tumor therapy.**

Major concerns:

Comment 1: The key issue with this manuscript is lack of novelty. Conceptually, the

findings of this work are trivial. First, HBO has been utilized to promote tumor accumulation and penetration of commercialized nanomedicines, such as Doxil and Abraxane, homemade nanomedicine, and bacteria synthesized melanin for synergistic photothermal immunotherapy. Therefore, the results illustrated in Figures 4, 5, 8 and 9 are nothing new. Second, HBO has been leveraged to deplete ECM. The mechanism by which HBO depletes ECM has been revealed that HBO overcomes hypoxia to suppress cancer associated fibroblasts. From this aspect, the results presented in Figure 2 of the current manuscript are nothing new. Third, HBO has been used to regulate tumor immune microenvironment and facilitate immune cell infiltration. Therefore, most of the results presented in this manuscript is trivial. Furthermore, HBO alone is sufficient to modulate tumor immunosuppressive microenvironment, what are the advantages of adding engineered bacteria and photothermal therapy.

Author reply: These issues pointed out by the respected reviewer are very important! For the first question, we have carried out many experiments to demonstrate the novelty of combining HBO with engineered bacteria.

For the first question, HBO has been employed to enhance the intratumoral accumulation and penetration of **nanomedicines**. However, there is currently no evidence supporting the hypothesis that HBO similarly improves the penetration and accumulation of **micrometer-sized living platforms**. We designed EcN-cypate as a representative bacterial drug and demonstrated the superior therapeutic outcomes of HBO-combined bacteria-mediated cancer therapy. **Our work highlights the critical role of HBO in facilitating the intratumoral delivery of bacteria-based drugs.** Furthermore, to validate the novelty and superiority of our strategy, we compared the intratumoral delivery efficiency of EcN-cypate with three other cypate-based drugs (i.e., (1) cypate-conjugated red blood cells, aiming to investigate the influence of size on drug delivery efficiency under HBO treatment; (2) cypate-conjugated *Bacillus coagulans*, aiming to study the impact of hypoxia targeting ability on the delivery efficiency of engineered bacteria under HBO treatment; (3) cypate-containing liposomes, serving as a typical nanosized drug and aiming to compare the therapeutic outcome with that of micrometer-sized EcN-cypate under HBO treatment). Afterwards, we systematically evaluated the efficacy of EcN-cypate and cypate-containing liposomal nanoparticles in inducing immune responses and promoting tumor eradication when combined with HBO. The detailed results and discussions are presented in the following text.

We synthesized LNP-cypate, a cypate-containing liposomal nanoparticle, as the typical nanodrug. Besides, we also designed two micrometer-sized cellular drugs modified with cypate: the cypate-conjugated red blood cell (RBC-cypate) and the cypate-conjugated probiotic *Bacillus coagulans* (BC-cypate), for comparison with the micrometer-sized EcN-cypate. RBCs, which are larger than EcN with an average size of $\sim 7 \mu\text{m}$, have been widely employed in tumor therapy (*Nat. Commun.* 2021, 12, 2637.; *Adv. Funct. Mater.* 2016, 26, 1757–1768.). Herein, we synthesized RBC-cypate to investigate the influence of size on drug delivery efficiency under HBO treatment. *Bacillus coagulans* (BC), a type of probiotic with a weaker hypoxic tendency than EcN, is usually utilized in intestinal therapy (*Int. J. Mol. Sci.* 2018, 19,

2084.). Herein, we synthesized BC-cypate to explore the impact of hypoxia targeting ability on the delivery efficiency of engineered bacteria under HBO treatment.

According to the previous study (*J. Control. Release* 2023, 357, 222–234.), we synthesized the typical liposome-based cypate nanoparticles (LNP-cypate) using the film-hydration and probe-sonication method. Specifically, hydrogenated soy phosphatidylcholine (HSPC), 1,2-distearoyl-*sn*-glycero-3-phosphoethanolamine-*N*-[methoxy(polyethylene glycol)-2000] (ammonium salt) (DSPE-PEG₂₀₀₀-OMe), cholesterol, and cypate were dissolved in a CHCl₃/CH₃OH mixed solution (CHCl₃/CH₃OH = 65: 35, vol/vol) at a mass ratio of 3: 1: 1: 0.5. Then, the mixture was blown dry by a nitrogen stream, dried under vacuum overnight, and hydrated with 2 mL PBS solution at 65 °C. The hydrated mixture was subjected to probe ultrasonication at the 25% power of an ultrasonic cell crusher (X0-650D, Xianou Tech, China), for 300 s (consisting of 15 cycles, with a 3 s pause after 3 s working per cycle) to obtain LNP-cypate. The synthetic procedures for RBC-cypate and BC-cypate were similar to EcN-cypate. In detail, 0.58 mg EDC•HCl and 0.35 mg NHS were added into the cypate (1 mg)-containing DMF solution, and the mixture was further stirred overnight at ~4 °C to yield the cypate NHS ester-containing solution. Subsequently, the cypate NHS ester-containing solution was separately reacted with 5×10^7 colony forming units (CFU) BC cells or 1×10^7 RBCs in PBS. After 4 h stirring, the BC-cypate or RBC-cypate was obtained by centrifugation and washed by PBS for three times to remove unreacted cypate. The concentrations of cypate in RBC-cypate, BC-cypate, and LNP-cypate were determined using a Duetta fluorescence and absorbance spectrometer (Horiba Scientific, USA).

As shown in Fig. R1a, the TEM result revealed that LNPs-cypate were spherical nanoparticles. DLS result revealed that they had a hydrodynamic diameter of 68.5 ± 5.7 nm (Fig. R1b). The confocal microscopic images validated the successful conjugation of cypate onto the surface of RBCs (Fig. R1c) and BC (Fig. R1d).

Fig. R1 a, TEM image of LNP-cypate. **b**, Hydrodynamic diameter of LNP-cypate.

Confocal fluorescence images of (c) RBC-cypate and (d) BC-cypate.

To investigate the impact of HBO on the intratumoral delivery efficiency of various platforms (EcN-cypate, LNP-cypate, RBC-cypate, and BC-cypate), we assessed cypate distributions in orthotopic 4T1 tumor-bearing mice (Fig. R2a). As shown in Fig. R2b, RBC-cypate exhibited the lowest intratumoral delivery efficiency and fastest metabolism, and **HBO treatment had negligible effect on enhancing the delivery efficiency of RBC-cypate.** The larger size of RBCs (~7 μm) probably prevents RBC-cypate to efficiently penetrate the blood vessels and accumulate in tumor regions. Notably, **HBO treatment slightly increased the intratumoral accumulation of BC-cypate, potentially due to the depletion of dense ECM,** which may limit the penetration of micrometer-sized drugs. However, the limited hypoxia targeting capacity of BC hindered the delivery efficiency of BC-cypate. In contrast, EcN-cypate, which are also micrometer-sized cellular drugs with an average size of 3 μm (which was similar to that (3 μm) of BC-cypate), exhibited significant intratumoral enrichment, which was further improved by HBO treatment. **These results indicate that HBO cannot facilitate the intratumoral delivery efficiency of all micrometer-sized platforms, and the size and hypoxia targeting capacity of drugs play important roles in the delivery process.** Additionally, HBO treatment also promoted the intratumoral delivery efficiency of nanoparticles (LNP-cypate) (Fig. R2b). **However, LNP-cypate displayed lower intratumoral accumulation at 48 h than that at 24 h, possibly due to rapid nanoparticle metabolism. Conversely, EcN-cypate exhibited prolonged intratumoral retention time, lasting for at least 48 h (Fig. R2b), which was beneficial for achieving repeated PTT.** Moreover, the semiquantitative results demonstrated that EcN-cypate realized significantly higher intratumoral accumulation than LNP-cypate at 48 h (Fig. R2c), highlighting the superiority of combining HBO with engineered bacteria (EcN-cypate). Afterwards, by analyzing the fluorescence intensity of major organs, we observed the strongest cypate signal in the liver, suggesting that the drug may undergo hepatic clearance (Fig. R2d). **Collectively, the combination of HBO and engineered bacteria (EcN-cypate) is a novel strategy, which is not simply replacing nanoparticles with engineered bacteria but is rationally designed by considering the size/hypoxia targeting effects and intratumoral retention time of drugs.**

Fig. R2 a, Experimental outline showing the procedures for evaluating the distributions of drugs (EcN-cypate, LNP-cypate, RBC-cypate, and BC-cypate) in orthotopic 4T1 tumor models. **b**, Ex vivo distributions of cypate at different time points (24, 48, and 72 h) post intravenous injection of EcN-cypate, LNP-cypate, RBC-cypate, or BC-cypate. “HBO+”: The mice were treated with HBO (1.5 ATA, 2 h) post injection of drugs. **c**, Semiquantitative distribution results of cypate in the tumors at 48 h post injection in different groups. **d**, Heat map showing the distribution of cypate in the major organs (liver, lung, and kidney). The doses of cypate in all groups (EcN-cypate, LNP-cypate, RBC-cypate, and BC-cypate) were 10 mg/kg.

Although the in vivo distribution results have already demonstrated the novelty of combining HBO with engineered bacteria (EcN-cypate), we further conducted a comparative analysis of the therapeutic outcomes between EcN-cypate and nanoparticle-based treatment using LNP-cypate. Firstly, we investigated the in vitro immune responses induced by PTT mediated by EcN-cypate and LNP-cypate. The

photographs suggested that a substantial portion of EcN was eradicated after laser irradiation due to the photothermal effect (Fig. R3a), which was further confirmed by the quantitative results (Fig. R3b). According to previous studies (*Science* 2022, 378, 858–864.; *Chem. Soc. Rev.* 2023, 52, 6617–6643.), dead bacteria can induce immune responses to enhance immunotherapy. To study the potential immune responses stimulated by dead bacteria and LNP-cypate (or EcN-cypate)-mediated PTT, we separately incubated the immature dendritic cells (DCs) with different suspensions including PBS, “LNP-cypate + 4T1”, “LNP-cypate + 4T1 + laser”, “EcN-cypate + laser” (dead bacteria), and “EcN-cypate + 4T1 + laser” groups. As shown in Fig. R3c and d, the “LNP-cypate + 4T1 + laser” group presented a higher percentage of mature DCs than the PBS and “LNP-cypate + 4T1” groups, owing to the PTT-induced ICD. Notably, the “EcN-cypate + laser” group displayed a higher percentage of mature DCs than the PBS group, attributed to the dead bacteria-induced immune responses. Consequently, the “EcN-cypate + 4T1 + laser” group exhibited the highest percentage of mature DCs among all groups, probably due to a combination of PTT-induced ICD and dead bacteria-induced immune responses. **The above results demonstrated that engineered bacteria (EcN-cypate)-mediated PTT could induce stronger immune responses than nanoparticles (LNP-cypate)-mediated PTT.**

Fig. R3 a, Representative photographs showing EcN colonization behaviors before and after laser irradiation (808 nm, 1 W/cm², 5 min). The numbers (10⁴ and 10²) in the image indicated the dilution factor of the original bacterial suspension. **b**, Quantitative bacteria counts in the EcN-cypate and “EcN-cypate + laser” groups. Data are presented as mean ± standard deviation ($n = 3$) and analyzed by two-tailed Student’s t -test (**** $P < 0.0001$). **c**, Quantitative statistics of mature DCs after different treatments and **(d)** corresponding representative flow cytometric analysis results of mature DCs (CD11c⁺CD80⁺CD86⁺). PBS: The immature DCs were incubated with PBS for 24 h. “LNP-cypate + 4T1”: The 4T1 cells were firstly incubated with LNP-cypate (cypate: 5 μg/mL) for 4 h, and then the suspensions were collected for incubating with immature DCs for 24 h. “LNP-cypate + 4T1 + laser”:

The 4T1 cells were firstly incubated with LNP-cypate (cypate: 5 $\mu\text{g}/\text{mL}$) for 4 h, followed by laser irradiation (808 nm, 1 W/cm^2 , 5 min), and then the suspensions were collected for incubating with immature DCs for 24 h. “EcN-cypate + laser”: EcN-cypate (cypate: 5 $\mu\text{g}/\text{mL}$) suspensions were collected after laser irradiation (808 nm, 1 W/cm^2 , 5 min) and then incubated with immature DCs for 24 h. “EcN-cypate + 4T1 + laser”: The 4T1 cells were firstly incubated with EcN-cypate (cypate: 5 $\mu\text{g}/\text{mL}$) for 4 h, followed by laser irradiation (808 nm, 1 W/cm^2 , 5 min), and then the suspensions were collected for incubating with immature DCs for 24 h. Data are presented as mean \pm standard deviation ($n = 3$) and analyzed by one-way analysis of variance (ANOVA) ($*P < 0.05$, $****P < 0.0001$). “ns” stands for nonsignificant difference.

To further assess the in vivo immune responses induced by EcN-cypate- and LNP-cypate-mediated PTT, we constructed orthotopic 4T1 tumor models (Fig. R4a). After laser irradiation, the tumors became darkened, signifying efficient therapeutic outcomes of PTT (Fig. R4b). As shown in Fig. R4c, we analyzed the cytotoxic T cells ($\text{CD3}^+\text{CD8}^+$) in the tumor-draining lymph nodes (TDLNs), and the “EcN-cypate + HBO + laser” group presented a higher proportion of $\text{CD3}^+\text{CD8}^+$ T cells (39.07%) than that of “LNP-cypate + HBO + laser” (31.65%), demonstrating that the HBO-combined engineered bacteria can induce more effective killing by the immune system. The quantitative analysis further supported this finding, showing that the “EcN-cypate + HBO + laser” group had the highest proportions of cytotoxic T cells, mature dendritic cells (DCs), and M1 macrophages among all groups (Fig. R4d–f), indicating the efficient immune responses induced by HBO-combined PTT. Furthermore, we evaluated the proportions of mature DCs and M1 macrophages in spleens and tumors, and the results demonstrated that the “EcN-cypate + HBO + laser” group induced desirable systemic immune responses (Fig. R4g–j). As shown in Fig. R4k, the immunofluorescence staining results revealed the higher fluorescence signals of the $\text{CD3}^+\text{CD4}^+$ and $\text{CD3}^+\text{CD8}^+$ T cells in the “EcN-cypate + HBO + laser” group than that in the “LNP-cypate + HBO + laser” group, indicating a better tumor eradication potential for the “EcN-cypate + HBO + laser” group.

Fig. R4 a, Schematic illustration of the experimental schedule. **b**, Representative photos of orthotopic 4T1 tumor-bearing mice with different treatments after laser irradiation (808 nm, 1 W/cm², 15 min). The orthotopic 4T1 tumors are marked by the green circles. **c**, Representative flow cytometric analysis results of cytotoxic T cells (CD3⁺CD8⁺) in the TDLNs. Flow cytometric results showing the levels of **(d)** cytotoxic T cells (CD3⁺CD8⁺), **(e)** mature DC cells (CD11c⁺CD80⁺CD86⁺), and **(f)** M1 macrophages (F4/80⁺CD11c⁺) in the TDLNs. Flow cytometric results showing the levels of **(g)** mature DC cells (CD11c⁺CD80⁺CD86⁺) and **(h)** M1 macrophages (F4/80⁺CD11c⁺) in the spleens. Flow cytometric results showing the levels of **(i)** mature DCs (CD11c⁺CD80⁺CD86⁺) and **(j)** M1 macrophages (F4/80⁺CD11c⁺) in the tumors. **k**, Representative confocal fluorescence images of the immunofluorescence staining results of CD3⁺CD4⁺ and CD3⁺CD8⁺ T cells in tumor slices. The tumors, spleens, and TDLNs were collected from the orthotopic 4T1 tumor-bearing mice in

different groups as indicated. Data are presented as mean \pm standard deviation ($n = 4$) and analyzed by one-way analysis of variance ANOVA (* $P < 0.05$, ** $P < 0.01$, *** $P < 0.001$, **** $P < 0.0001$).

In addition, we detected the levels of cytokines in tumor regions with ELISA, and the “EcN-cypate + HBO + laser” group showed significant higher levels of proinflammatory cytokines including interferon- γ (IFN- γ), interleukin-1 β (IL-1 β), interleukin-6 (IL-6), and tumor necrosis factor- α (TNF- α) than the “LNP-cypate + HBO + laser” group (Fig. R5a), indicating the enhanced killing efficiency of bacteria-mediated PTT. The immunofluorescence staining results revealed that decreased fluorescence signals of CD206 (the marker of M2 macrophages), FoxP3 (the marker of Tregs), and Ly6G (the marker of MDSCs) were observed in the “EcN-cypate + HBO + laser” group (Fig. R5b), indicating the reprogramming of immunosuppressive TME. Despite HBO treatment promoting the intratumoral enrichment of both EcN-cypate and LNP-cypate, EcN-cypate demonstrated prolonged intratumoral retention (at least 48 h), resulting in superior PTT outcomes after repeated laser irradiation. Furthermore, the dead bacteria induced by PTT also contributed to the enhancement of immune responses, and thus **the “EcN-cypate + HBO + laser” group achieved a better immune therapeutic outcome than the “LNP-cypate + HBO + laser” group, indicating the superiority of combining HBO with engineered bacteria (EcN-cypate).**

Fig. R5 a, Intratumoral levels of TNF- α , IFN- γ , IL-1 β , and IL-6 analyzed by ELISA. **b**, Representative immunofluorescence staining results of CD206, FoxP3, and Ly6G in the tumor tissue slices from the orthotopic 4T1 tumor-bearing mice after different treatments as indicated. Statistical data are presented as mean \pm standard deviation ($n = 3$) and analyzed by one-way ANOVA (* $P < 0.05$, ** $P < 0.01$, **** $P < 0.0001$).

Encouraged by the strong systemic immune responses induced by HBO-combined PTT, we further conducted a comprehensive evaluation of therapeutic outcomes in orthotopic 4T1 tumor models (Fig. R6a). Tumor growth curves were recorded for various groups, including PBS (Fig. R6b), “LNP-cypate + laser” (Fig. R6c), “LNP-cypate + HBP + laser” (Fig. R6d), “EcN-cypate + laser” (Fig. R6e), and “EcN-cypate + HBO + laser” (Fig. R6f). As shown in Fig. R6g and h, the “EcN-cypate + HBO + laser” group significantly inhibited the growth of tumors, suggesting the potential of HBO-combined engineered bacteria for cancer therapy. Furthermore, the tumor weight data confirmed the antitumor efficiency of the “EcN-cypate + HBO + laser” group (Fig. R6i). Afterwards, we compared the efficacy of EcN-cypate with that of LNP-cypate in preventing tumor metastasis. The results shown in Fig. R6j and k suggested the reduced lung metastasis in the “EcN-cypate + HBO + laser” group, indicating the potential of such a combined treatment for eliciting long-term immune responses. The changes of body weight proved the biocompatibility of LNP-cypate and EcN-cypate (Fig. R6l), and the H&E and TUNEL staining results revealed the desirable tumor killing efficiency of HBO-combined EcN-cypate-mediated PTT (Fig. R6m).

Fig. R6 a, Scheme showing the experimental schedule. Detailed tumor growth curves of each orthotopic 4T1 tumor-bearing mouse in the (b) PBS, (c) “LNP-cypate + laser”, (d) “LNP-cypate + HBP + laser”, (e) “EcN-cypate + laser”, and (f) “EcN-cypate + HBO + laser” groups. g, Tumor volume changes of orthotopic 4T1 tumor-bearing mice in different groups. h, Photographs of the tumor tissues collected from the orthotopic 4T1 tumor-bearing mice at day 30 after different treatments. i, Tumor weights in different groups at day 30. j, Quantification results of the tumor nodules in the lungs collected from different groups. k, Representative photographs and H&E staining images of the lungs collected from different groups. The tumor nodules were indicated by red arrows. l, Body weight fluctuations of the mice in different groups. m, H&E- and TUNEL assay kit-stained tumor slices of orthotopic 4T1 tumor-bearing mice after different treatments. Statistical data are presented as mean \pm standard deviation ($n = 5$) and analyzed by one-way ANOVA (** $P < 0.01$, *** $P < 0.001$, **** $P < 0.0001$).

In conclusion, our strategy of combining HBO with engineered bacteria (EcN-cypate) for tumor therapy was rationally designed, and EcN-cypate

realized superior intratumoral accumulation and prolonged retention compared with nanoparticles (LNP-cypate) after HBO treatment. The synergy of HBO with EcN-cypate, as opposed to LNP-cypate, led to more efficient systemic immune responses and enhanced therapeutic outcomes, underscoring the novelty and efficacy of our approach.

For the second question, we carried out transcriptomic analysis, demonstrating the depletion of ECM (Fig. 2), which elucidates the enhanced intratumoral accumulation and penetration of bacteria after HBO treatment (Fig. 3). Although the mechanism of HBO is studied, Fig. 2 is still necessary for a comprehensive understanding of our study.

Regarding the third question, HBO treatment has previously been employed to enhance antitumor immune therapy. However, HBO can only enhance the intratumoral infiltration of immune cells by depleting the dense ECM. Furthermore, HBO treatment alone is insufficient to elevate the proportions of immunostimulatory cells (e.g., cytotoxicity T cells) to modulate tumor immunosuppressive microenvironment (Fig. 8). The bacteria-mediated PTT (“EcN-cypate + laser”) induced immunogenic cell death (ICD) (Fig. 5) and systemic immune responses, effectively modulating the tumor immunosuppressive microenvironment (Fig. 8). Notably, HBO not only facilitates the accumulation and penetration of engineered bacteria (EcN-cypate) but also depletes the ECM, amplifying the immune responses triggered by EcN-cypate-mediated PTT. Thus, the bacteria-mediated PTT played a primary role, while HBO played an amplification role in modulating tumor immunosuppressive microenvironment. Consequently, the HBO and engineered bacteria strategy **not only expands the application range of HBO therapy but also catalyzes the potential development of innovative bacteria-based tumor therapies in the future.**

Comment 2: The mechanistic studies in insufficient. For instance, the groups in Figure 9 are not enough. The authors need to set 5 groups, similar to the results in Figure 8. It is not clear which therapy plays a decisive role in the combination therapy, since there are so many treatments. Furthermore, there is no correlation between Figure 8 and Figure 9, in terms of molecules, proteins or immune cells. The authors need to establish a tight connection between Figure 8 and Figure 9. Similar problems exist in Figure 2 and Figure 3. The mechanism by which HBO enhances bacteria tumor accumulation and penetration is not clear. Is this mechanism similar to the case of HBO-boosted nanomedicine?

Author reply: We sincerely appreciate the respected reviewer for this very constructive comment! Transcriptomic analysis, a widely employed method **between two groups** (Adv. Mater. 2023, 2304257), was chosen in our study to compare the gene expression between the treated group (“EcN-cypate + HBO + laser”) and the control group (PBS). Given the impracticality of establishing five groups for comparison, we conducted a transcriptomic analysis comparing the “EcN-cypate +

HBO” group with the “PBS” group to underscore the pivotal role of PTT in our approach. As shown in Figs. R7 and 8, no detectable differentially expressed genes (DEGs) associated with cancer cell death or immune responses were found between the “EcN-cypate + HBO” and the “PBS” groups, indicating the insignificant therapeutic effect of the “EcN-cypate + HBO” group. This comparative analysis further elucidated that **PTT plays a decisive role** in the combination therapy, and the HBO treatment enhanced the intratumoral delivery of EcN-cypate (Fig. 6) and the infiltration of immune cells (Fig. 8) for improved therapeutic outcomes (Fig. 8).

We believe that Figures 8 and 9 are closely related. For example, the analysis results of the helper T cells (CD3⁺CD4⁺) and cytotoxic T cells (CD3⁺CD8⁺) in Fig. 8j (Page 23) can be confirmed by the gene expression of “activated T cell proliferation” in Fig. 9c (Page 26). Additionally, we found the activation of “PD-L1 expression and PD-1 checkpoint pathway in cancer” in the treated group (Fig. 9d, Page 26), and thus we combined EcN-cypate with PD-1 blockade therapy to achieve better tumor eradication (Fig. 10, Page 29).

Figures 2 and 3 are closely connected, depicting a sequential progression in our study. Initially, we verified the reduction in dense ECM at the fundamental gene expression level by assessing the changes in the tumor microenvironment following HBO treatment (Fig. 2). **This observation led us to hypothesize that the diminished ECM would enhance the intratumoral penetration and accumulation of micron-sized engineered bacteria.** We subsequently validated this concept through a series of in vitro and in vivo experiments (Fig. 3), leading to the development of HBO combined with engineered bacteria for tumor therapy. Our strategy is based on the idea that HBO promoting bacterial intratumoral enrichment shares similarities with the mechanism that HBO enhances the intratumoral delivery efficiency of nanoparticulate drugs by depleting ECM. However, it is essential to note that the micrometer-sized dimension and hypoxia-targeting capacity of bacteria significantly influenced their intratumoral delivery efficiency (Fig. R2). Thus, EcN-cypate achieved desirable intratumoral penetration and accumulation due to a synergistic combination of the inherent characteristics of bacteria and HBO. As depicted in Fig. R2, our results highlight a distinct advantage of our strategy: **after HBO treatment, bacterial drugs exhibit enhanced intratumoral enrichment and prolonged retention compared to nanomaterials.**

Fig. R7 Dot plot showing the GO enrichment analysis results of the top 30 downregulated DEGs in the 4T1 tumors from the mice in the “EcN-cypate + HBO” and “PBS” groups. $n = 3$ mice per group. Statistical significance was calculated via two-tailed Student’s t -test.

Fig. R8 Dot plot showing the GO enrichment analysis results of the top 30 upregulated DEGs in the 4T1 tumors from the mice in the “EcN-cypate + HBO” and “PBS” groups. $n = 3$ mice per group. Statistical significance was calculated via two-tailed Student’s t -test.

Comment 3: The results cannot support the authors’ claims. For instance, no result

demonstrates that the combination therapy induces a long-term immune effect. To corroborate this conclusion, the authors are suggested to perform a rechallenge tumor model.

Author reply: We deeply appreciate the respected reviewer’s very professional comment! As suggested, we have constructed the rechallenge tumor models (Fig. R9a). The “EcN-cypate + HBO + PTT + anti-PD-1” and “EcN-cypate + HBO + PTT” groups revealed a significant long-term immune effect in eradicating distant tumors compared with the PBS group (Fig. R9b–d). Remarkably, the “EcN-cypate + HBO + PTT + anti-PD-1” group achieved a satisfactory therapeutic outcome (Fig. R9e–g), indicating the long-term immune effect induced by our strategy. Besides, the immunofluorescence staining results suggested the increased CD3⁺CD4⁺ and CD3⁺CD8⁺ T cells within distant tumors in both the “EcN-cypate + HBO + PTT + anti-PD-1” and “EcN-cypate + HBO + PTT” groups (Fig. R9h), indicating their sustained immune effects for killing tumor cells. Collectively, both the “EcN-cypate + HBO + PTT + anti-PD-1” and “EcN-cypate + HBO + PTT” groups exhibited long-term immune effects due to PTT-induced systemic immune responses (Page 23, Fig. 8). Moreover, the “EcN-cypate + HBO + PTT + anti-PD-1” group exhibited an enhanced capacity to amplify the killing potential of immune cells when combined with PD-1 blockade therapy, resulting in an improved long-term immune effect. The corresponding figure and related discussions were separately added in the supporting information (Page S31, Fig. S22) and manuscript (Page 28).

Fig. R9 a, Schematic illustration of the animal experimental design for evaluating the

therapeutic and immunostimulation outcomes of different groups in rechallenge tumor models. Detailed growth curves of each distant tumors in the (b) PBS, (c) “EcN-cypate + HBO + laser”, and (d) “EcN-cypate + HBO + anti-PD-1 + laser” groups. e, Average tumor volume changes of distant tumors in different groups. f, Photographs showing the distant tumor tissues collected from the 4T1 tumor-bearing mice at day 32 after different treatments. g, Distant tumor weights in different groups at day 32. h, Representative confocal fluorescence images of the immunofluorescence staining results of CD3⁺CD4⁺ and CD3⁺CD8⁺ T cells in distant tumor slices. Statistical data are presented as mean ± standard deviation ($n = 5$) and analyzed by one-way ANOVA (* $P < 0.05$, *** $P < 0.001$, **** $P < 0.0001$).

Minor concerns:

Comment 4: There is no discussion in this manuscript. The authors are suggested to compare their results with published results. The contribution of the current study is not clear.

Author reply: Thank you very much for this very important comment! As suggested, we have compared our strategy of combing HBO with engineered bacteria to published studies. In bacteria-mediated therapies (*Int. J. Med. Microbiol.* 2007, 297, 151.; *Adv. Sci.* 2021, 8, 2003572), the administration of bacteria into the body can cause a variety of immune cell responses to achieve antitumor efficacy. However, most of the bacteria are colonized in the core necrosis area of the tumor with less distribution in the outermost area of the tumor. Therefore, the immune cells infiltrated in the tumor may have spatial distribution heterogeneity, resulting in an unsatisfactory antitumor effect. This outcome suggests that the limited infiltration of immune cells constrains the efficiency of bacteria-mediated therapy. **Nevertheless, our strategy addressed this limitation by combining bacteria-mediated therapy with HBO.** Although HBO has been found to boost the delivery efficiency of nanoparticles within tumors (*Nano Today* 2021, 40, 101248), there is currently no evidence supporting the hypothesis that HBO similarly improves the penetration and accumulation of micrometer-sized living platforms, such as bacteria. **We proved that HBO can also facilitate the intratumoral penetration and accumulation of bacteria for the first time, expanding the application range of HBO.** In “Comment 1”, we further demonstrated that the combination of EcN-cypate and HBO represents a rational and innovative design by considering the size/hypoxia targeting effects and intratumoral retention time of drugs.

Comment 5: It is not clear whether the engineered bacteria proliferate within tumors or normal tissues?

Author reply: This is a very excellent comment! As suggested, we tried to assess the proliferation of engineered bacteria within tumors and normal tissues by injecting the bacteria into tumors or skin tissues, However, the above method resulted in a significant loss of samples during injection. In detail, the high tumor interstitial pressure and the loose normal skin tissue could lead to the diffusion of injected liquid samples beyond the intended target area, thereby leading to the difficulty of accurately

determining the bacterial load in the tumor site or normal tissue. Thus we chose a more controlled approach by i.v. injecting the engineered bacteria (EcN-cypate). Afterwards, we have evaluated the distribution of EcN-cypate in major organs and tumors at different time points (Fig. R10a). As shown in Fig. R10b, EcN cells were almost completely cleared by the liver and lung after 48 h, demonstrating that **the engineered bacteria cannot proliferate within normal tissues**. We have demonstrated that EcN-cypate still possessed the proliferative capacity after modification with cypate (Fig. 4f, Page 13), and **EcN could also proliferate within tumors** (Fig. 3, Page 10). Similarly, our investigation revealed that **EcN-cypate exhibited notable proliferative capability within tumors** (Fig. R10c), indicating the robust viability of EcN-cypate.

Fig. R10 a, Representative photographs showing EcN-cypate colonization behaviors at 24 and 48 h post injection. The numbers (10^4 , 10^2) in the image indicated the dilution factor of the original bacterial suspension. **b**, Heat map showing the distribution of EcN-cypate in the major organs (heart, liver, spleen, lung, and kidney) and tumors. **c**, Quantitative bacteria counts in tumors at 24 and 48 h post injection. Data are presented as mean \pm standard deviation ($n = 3$) and analyzed by two-tailed Student's t -test ($*P < 0.05$).

Comment 6: The results presented in Figure 10 are inconsistent with the description. No complete tumor eradication has been achieved in the last group although so many therapies have been combined together.

Author reply: We deeply appreciate this respected reviewer for pointing out this mistake! We employed a facile therapeutic approach, utilizing only two drugs (EcN-cypate and anti-PD-1) and non-invasive procedures involving laser irradiation and HBO treatment. Notably, the “EcN-cypate + HBO + laser + anti-PD-1” group

realized better therapeutic outcomes compared with other groups (Page 29, Fig. 10), leading to substantial tumor ablation. We have accordingly adjusted the corresponding expressions in the manuscript to accurately reflect this result: “In contrast, the “EcN-cypate + HBO + laser + anti-PD-1” treatment exhibited the best therapeutic effect among all the groups and it achieved desirable tumor eradication efficiency”. (Page 27) Thank you so much!

Responses to Reviewer 3:

General comment: In this manuscript, the authors claimed that hyperbaric oxygen (HBO) can deplete the extracellular matrix (ECM) and thus enhance the accumulation and penetration of bacteria within the tumor. Based on that, the authors modify an anaerobe-Escherichia coli Nissle 1917 (EcN) with a photothermal fluorophore (cypate) (EcN-cypate) to achieve photothermal therapy (PTT), which also can induce the immunogenic cancer cell death (ICD) in vitro. Then the combination of PD-1 blockade therapy and EcN-cypate-based PTT after HBO treatment could realize long-term immunosurveillance to inhibit lung metastasis. The design of this work was lack of novelty and some of the data was not solid enough. I don't suggest the publication of this manuscript in Nature Communication. Some questions and suggestions are listed below:

Author reply: We deeply appreciate the respected reviewer for his/her very careful and insightful comments which help us to improve the quality of the manuscript significantly!

While it has been reported that HBO can facilitate the intratumoral delivery efficiency of nanoparticles, **there is no evidence that HBO can also enhance the penetration and accumulation of micrometer-sized and live platforms (e.g., bacteria)**. In this study, we demonstrate that HBO effectively promotes the intratumoral delivery of engineered bacteria (EcN-cypate) for achieving better therapeutic outcomes, which expands the application range of HBO. Additionally, we have carried out additional experiments to further prove the novelty of our strategy in the following comments.

On the other hand, recently, bacteria-mediated cancer therapies have attracted increasing interest due to their innate ability to potentiate immune responses (*Science* 2022, 378, 858–864.; *Science* 2023, 382, 211–218.; *Chem. Soc. Rev.* 2023, 52, 6617–6643.). **Improving the intratumoral accumulation and penetration of micrometer-sized engineered bacteria is crucial for bacteria-based cancer treatments. Our innovative strategy addresses this concern by combining engineered bacteria with a facile and non-invasive HBO treatment, which may foster the future development of new and effective bacteria-based tumor therapies.**

In the following, we have listed the main advantages of our system:

(1) **Uniqueness of Micrometer-Sized Platform:** While previous studies have explored the synergy of hyperbaric oxygen (HBO) with nanoparticles to enhance the tumor accumulation and penetration of the latter, aiming to achieve improved antitumor therapeutic outcomes, our work introduces a unique perspective. Notably, no existing reports have demonstrated the similar effects when employing micrometer-sized platforms such as engineered bacteria. Our approach reveals the potential of combining HBO with micrometer-sized bacteria, providing insights into enhancing the accumulation and penetration of engineered bacteria for realizing effective cancer therapy. In addition, as shown below, we found that the other two

micrometer-sized cellular drugs modified with cypate (i.e., *the cypate-conjugated red blood cell (RBC-cypate)* and the *cypate-conjugated probiotic Bacillus coagulans (BC-cypate)*) **exhibit strikingly different HBO-dependent tumor accumulation behaviors**, as compared with the micrometer-sized EcN-cypate, indicating that a **suitable micrometer size is also important for realizing an enhanced tumor accumulation after HBO treatment. Nevertheless, this finding cannot be accurately predicted without detailed experimental demonstration.**

(2) **Live Cells with Hypoxia-Targeting Capacity:** Different from the **“dead” nanoparticles**, EcN-cypate, being **a live cell**, possesses the inherent **hypoxia-targeting capability**. This distinction prompted us to investigate the in vivo distribution of bacteria after HBO treatment, revealing that they may differ significantly from “dead” nanoparticles. According to your very professional suggestions, we expanded our evaluation to include different platforms, encompassing “dead” nanoparticles (i.e., cypate-containing liposomes, serving as a typical nanosized drug) and the above-mentioned two “live” micrometer-sized cellular drugs (i.e., (1) cypate-conjugated red blood cells (RBC-cypate), aiming to investigate **the influence of size** on drug delivery efficiency under HBO treatment; (2) cypate-conjugated *Bacillus coagulans* (BC-cypate), aiming to study the impact of **hypoxia targeting ability** on engineered bacteria delivery efficiency under HBO treatment), to assess their in vivo distributions and therapeutic outcomes. The detailed experimental results and discussions are listed in the following comments. The corresponding results revealed that, in addition to the **size factor** as mentioned in the above paragraph, the **dead/live state** as well as the **hypoxia-targeting ability** can significantly affect the tumor accumulation of drugs after HBO treatment, which can be demonstrated by the corresponding experimental results on **the live cells with hypoxia-targeting ability (cypate-containing liposomes vs. EcN-cypate)**, as well as **the live cells bearing different hypoxia-targeting abilities (BC-cypate vs. EcN-cypate)**. Also, **such a conclusion cannot be drawn without clear experimental demonstration.**

(3) **Potentiated Bacteria-Mediated Cancer Therapy:** Recently, bacteria-mediated cancer therapies have attracted increasing interest due to their innate ability to boost immune responses (*Science* 2022, 378, 858–864.; *Science* 2023, 382, 211–218.; *Chem. Soc. Rev.* 2023, 52, 6617–6643.). **Improving the intratumoral accumulation and penetration of micrometer-sized engineered bacteria is crucial for bacteria-based cancer treatments.** Our innovative strategy addresses this concern by combining engineered bacteria with a **facile and non-invasive HBO treatment**, which may foster the future development of new and effective bacteria-based tumor therapies.

(4) **Specific Merits of Cypate Covalent Conjugation:** There are 5 merits for the covalent cypate conjugation on bacterial cells: (1) **Facile one-step conjugation:** the covalent conjugation of cypate onto the EcN cell surface is very simple, and only requires a one-step mixture. (2) **Excellent biocompatibility:** such a covalent one-step cypate conjugation does not affect the viability of the EcN cells, as evidenced by the results shown in Figure 4f on Page 13. (3) **High conjugation stability and robust immunostimulation effect:** the chemical conjugation approach employed in our

work is more stable compared with the commonly adopted bacterial modification strategies based on physical coating. Furthermore, the close and tight conjugation between cypate and the EcN cell wall can potentiate the cell killing effectiveness of the heat generated during PTT, leading to the release of abundant bacterial antigens to stimulate strong immune responses for achieving enhanced immunotherapy. (4) **Simultaneous fluorescence imaging capacity:** cypate can not only achieve NIR light-mediated PTT effect, but also realize excellent fluorescence imaging of engineered bacteria (EcN-cypate) to visualize the real-time intratumoral distribution of bacteria in vivo. (5) **Repeated PTT ability of cypate:** more importantly, EcN-cypate can realize repeated PTT to efficiently eliminate bacteria and tumor cells for achieving improved therapeutic and immunostimulation outcomes (Fig. S16, Page S26).

Collectively, the combination of HBO and engineered bacteria (EcN-cypate) represents a novel strategy, **which is not simply replacing nanoparticles with engineered bacteria, and this strategy may benefit the development of bacteria-mediated tumor therapy.**

Comment 1: many papers have proved that anaerobic bacteria can load photosensitizers after chemical modification to perform PDT or PTT on tumors and activate immune response therapy (*Adv. Sci.* 2021, 8, 2003572). It also has been reported that after hyperbaric oxygen (HBO) treatment of tumors, drug delivery efficiency will be improved (*Nano Today* 2021, 40, 101248), and the accumulation and penetration of immune cells in tumor tissues will be increased (*Adv. Sci.* 2021, 8, 2100233). Based on these publications, the authors verified that HBO treatment increases the accumulation of EcN-cypate in tumors and facilitates the intratumoral infiltration of immune cells to realize tumor therapy through PTT and ICD-induced immunotherapy. It seems to be a combination of previous reports without obvious innovation. What is the innovation in this paper?

Author reply: This is a very professional comment! After carefully reading the literature (*Adv. Sci.* 2021, 8, 2003572), we identified a corresponding description “Although the administration of bacteria into the body can cause a variety of immune cell responses to produce anti-tumor efficacy, most of the bacteria are colonized in the core necrosis area of the tumor with less distribution in the outermost area of the tumor. Therefore, the immune cells infiltrated in the tumor may have spatially distributed heterogeneity, resulting in an unsatisfactory anti-tumor effect.”. This result indicated that the limited infiltration of immune cells constrains the efficiency of bacteria-mediated therapy. However, **our strategy addressed the above problem by combining bacteria-mediated therapy with HBO.** Although HBO has been found to boost the delivery efficiency of nanoparticles within tumors (*Nano Today* 2021, 40, 101248), there is currently no evidence supporting the hypothesis that HBO similarly improves the penetration and accumulation of micrometer-sized living platforms, such as bacteria. **We proved that HBO can also facilitate the intratumoral penetration and accumulation of bacteria for the first time, expanding the application range**

of HBO. Furthermore, we demonstrate that the combination of EcN-cypate and HBO is a rational and innovative design by the following experiments.

We synthesized LNP-cypate, a cypate-containing liposomal nanoparticle, as the typical nanodrug. Besides, we also designed two micrometer-sized cellular drugs modified with cypate: the cypate-conjugated red blood cell (RBC-cypate) and the cypate-conjugated probiotic *Bacillus coagulans* (BC-cypate), for comparison with the micrometer-sized EcN-cypate. RBCs, which are larger than EcN with an average size of $\sim 7 \mu\text{m}$, have been widely employed in tumor therapy (*Nat. Commun.* 2021, 12, 2637.; *Adv. Funct. Mater.* 2016, 26, 1757–1768.). Herein, we synthesized RBC-cypate to investigate the influence of size on drug delivery efficiency under HBO treatment. *Bacillus coagulans* (BC), a type of probiotic with a weaker hypoxic tendency than EcN, is usually utilized in intestinal therapy (*Int. J. Mol. Sci.* 2018, 19, 2084.). Herein, we synthesized BC-cypate to explore the impact of hypoxia targeting ability on the delivery efficiency of engineered bacteria under HBO treatment.

According to the previous study (*J. Control. Release* 2023, 357, 222–234.), we synthesized the typical liposome-based cypate nanoparticles (LNP-cypate) using the film-hydration and probe-sonication method. Specifically, hydrogenated soy phosphatidylcholine (HSPC), 1,2-distearoyl-*sn*-glycero-3-phosphoethanolamine-*N*-[methoxy(polyethylene glycol)-2000] (ammonium salt) (DSPE-PEG₂₀₀₀-OMe), cholesterol, and cypate were dissolved in a CHCl₃/CH₃OH mixed solution (CHCl₃/CH₃OH = 65: 35, vol/vol) at a mass ratio of 3: 1: 1: 0.5. Then, the mixture was blown dry by a nitrogen stream, dried under vacuum overnight, and hydrated with 2 mL PBS solution at 65 °C. The hydrated mixture was subjected to probe ultrasonication at the 25% power of an ultrasonic cell crusher (X0-650D, Xianou Tech, China), for 300 s (consisting of 15 cycles, with a 3 s pause after 3 s working per cycle) to obtain LNP-cypate. The synthetic procedures for RBC-cypate and BC-cypate were similar to EcN-cypate. In detail, 0.58 mg EDC•HCl and 0.35 mg NHS were added into the cypate (1 mg)-containing DMF solution, and the mixture was further stirred overnight at $\sim 4 \text{ }^\circ\text{C}$ to yield the cypate NHS ester-containing solution. Subsequently, the cypate NHS ester-containing solution was separately reacted with 5×10^7 colony forming units (CFU) BC cells or 1×10^7 RBCs in PBS. After 4 h stirring, the BC-cypate or RBC-cypate was obtained by centrifugation and washed by PBS for three times to remove unreacted cypate. The concentrations of cypate in RBC-cypate, BC-cypate, and LNP-cypate were determined using a Duetta fluorescence and absorbance spectrometer (Horiba Scientific, USA).

As shown in Fig. R1a, the TEM result revealed that LNPs-cypate were spherical nanoparticles. DLS result revealed that they had a hydrodynamic diameter of $68.5 \pm 5.7 \text{ nm}$ (Fig. R1b). The confocal microscopic images validated the successful conjugation of cypate onto the surface of RBCs (Fig. R1c) and BC (Fig. R1d).

Fig. R1 **a**, TEM image of LNP-cypate. **b**, Hydrodynamic diameter of LNP-cypate. Confocal fluorescence images of **(c)** RBC-cypate and **(d)** BC-cypate.

To investigate the impact of HBO on the intratumoral delivery efficiency of various platforms (EcN-cypate, LNP-cypate, RBC-cypate, and BC-cypate), we assessed cypate distributions in orthotopic 4T1 tumor-bearing mice (Fig. R2a). As shown in Fig. R2b, RBC-cypate exhibited the lowest intratumoral delivery efficiency and fastest metabolism, and **HBO treatment had negligible effect on enhancing the delivery efficiency of RBC-cypate**. The larger size of RBCs ($\sim 7 \mu\text{m}$) probably prevents RBC-cypate to efficiently penetrate the blood vessels and accumulate in tumor regions. Notably, **HBO treatment slightly increased the intratumoral accumulation of BC-cypate, potentially due to the depletion of dense ECM**, which may limit the penetration of micrometer-sized drugs. However, the limited hypoxia targeting capacity of BC hindered the delivery efficiency of BC-cypate. In contrast, EcN-cypate, which are also micrometer-sized cellular drugs with an average size of $3 \mu\text{m}$ (which was similar to that ($3 \mu\text{m}$) of BC-cypate), exhibited significant intratumoral enrichment, which was further improved by HBO treatment. **These results indicate that HBO cannot facilitate the intratumoral delivery efficiency of all micrometer-sized platforms, and the size and hypoxia targeting capacity of drugs play important roles in the delivery process.** Additionally, HBO treatment also promoted the intratumoral delivery efficiency of nanoparticles (LNP-cypate) (Fig. R2b). **However, LNP-cypate displayed lower intratumoral accumulation at 48 h than that at 24 h, possibly due to rapid nanoparticle metabolism. Conversely, EcN-cypate exhibited prolonged intratumoral retention time, lasting for at least 48 h (Fig. R2b), which was beneficial for achieving repeated PTT.** Moreover, the semiquantitative results demonstrated that EcN-cypate realized significantly higher intratumoral accumulation than LNP-cypate at 48 h (Fig. R2c), highlighting the superiority of combining HBO with engineered bacteria (EcN-cypate). Afterwards, by analyzing the fluorescence intensity of major organs, we observed the strongest

cypate signal in the liver, suggesting that the drug may undergo hepatic clearance (Fig. R2d). Collectively, the combination of HBO and engineered bacteria (EcN-cypate) is a novel strategy, which is not simply replacing nanoparticles with engineered bacteria but is rationally designed by considering the size/hypoxia targeting effects and intratumoral retention time of drugs.

Fig. R2 a, Experimental outline showing the procedures for evaluating the distributions of drugs (EcN-cypate, LNP-cypate, RBC-cypate, and BC-cypate) in orthotopic 4T1 tumor models. **b**, Ex vivo distributions of cypate at different time points (24, 48, and 72 h) post intravenous injection of EcN-cypate, LNP-cypate, RBC-cypate, or BC-cypate. “HBO+”: The mice were treated with HBO (1.5 ATA, 2 h) post injection of drugs. **c**, Semiquantitative distribution results of cypate in the tumors at 48 h post injection in different groups. **d**, Heat map showing the distribution of cypate in the major organs (liver, lung, and kidney). The doses of cypate in all groups (EcN-cypate, LNP-cypate, RBC-cypate, and BC-cypate) were 10 mg/kg.

Although the *in vivo* distribution results have already demonstrated the novelty of combining HBO with engineered bacteria (EcN-cypate), we further conducted a comparative analysis of the therapeutic outcomes between EcN-cypate and nanoparticle-based treatment using LNP-cypate. Firstly, we investigated the *in vitro* immune responses induced by PTT mediated by EcN-cypate and LNP-cypate. The photographs suggested that a substantial portion of EcN was eradicated after laser irradiation due to the photothermal effect (Fig. R3a), which was further confirmed by the quantitative results (Fig. R3b). According to previous studies (*Science* 2022, 378, 858–864.; *Chem. Soc. Rev.* 2023, 52, 6617–6643.), dead bacteria can induce immune responses to enhance immunotherapy. To study the potential immune responses stimulated by dead bacteria and LNP-cypate (or EcN-cypate)-mediated PTT, we separately incubated the immature dendritic cells (DCs) with different suspensions including PBS, “LNP-cypate + 4T1”, “LNP-cypate + 4T1 + laser”, “EcN-cypate + laser” (dead bacteria), and “EcN-cypate + 4T1 + laser” groups. As shown in Fig. R3c and d, the “LNP-cypate + 4T1 + laser” group presented a higher percentage of mature DCs than the PBS and “LNP-cypate + 4T1” groups, owing to the PTT-induced ICD. Notably, the “EcN-cypate + laser” group displayed a higher percentage of mature DCs than the PBS group, attributed to the dead bacteria-induced immune responses. Consequently, the “EcN-cypate + 4T1 + laser” group exhibited the highest percentage of mature DCs among all groups, probably due to a combination of PTT-induced ICD and dead bacteria-induced immune responses. **The above results demonstrated that engineered bacteria (EcN-cypate)-mediated PTT could induce stronger immune responses than nanoparticles (LNP-cypate)-mediated PTT.**

Fig. R3 a, Representative photographs showing EcN colonization behaviors before and after laser irradiation (808 nm, 1 W/cm², 5 min). The numbers (10⁴ and 10²) in the image indicated the dilution factor of the original bacterial suspension. **b**, Quantitative bacteria counts in the EcN-cypate and “EcN-cypate + laser” groups. Data are presented as mean \pm standard deviation ($n = 3$) and analyzed by two-tailed Student’s *t*-test (**** $P < 0.0001$). **c**, Quantitative statistics of mature DCs after

different treatments and **(d)** corresponding representative flow cytometric analysis results of mature DCs (CD11c⁺CD80⁺CD86⁺). PBS: The immature DCs were incubated with PBS for 24 h. “LNP-cypate + 4T1”: The 4T1 cells were firstly incubated with LNP-cypate (cypate: 5 µg/mL) for 4 h, and then the suspensions were collected for incubating with immature DCs for 24 h. “LNP-cypate + 4T1 + laser”: The 4T1 cells were firstly incubated with LNP-cypate (cypate: 5 µg/mL) for 4 h, followed by laser irradiation (808 nm, 1 W/cm², 5 min), and then the suspensions were collected for incubating with immature DCs for 24 h. “EcN-cypate + laser”: EcN-cypate (cypate: 5 µg/mL) suspensions were collected after laser irradiation (808 nm, 1 W/cm², 5 min) and then incubated with immature DCs for 24 h. “EcN-cypate + 4T1 + laser”: The 4T1 cells were firstly incubated with EcN-cypate (cypate: 5 µg/mL) for 4 h, followed by laser irradiation (808 nm, 1 W/cm², 5 min), and then the suspensions were collected for incubating with immature DCs for 24 h. Data are presented as mean ± standard deviation ($n = 3$) and analyzed by one-way analysis of variance (ANOVA) (* $P < 0.05$, **** $P < 0.0001$). “ns” stands for nonsignificant difference.

To further assess the in vivo immune responses induced by EcN-cypate- and LNP-cypate-mediated PTT, we constructed orthotopic 4T1 tumor models (Fig. R4a). After laser irradiation, the tumors became darkened, signifying efficient therapeutic outcomes of PTT (Fig. R4b). As shown in Fig. R4c, we analyzed the cytotoxic T cells (CD3⁺CD8⁺) in the tumor-draining lymph nodes (TDLNs), and the “EcN-cypate + HBO + laser” group presented a higher proportion of CD3⁺CD8⁺ T cells (39.07%) than that of “LNP-cypate + HBO + laser” (31.65%), demonstrating that the HBO-combined engineered bacteria can induce more effective killing by the immune system. The quantitative analysis further supported this finding, showing that the “EcN-cypate + HBO + laser” group had the highest proportions of cytotoxic T cells, mature dendritic cells (DCs), and M1 macrophages among all groups (Fig. R4d–f), indicating the efficient immune responses induced by HBO-combined PTT. Furthermore, we evaluated the proportions of mature DCs and M1 macrophages in spleens and tumors, and the results demonstrated that the “EcN-cypate + HBO + laser” group induced desirable systemic immune responses (Fig. R4g–j). As shown in Fig. R4k, the immunofluorescence staining results revealed the higher fluorescence signals of the CD3⁺CD4⁺ and CD3⁺CD8⁺ T cells in the “EcN-cypate + HBO + laser” group than that in the “LNP-cypate + HBO + laser” group, indicating a better tumor eradication potential for the “EcN-cypate + HBO + laser” group.

Fig. R4 a, Schematic illustration of the experimental schedule. **b**, Representative photos of orthotopic 4T1 tumor-bearing mice with different treatments after laser irradiation (808 nm, 1 W/cm², 15 min). The orthotopic 4T1 tumors are marked by the green circles. **c**, Representative flow cytometric analysis results of cytotoxic T cells (CD3⁺CD8⁺) in the TDLNs. Flow cytometric results showing the levels of **(d)** cytotoxic T cells (CD3⁺CD8⁺), **(e)** mature DC cells (CD11c⁺CD80⁺CD86⁺), and **(f)** M1 macrophages (F4/80⁺CD11c⁺) in the TDLNs. Flow cytometric results showing the levels of **(g)** mature DC cells (CD11c⁺CD80⁺CD86⁺) and **(h)** M1 macrophages (F4/80⁺CD11c⁺) in the spleens. Flow cytometric results showing the levels of **(i)** mature DCs (CD11c⁺CD80⁺CD86⁺) and **(j)** M1 macrophages (F4/80⁺CD11c⁺) in the tumors. **k**, Representative confocal fluorescence images of the immunofluorescence staining results of CD3⁺CD4⁺ and CD3⁺CD8⁺ T cells in tumor slices. The tumors, spleens, and TDLNs were collected from the orthotopic 4T1 tumor-bearing mice in

different groups as indicated. Data are presented as mean \pm standard deviation ($n = 4$) and analyzed by one-way analysis of variance ANOVA (* $P < 0.05$, ** $P < 0.01$, *** $P < 0.001$, **** $P < 0.0001$).

In addition, we detected the levels of cytokines in tumor regions with ELISA, and the “EcN-cypate + HBO + laser” group showed significant higher levels of proinflammatory cytokines including interferon- γ (IFN- γ), interleukin-1 β (IL-1 β), interleukin-6 (IL-6), and tumor necrosis factor- α (TNF- α) than the “LNP-cypate + HBO + laser” group (Fig. R5a), indicating the enhanced killing efficiency of bacteria-mediated PTT. The immunofluorescence staining results revealed that decreased fluorescence signals of CD206 (the marker of M2 macrophages), FoxP3 (the marker of Tregs), and Ly6G (the marker of MDSCs) were observed in the “EcN-cypate + HBO + laser” group (Fig. R5b), indicating the reprogramming of immunosuppressive TME. Despite HBO treatment promoting the intratumoral enrichment of both EcN-cypate and LNP-cypate, EcN-cypate demonstrated prolonged intratumoral retention (at least 48 h), resulting in superior PTT outcomes after repeated laser irradiation. Furthermore, the dead bacteria induced by PTT also contributed to the enhancement of immune responses, and thus **the “EcN-cypate + HBO + laser” group achieved a better immune therapeutic outcome than the “LNP-cypate + HBO + laser” group, indicating the superiority of combining HBO with engineered bacteria (EcN-cypate).**

Fig. R5 a, Intratumoral levels of TNF- α , IFN- γ , IL-1 β , and IL-6 analyzed by ELISA. **b**, Representative immunofluorescence staining results of CD206, FoxP3, and Ly6G in the tumor tissue slices from the orthotopic 4T1 tumor-bearing mice after different treatments as indicated. Statistical data are presented as mean \pm standard deviation ($n = 3$) and analyzed by one-way ANOVA (* $P < 0.05$, ** $P < 0.01$, **** $P < 0.0001$).

Encouraged by the strong systemic immune responses induced by HBO-combined PTT, we further conducted a comprehensive evaluation of therapeutic outcomes in orthotopic 4T1 tumor models (Fig. R6a). Tumor growth curves were recorded for various groups, including PBS (Fig. R6b), “LNP-cypate + laser” (Fig. R6c), “LNP-cypate + HBP + laser” (Fig. R6d), “EcN-cypate + laser” (Fig. R6e), and “EcN-cypate + HBO + laser” (Fig. R6f). As shown in Fig. R6g and h, the “EcN-cypate + HBO + laser” group significantly inhibited the growth of tumors, suggesting the potential of HBO-combined engineered bacteria for cancer therapy. Furthermore, the tumor weight data confirmed the antitumor efficiency of the “EcN-cypate + HBO + laser” group (Fig. R6i). Afterwards, we compared the efficacy of EcN-cypate with that of LNP-cypate in preventing tumor metastasis. The results shown in Fig. R6j and k suggested the reduced lung metastasis in the “EcN-cypate + HBO + laser” group, indicating the potential of such a combined treatment for eliciting long-term immune responses. The changes of body weight proved the biocompatibility of LNP-cypate and EcN-cypate (Fig. R6l), and the H&E and TUNEL staining results revealed the desirable tumor killing efficiency of HBO-combined EcN-cypate-mediated PTT (Fig. R6m).

Fig. R6 a, Scheme showing the experimental schedule. Detailed tumor growth curves of each orthotopic 4T1 tumor-bearing mouse in the (b) PBS, (c) “LNP-cypate + laser”, (d) “LNP-cypate + HBP + laser”, (e) “EcN-cypate + laser”, and (f) “EcN-cypate + HBO + laser” groups. g, Tumor volume changes of orthotopic 4T1 tumor-bearing mice in different groups. h, Photographs of the tumor tissues collected from the orthotopic 4T1 tumor-bearing mice at day 30 after different treatments. i, Tumor weights in different groups at day 30. j, Quantification results of the tumor nodules in the lungs collected from different groups. k, Representative photographs and H&E staining images of the lungs collected from different groups. The tumor nodules were indicated by red arrows. l, Body weight fluctuations of the mice in different groups. m, H&E- and TUNEL assay kit-stained tumor slices of orthotopic 4T1 tumor-bearing mice after different treatments. Statistical data are presented as mean \pm standard deviation ($n = 5$) and analyzed by one-way ANOVA (** $P < 0.01$, *** $P < 0.001$, **** $P < 0.0001$).

In conclusion, our strategy of combining HBO with engineered bacteria (EcN-cypate) for tumor therapy was rationally designed, and EcN-cypate

realized superior intratumoral accumulation and prolonged retention compared with nanoparticles (LNP-cypate) after HBO treatment. The synergy of HBO with EcN-cypate, as opposed to LNP-cypate, led to more efficient systemic immune responses and enhanced therapeutic outcomes, underscoring the novelty and efficacy of our approach.

Comment 2: Figure 3b shows that the tumor multicellular spheroids have differences in shape and size in the two groups of HBO- and HBO+. Figure 3c shows that the tumor cell densities are different in these two groups. The penetration and aggregation effect of anaerobic bacteria will of course be poor in HBO- group due to the higher tumor cell density of that. The authors need to re-perform the experiment to exclude the influence of the size and density of tumor multicellular spheroids on the experimental results.

Author reply: We sincerely appreciate the respected reviewer for raising this very constructive comment! As suggested, we have re-performed the experiments and updated the figures (Figs. 3 and S3) along with the relevant descriptions in the revised manuscript (Page 8). To further investigate the influence of the size and density of tumor multicellular spheroids (MCSs) on the experimental results, we incubated MCSs with different sizes (~400 and ~500 μm). As shown in Fig. R7a, the “HBO+” groups exhibited enhanced penetration and accumulation of mCherry-expressing EcN (termed mCherry@EcN), as evidenced by the significantly increased red dots within the MCSs. To strengthen the evidence of HBO’s influence, fluorescence images of MCSs with a size of approximately 500 μm are presented in Fig. R7b, with quantitative results in Fig. R7c further confirming the improved accumulation of mCherry@EcN after HBO treatment. Owing to the hypoxia-targeting nature of EcN cells, they tend to migrate towards oxygen-deficient regions. During the HBO treatment, the oxygen concentration in the external environment of MCS increased, creating a relatively hypoxic condition within the MCS. Consequently, EcN prefers to penetrate the more hypoxic inner region of MCS, thereby achieving enhanced accumulation following HBO treatment.

Fig. R7 a, Confocal microscopic images showing the mCherry@EcN-treated MCSs (size = ~400 or ~500 μm) in the absence or presence of HBO treatment (0 or 2 h). **b**, Confocal fluorescence images showing the mCherry@EcN-treated MCSs (size = ~500 μm) in the absence or presence of HBO treatment (0 or 2 h). **c**, Corresponding quantitative fluorescence intensities of mCherry@EcN inside the 3D tumor spheroids in (b).

Comment 3: EcN can target tumor tissues due to its anaerobic nature. However, after HBO treatment, although the influence of ECM is reduced, it also increases the oxygen content of tumor tissues. The author needs to supplement the staining data of HIF-1α in major organs (heart, liver, spleen, lung, kidney) of mice after HBO treatment to confirm whether tumors can maintain a relatively oxygen-deficient state compared with these major organs after HBO.

Author reply: We deeply appreciate the respected reviewer for raising this very excellent comment! As suggested, we stained the major organs and tumors derived from the tumor-bearing mice with/without HBO treatment. As shown in Fig. R8, the tumor region still displayed the maximal fluorescence signals of HIF-1α, demonstrating that EcN-cyate can still target the hypoxic tumors after HBO

treatment. Moreover, the related figure and discussion have been added in the revised supplementary information (Page S20, Fig. S10) and manuscript (Page 16 and 17), respectively.

Fig. R8 Representative immunofluorescence images showing the HIF-1 α expression in major organs and tumors derived from the tumor-bearing mice with/without HBO treatment.

Comment 4: On page 9, line 192, the author claimed that ‘Besides, we demonstrated that the HBO treatment had a negligible influence on the distribution of EcN within the major organs of the treated mice (Supplementary Fig. 3c and d).’ Figure S3d in Supplementary showed that there was no significant difference in EcN-cypate concentration of the heart, liver, spleen, and kidney after hyper-pressure oxygen treatment. However, EcN-cypate concentration in the lungs increased significantly (it seems to have more than doubled) after HBO treatment. How do the authors explain that?

Author reply: We greatly appreciate the respected reviewer for raising this issue! Notably, these photographs of LB plates are presented as representative examples, and a statistical analysis reveals no significant difference between groups “HBO+” and “HBO-”. In detail, the bacterial counts (CFU/g tissue) in the lungs of the “HBO+” and “HBO-” groups are (3454, 2888, and 2333) and (3090, 3000, and 1777), respectively. Therefore, HBO treatment had a negligible influence on the distribution of EcN within the major organs of the treated mice. Thanks a lot!

Comment 5: In Figure 4g, why does the temperature at the top of the centrifuge tube drop after 1 minute of laser irradiation? And after 4 minutes, and 7 minutes of exposure, the temperature at the top of the centrifuge tube went up compared with that in 1 minute. It seems illogical. It is hoped that the author can re-examine the photothermal data of the material.

Author reply: We sincerely appreciate the respected reviewer’s very constructive, professional, and helpful comment! After carefully comparing the raw data with Fig. 4g, we found that the scale bars in the images were not normalized. The raw data are presented in Fig. R9, and the normalized figures have been now adopted in Fig. 4g (Page 13). We deeply appreciate you for pointing out our mistake to improve the quality of our manuscript!

Fig. R9 Thermal images of EcN-cypate (cypate: 20 $\mu\text{g}/\text{mL}$) dispersions collected after NIR laser irradiation (808 nm, 1 W/cm^2) for different time periods. These images are raw data with normalized scale bar.

Comment 6: In Figure 4i, significant cytotoxicity (the cell survival rate is about 50%) at the concentration of 5 $\mu\text{g}/\text{mL}$ under laser irradiation. However, in Figure 4j, at the same condition, the cell survival rates are well above 50%. The two results are contradictory. How do the authors explain that?

Author reply: This is an excellent comment! The full field of the confocal image was too large, and thus we just selected the partial field of the image. However, we overlooked the variability in the proportion of dead and live cells in different regions of the image, impacting the consistency between Fig. 4i and Fig. 4j. To rectify this oversight and ensure the representation of our findings accurately, we have replaced Fig. 4j with a more representative image on Page 13. We deeply appreciate you for pointing out this discrepancy, which significantly helps us to improve the quality and accuracy of our manuscript!

Comment 7: In figure 5c and figure 5e, The two results are also clearly contradictory. Please check and re-perform the experiments carefully.

Author reply: During the process of immunogenic cancer cell death (ICD), HMGB1 located within the cell nucleus is released into the extracellular regions. Subsequently, there is a decrease in the intracellular contents of HMGB1, while the extracellular

contents of HMGB1 increase. Thus, there appears to be a contradiction between the results presented in Fig. 5c (representing the **extracellular contents** of HMGB1) and Fig. 5e (depicting the **intracellular contents** of HMGB1). Thank you very much for raising such a professional comment!

Comment 8: Figure 6b shows that the distribution of EcN-cypate in major organs (heart, liver, spleen, lung, and kidney) in mice is significantly increased regardless of 24, 48, or 72 hours after being treated with HBO. This seems to contradict the conclusion declared by the author on page 9, line 192. ‘Besides, we demonstrated that the HBO treatment had a negligible influence on the distribution of EcN within the major organs of the treated mice (Supplementary Fig. 3c and d).’ How does the author explain that?

Author reply: This issue pointed out by the respected reviewer is very important! Notably, Fig. 6b shows the semiquantitative distribution results of **cypate** in the tumors and major organs **at 48 h** post injection in different groups. Interestingly, the distribution of LNP-cypate in major organs was also significantly increased following HBO treatment (Fig. R2b and d). These results indicated that the cypate molecules in EcN-cypate and LNP-cypate achieved improved accumulation within major organs after HBO treatment. **We speculated that the bacterial component of EcN-cypate (EcN cell) may be rapidly eliminated due to the defense of the immune system, with or without HBO treatment, leaving the residual cypate molecules staying within the major organs such as liver.** As demonstrated in Fig. R10, the bacterial component of EcN-cypate (EcN cell) exhibited a comparable distribution within major organs in both the “HBO+” and “HBO–” groups. Thus, HBO treatment had a negligible influence on the distribution of EcN within the major organs of the treated mice. Afterwards, at day 2, the major organs in the “EcN-cypate + HBO” group displayed stronger fluorescence signals compared with the “EcN-cypate” group due to the retention of residual cypate molecules within the major organs. Importantly, we assessed the in vivo distribution of EcN-cypate at day 7, revealing negligible fluorescence signals in major organs except the liver (Page S20, Fig. S9). These results suggest that, although cypate molecules remain in the major organs for a certain period, they were ultimately cleared by the mouse body, indicating the biosafety of EcN-cypate.

Fig. R10 Heat map showing the distribution of EcN in the major organs (heart, liver, spleen, lung, and kidney). “HBO+”: The tumor-bearing mice were treated with HBO

(1.5 ATA, 2 h) post injection of EcN-cypate (cypate dose: 10 mg/kg). “HBO–”: The tumor-bearing mice were treated with EcN-cypate (cypate dose: 10 mg/kg) only.

Comment 9: Figure 6d shows the thermal images of tumor-bearing mice with different treatments after receiving NIR laser irradiation. However, the temperature of the mouse body in the EcN-cypate+HBO group is higher than other groups at any time point. Thus It's hard to come to that conclusion the authors claim.

Author reply: This is a very important comment! Considering that the scale bar was not normalized between different groups, there appeared to be an anomaly in the temperature of the mouse body in the EcN-cypate+HBO group, which seemed higher than in other groups. To rectify this, we have normalized the scale bar in each group (Fig. R11), and the relevant figures have been replaced (Fig. 6d, Page 17).

Fig. R11 Representative thermal images of tumor-bearing mice in different groups. Laser irradiation condition: 808 nm, 1 W/cm². These images are raw data with normalized scale bar.

Comment 10: It is hoped that the author could provide the weight change curve and HE staining data of important organs of mice such as the heart, liver, spleen, lung, and kidney in different groups during photothermal therapy combined with checkpoint immunotherapy, to demonstrate the toxicity of EcN-cypate+HOB based photothermal - checkpoint immunotherapy combined therapy.

Author reply: This is a very professional comment! We have recorded the weight change curves previously (Fig. R12), and the corresponding figure has been included in the supplementary data (Page S29, Fig. S20). To demonstrate the safety of our strategy during PTT, we collected the major organs from different groups (PBS, “EcN-cypate + HBO + laser”, and “EcN-cypate + HBO + anti-PD-1 + laser”) for HE staining after laser irradiation. As shown in Fig. R13, no significant damage was observed in the tissues, indicating the good biocompatibility of our therapy. The corresponding figure has been included in the revised supplementary data (Page S30, Fig. S21).

Fig. R12 Body weight fluctuations of the mice in different groups.

Fig. R13 Representative H&E-stained tissue slices of major organs in the 4T1 tumor-bearing BALB/c mice that were sacrificed at 2 d after intravenous injection of PBS or EcN-cypate suspension (cypate dose: 10 mg/kg). The mice in the “EcN-cypate + HBO + laser” and “EcN-cypate + HBO + anti-PD-1 + laser” groups were irradiated by an 808 nm laser (1 W/cm², 15 min) at 24 and 48 h post injection of EcN-cypate.

Comment 11: In the part on experimental methods, the author did not mention some key data such as the size of the tumor at the beginning of treatment. Please complete the experimental procedures.

Author reply: As suggested, we have added the related description “When the tumor volume reached an average volume of $60 \pm 20 \text{ mm}^3$, mice were divided into different groups and treated for different purposes.” in the experimental methods (Page S4). Thank you!

Comment 12: The author mentioned that two blood biochemical experiments were carried out (In Page S10, line 284, ‘In vivo biosafety assessment’ and Page S12, line 334, ‘Evaluation of the in vivo biocompatibility of EcN-cypate’), but only a set of blood biochemical data was found in Figure S10 in the paper, which is confusing. The author should check and proofread it.

Author reply: After carefully checking and proofreading of the experimental descriptions, we identified that we described the same experimental procedures on both Page S10 and Page S12 for twice. We have rectified this error by removing the

redundant description on the original Page S12. We sincerely apologize for this oversight and deeply appreciate you for raising this problem! Thank you so much for the precious time and great effort you have put in improving the quality of our work and the manuscript!

REVIEWERS' COMMENTS

Reviewer #1 (Remarks to the Author):

The authors have done a good job of addressing reviewer concerns. In my opinion, moving from nanoparticles to a micron-scale live bacterial delivery platform is a technical advance worthy of publication.

Reviewer #2 (Remarks to the Author):

Although the authors have performed large amounts of extra experiments, they did not address the major concerns. First, as pointed out by all three reviewers, the manuscript lacks novelty. The authors justified that HBO with engineered bacteria (EcN-cypate) for tumor therapy was rationally designed, as demonstrated in Figure R4-Figure R6. However, it seems that the ratio between group 5 and group 4 resemble that between group 3 and group 2, indicating HBO has no preference for engineered bacteria (EcN-cypate). The main findings of this manuscript are nothing new. Second, as pointed in previous comments, most of the results reported in this manuscript are duplicate of the existing studies and provide no insights in terms of engineered bacteria, HBO, or PTT. Third, the results reported in this study lack of depth. Why and how HBO depletes ECM?

Reviewer #3 (Remarks to the Author):

The authors have addressed the reviewer's comments, and the reviewer suggests accepting this work without additional questions.

Responses to Reviewer 1:

General comment: The authors have done a good job of addressing reviewer concerns. In my opinion, moving from nanoparticles to a micron-scale live bacterial delivery platform is a technical advance worthy of publication.

Author reply: We deeply appreciate the respected reviewer's very kind comment!

Responses to Reviewer 2

General Comment: Although the authors have performed large amounts of extra experiments, they did not address the major concerns. First, as pointed out by all three reviewers, the manuscript lacks novelty. The authors justified that HBO with engineered bacteria (EcN-cypate) for tumor therapy was rationally designed, as demonstrated in Figure R4-Figure R6. However, it seems that the ratio between group 5 and group 4 resemble that between group 3 and group 2, indicating HBO has no preference for engineered bacteria (EcN-cypate). The main findings of this manuscript are nothing new. Second, as pointed in previous comments, most of the results reported in this manuscript are duplicate of the existing studies and provide no insights in terms of engineered bacteria, HBO, or PTT. Third, the results reported in this study lack of depth. Why and how HBO depletes ECM?

Author reply: We deeply appreciate the respected reviewer for these very professional comments! We would also like to express our great appreciation to you for the precious time, invaluable expertise, and superb professionalism you have put in improving the quality of our manuscript!

For the first question, we have calculated the mean values of various cellular proportions and the related ratios in groups 2–5 (termed G2–G5) to demonstrate the preference of HBO for engineered bacteria (EcN-cypate). Herein, Figure R1, previously Figure R4 in the last revision, was chosen as the representative figure for analyzing the ratios between different groups. Notably, the ratios between group 5 and group 4 (i.e., Ratio G5/G4) were significantly higher than those between group 3 and group 2 (i.e., Ratio G3/G2) in various tissue-derived cellular proportions except the CD3⁺CD8⁺ T cells (%) in TDLN and CD11c⁺CD80⁺CD86⁺ DCs (%) in spleen (Table R1). Furthermore, **the statistical analysis of the G3/G2 and G5/G4 ratios demonstrated that these two ratios have a significant difference (** $P < 0.01$),** suggesting the enhanced immune responses induced by the strategy of combining HBO with bacterial drugs (Figure R2). Therefore, the synergy of HBO with EcN-cypate led to efficient systemic immune responses, underscoring the novelty and efficacy of our approach.

For the second question, HBO has been employed to augment the intratumoral accumulation and penetration of **nanomedicines**. However, there is currently no clue/evidence supporting the hypothesis that HBO similarly improves the penetration and accumulation of **micrometer-sized living platforms**. Recently, bacteria-mediated cancer therapies have attracted increasing interest due to their intrinsic capacity to boost immune responses (*Science* 2022, 378, 858–864.; *Science* 2023, 382, 211–218.; *Chem. Soc. Rev.* 2023, 52, 6617–6643.). **Improving the intratumoral accumulation and penetration of micrometer-sized engineered bacteria is crucial for bacteria-based cancer treatments.** Our innovative strategy addresses this concern by combining engineered bacteria with a **facile and noninvasive HBO treatment**, which may foster the future development of new and effective bacteria-based tumor therapies.

For the third question, HBO modulates aberrant mechanical TME by depleting

collagen fiber, collagen I, and fibronectin in ECM, thereby releasing solid stress in solid tumors (*Adv. Sci.* 2021, 8, 2100233). In our research, the ECM-related DEGs (e.g., extracellular region, extracellular space, **extracellular matrix**, external encapsulating structure, and **collagen-containing extracellular matrix**) and some biological process (BP)-related DEGs (e.g., **cell adhesion**, biological adhesion, and **cell–cell adhesion**) were significantly influenced after HBO treatment (Fig. 2d, Page 7). Additionally, the KEGG enrichment analysis results suggested that the DEGs were enriched in the ECM-related pathways (e.g., **ECM receptor interaction**) and cell adhesion-associated signaling pathways (e.g., **focal adhesion**) (Fig. 2e, Page 7). Furthermore, two major components (fibronectin and collagen) of the ECM were stained, and **a significant decrease of fibronectin and collagen** was observed after HBO treatment (Fig. 2f and g, Page 7), further demonstrating the depletion of ECM. Collectively, we demonstrated that HBO can deplete the ECM of tumors and influence the biological process of tumor cells (e.g., cell adhesion), which may benefit the intratumoral delivery and penetration of bacteria.

We believe that our breakthrough in bacteria-mediated photothermal immunotherapy, enhanced by HBO treatment, represents a significant advancement in cancer therapy that will attract the extensive attention from the researchers in the biotechnology field and promote the clinical translation of bacteria-based tumor therapy.

Thank you so much for your great effort in reviewing our manuscript!

Figure R1. **a**, Schematic illustration of the experimental schedule. **b**, Representative photos of orthotopic 4T1 tumor-bearing mice with different treatments after laser irradiation (808 nm, 1 W/cm², 15 min). The orthotopic 4T1 tumors are marked by the green circles. **c**, Representative flow cytometric analysis results of cytotoxic T cells (CD3⁺CD8⁺) in the TDLNs. Flow cytometric results showing the levels of **(d)** cytotoxic T cells (CD3⁺CD8⁺), **(e)** mature DCs (CD11c⁺CD80⁺CD86⁺), and **(f)** M1 macrophages (F4/80⁺CD11c⁺) in the TDLNs. Flow cytometric results showing the levels of **(g)** mature DCs (CD11c⁺CD80⁺CD86⁺) and **(h)** M1 macrophages (F4/80⁺CD11c⁺) in the spleens. Flow cytometric results showing the levels of **(i)** mature DCs (CD11c⁺CD80⁺CD86⁺) and **(j)** M1 macrophages (F4/80⁺CD11c⁺) in the tumors. **k**, Representative confocal fluorescence images of the immunofluorescence staining results of CD3⁺CD4⁺ and CD3⁺CD8⁺ T cells in tumor slices. The tumors, spleens, and TDLNs were collected from the orthotopic 4T1 tumor-bearing mice in

different groups as indicated. Data are presented as mean \pm standard deviation ($n = 4$) and analyzed by one-way analysis of variance ANOVA ($*P < 0.05$, $**P < 0.01$, $***P < 0.001$, $****P < 0.0001$).

Table R1. Mean values of various cellular proportions and related ratios in Figure R1. Ratio G3/G2: The ratio of mean cellular proportions between G3 and G2. Ratio G5/G4: The ratio of mean cellular proportions between G5 and G4.

Tissue	Cellular proportions	G2	G3	G4	G5	Ratio (G3/G2)	Ratio (G5/G4)
TDLN	CD3 ⁺ CD8 ⁺ T cells (%)	26.33	31.51	32.31	37.00	1.20	1.15
	CD11c ⁺ CD80 ⁺ CD86 ⁺ DCs (%)	13.96	16.65	16.91	27.08	1.19	1.60
	F4/80 ⁺ CD11c ⁺ M1 cells (%)	10.26	12.48	12.15	20.99	1.22	1.73
Spleen	CD11c ⁺ CD80 ⁺ CD86 ⁺ DCs (%)	29.69	32.26	32.12	34.98	1.09	1.09
	F4/80 ⁺ CD11c ⁺ M1 cells (%)	13.29	16.69	16.45	23.38	1.26	1.42
Tumor	CD11c ⁺ CD80 ⁺ CD86 ⁺ DCs (%)	21.28	27	24.87	32.28	1.27	1.30
	F4/80 ⁺ CD11c ⁺ M1 cells (%)	32.07	37.08	36.38	46.24	1.16	1.27

Figure R2. Statistical results of the G3/G2 and G5/G4 ratios in Table R1. The significance between two groups was analyzed by two-tailed Student's *t*-test. $**P < 0.01$.

Responses to Reviewer 3

General Comment: The authors have addressed the reviewer's comments, and the reviewer suggests accepting this work without additional questions.

Author reply: We deeply appreciate the respected reviewer's previous precious suggestions for improving our work!